# A GPER-PKA-Centrin axis regulates centrosome numbers and centriole integrity in colon cancer cells
Jeanine Fahrländer[1], Miriam Bühler[1], Julia Martins Shih [1], Catrin Zordick[1], Madleen Busse[2], Markus Becker [1], Gilbert Schönfelder[1,3] & Ailine Stolz [1]✉

Centrosome amplification is a hallmark of colorectal cancer (CRC), yet its molecular origins remain elusive. Protein Kinase A (PKA), anchored to centrosomes, regulates a number of mitotic processes, though its role in maintaining centrosome integrity remains poorly understood. Here, we show that PKA is central to a signaling cascade involving the G protein-coupled estrogen receptor GPER1, which is essential for centrosome integrity in colon cancer cells. Activation of GPER1 by estrogens or a specific agonist triggers Gαs protein-mediated stimulation of adenylyl cyclase, elevating cAMP levels and thereby increasing PKA activity. Excessive activation of GPER1-PKA leads to extra centrosomes with enlarged, displaced Centrin-2 *foci*. Recruited to centrosomes by AKAP450, PKA phosphorylates Centrin-2, predominantly at aberrant centrioles and unexpectedly even outside of mitosis. These findings reveal a GPER1-PKA-Centrin signaling axis in CRC cells that regulates centrosome numbers and centriole integrity, shedding light on centrosome abnormalities that drive neoplastic transformation and tumor progression.

Centrosomes are evolutionarily conserved organelles that act as primary microtubule organizing centers in most eukaryotic cells, orchestrating the microtubule cytoskeleton and governing essential cellular processes such as cell division, cell polarity, motility, migration, and cilia formation[1]. Given their pivotal role, defects in centrosome structure, number, or function are clearly implicated in various human diseases and cancers including colorectal cancer (CRC)[2–4].

Centrosomes are micron-scale structures composed of a pair of centrioles surrounded by a proteinaceous matrix, with γ-tubulin ring complexes being the primary components required for microtubule nucleation. Animal centrioles consist of nine microtubule triplets arranged radially in a cylinder, measuring approximately 200-250 nm in diameter and ~500 nm in length[5,6]. Present in all higher eukaryotes, centrioles organize the pericentriolar material (PCM) and are involved in mitotic spindle assembly and ciliogenesis. Centrins, a group of highly evolutionary conserved calcium-binding proteins, are found in various cellular locations, including primary cilia, the PCM, and both the central and distal regions of the centriole[7]. Mammals possess four isoforms, with Centrin-1 expressed in the testis, ciliated cells, and the retina[8,9], Centrin-4 in ciliated cells in the brain[10], and Centrin-2 and Centrin-3 ubiquitously expressed in somatic cells[11]. Their

early appearance at the distal end of centrioles during biogenesis makes Centrin 1-3 in particular a common marker in immunofluorescence studies[5]. Centriole homeostasis is maintained by a rigid cycle[1,12], permitting duplication only once per cell cycle and assembly of only one new centriole per existing one[13]. During the G1 to S transition of the cell cycle, polo like kinase 4 (Plk4) initiates assembly of one new procentriole orthogonal to each pre-existing mother centriole[14]. Procentrioles elongate during the S and G2 phases, remaining closely associated with the adjacent mother centriole until late M phase, when they disengage[13]. At the end of G2, Aurora A kinase initiates a phosphorylation cascade that culminates in the separation of the two mature centrosomes during mitosis[15].

Despite our extensive knowledge about centriole assembly, little is known about how centriole size and integrity is established and maintained[1]. Centriole number, structure, and size are generally maintained at a relatively constant level within cells[5,16]. Excessive elongation of centrioles has been linked to amplification and chromosome missegregation[5]. However, the molecular mechanisms controlling centriole length are not yet fully understood[17]. While lower eukaryotes demonstrate Centrin-2 being necessary for centriole duplication, positioning, and ciliogenesis[11], its function during vertebrate centriole duplication remains ambiguous and

[1]German Federal Institute for Risk Assessment (BfR), German Centre for the Protection of Laboratory Animals (Bf3R) and Experimental Toxicology, Berlin, Germany. [2]German Federal Institute for Risk Assessment, Biological Safety, Ultrastructural Microscopy Laboratory, Berlin, Germany. [3]Institute of Clinical Pharmacology and Toxicology, Charité - Universitätsmedizin Berlin, corporate member of Freie Universität Berlin, Humboldt-Universität zu Berlin, and Berlin Institute of Health, Berlin, Germany. ✉e-mail: aline.stolz@bfr.bund.de

contentious[11,18–20]. Instead, Centrin-2 appears to primarily regulate centriole length and integrity[7,21,22]. Various kinases phosphorylate Centrin-2, with Aurora A and Protein Kinase A (PKA) specifically targeting serine 170, thereby regulating Centrin-2 stability, centrosome amplification, and centriolar separation[23,24]. Centrin phosphorylation is abnormally elevated in human breast tumors with amplified centrosomes containing unusually large and supernumerary Centrin-labeled centrioles[25], underscoring the necessity for tight regulation of Centrin-2 phosphorylation.

PKA is a serine-threonine protein kinase complex that governs numerous cellular processes through substrate phosphorylation, including those involved in centrosome function, such as Aurora A kinase, Nde1 (neurodevelopment protein 1), centrin, pericentrin and dynein[24,26–28]. The inactive form of this holoenzyme is a heterotetramer, which is composed of a regulatory homodimer (PKA-RIα, RIβ, RIIα, RIIβ) and a catalytic homodimer (PKA-Cα, -Cβ, -Cγ)[29]. The binding of four cAMP molecules to each PKA-R subunit triggers holoenzyme dissociation, activating the catalytic subunits. PKA expression patterns and distribution are primarily dictated by the type of regulatory subunits, with PKA-Rα being ubiquitously expressed compared to the tissue-specific PKA-Rβ isoforms[30]. Spatial and temporal control of PKA is facilitated by its regulatory subunits interacting with A kinase anchoring proteins (AKAPs), which recruit and compartmentalize the holoenzyme in the cytoplasm to specific subcellular sites, close to their substrates[31–33]. AKAP450 (also known as AKAP9/350 or CG-NAP) and PKA localize at the centrosome, suggesting a potential role for AKAP450 in recruiting PKA[27,34–37].

Several signaling pathways may underlie the etiology of centrosome amplification, including those involving p53, APC, or β-Catenin[3]. We have previously identified a novel role for the alternate G protein–coupled estrogen receptor, GPER1, in driving numerical centrosome amplification in colon (cancer)-derived cell lines[38]. GPER1 can be activated by various stimuli, including endogenous estrogens like 17β-estradiol (E2), numerous xenoestrogens as for example Bisphenol A (BPA) and Diethylstilbestrol (DES), anti-estrogens such as tamoxifen (Tam) and fulvestrant (ICI182,780; [ICI]), and synthetic GPER1-selective ligands like G-1[39]. GPER1 triggers rapid signaling cascades, including calcium flux and kinase activation, through multiple G-proteins such as Gαs[40,41], Gαi/o[42,43], and Gβγ[44]. Stimulation of GPER1 by activating ligands induces Gαs-dependent activation of adenylyl cyclase and subsequent release of cAMP, which in turn stimulates centrosome-associated and cAMP-dependent PKA[40,45–48].

This study demonstrates that activation of GPER1 by hormonal cues or receptor-specific agonists initiates a signaling cascade in CRC cells, leading to numerical and structural centrosome defects. This cascade involves Gαs protein-mediated activation of adenylyl cyclase, resulting in elevated levels of cAMP and increased PKA activation. Upon activation, PKA is recruited to centrosomes *via* AKAP450, where it orchestrates the phosphorylation of Centrin-2 at serine 170. Notably, this occurs inappropriately at non-mitotic amplified centrosomes. Furthermore, excessive activation of this GPER1-PKA-Centrin-2 axis leads to centrosome amplification and aberrant centriole structures that lack the characteristic repertoire of centriole-associated proteins. Instead, these structures are marked by enlarged Centrin-2 *foci*, which are clearly separated from normal-sized centrioles. Collectively, our findings provide compelling evidence for the involvement of the GPER1-PKA-Centrin-2 axis in regulating centrosome numbers and centriole integrity in a colon cancer-derived in vitro model.

## Results

### GPER1-triggered centrosome amplification depends on Gαs/adenylyl cyclase signaling in CRC cells

Activated GPER1 causes centrosome amplification and whole chromosomal instability in human colon cells[38]. However, the downstream targets mediating GPER1's role in numerical centrosome integrity are still missing. GPER1 binding to G protein Gαs triggers the activation of PKA, which regulates various cellular processes, including mitosis[37,48,49]. To investigate whether Gαs is involved in GPER1-induced centrosome amplification, we pre-treated HCT116 and HCT-15 cells with the Gαs-selective antagonist

NF449[50] before exposing them to the GPER1-activating estrogens E2, BPA, or DES[38]. Subsequently, we determined the proportion of cells with more than two centrosomes – since a maximum of two is expected per cell—by quantifying centrosomal γ-tubulin *foci* with immunofluorescence microscopy at interphase[38] (Fig. 1a, b). This conservative classification approach may underestimate early amplification events, particularly in G1, but simultaneously minimizes false-positive amplifications, e. g., those caused by highly motile daughter centrioles[51] that may be misidentified as supernumerary centrosomes. Since the centrosome cycle is linked to the cell division cycle[1], we exposed cells for a period of 48 h to ensure that cells undergo more than one cell cycle and thus ensure the establishment of the phenotype[38]. As expected, exposure to estrogens led to an increase in the number of cells with amplified centrosomes compared to the solvent control in both CRC cell lines (Fig. 1a, b). However, pre-exposure to NF449 prevented the estrogen-induced increase in centrosome amplification. Similarly, inhibition of Gαs suppresses centrosome amplification caused by anti-estrogenic GPER1 activators such as ICI182,780 and tamoxifen[52] (Fig. 1c, d, left panel), or the potent and selective GPER1 agonist G-1[53] (Fig. 1c, d, right panel). This suggests that the type of ligand (steroidal/non-steroidal) is less important than the activation of GPER1 per se for G protein-mediated regulation of centrosome numbers. Notably, the suppression of centrosome amplification by Gαs inhibition occurs independently of effects on cell proliferation, which remained unaffected in both cell lines (Supplementary Fig. 1a). Pertussis toxin (PTX) shifts activity from Gαi/o, which inhibits adenylyl cyclase and cAMP production[54], towards Gαs activity, thereby increasing adenylyl cyclase activity[55]. Treatment of HCT116 and HCT-15 cells with PTX induced centrosome amplification in both CRC cell lines (Supplementary Fig. 1b). Thus, GPER1-induced Gαs protein signaling indeed appears to be important for the numerical integrity of the centrosome in colon cancer cells.

Since GPER1/Gαs activates adenylyl cyclase, which in turn increases intracellular cAMP level followed by activation of PKA[40,45,48], we assumed a role of the adenylyl cyclase in regulating centrosome numbers in response to GPER1 activation. To test this hypothesis, we first aimed to confirm that our CRC cells respond to GPER1 stimulation by producing cAMP. We used Promega's inverse cAMP-Glo™ assay, which measures available intracellular ATP levels that decrease with increasing cAMP production in response to adenylyl cyclase activation. Indeed, exposure of CRC cells to the GPER1 activators, E2, BPA, DES, or G-1 decreased available intracellular ATP level compared to the solvent control (Fig. 1e, f). The extent of ATP decrease (and by this, cAMP increase) was comparable to those achieved with forskolin (FSK), a well-established stimulator of adenylyl cyclase and PKA[56]. Pre-treatment with NF449 or the adenylyl cyclase inhibitor SQ 22,536 before GPER1 activation restored elevated cAMP to baseline levels, confirming a GPER1-Gαs-adenylyl cyclase signaling axis in our colon cancer cell system (Supplementary Fig. 1c).

We next examined whether activation of adenylyl cyclase regulates centrosome numbers. Indeed, exposure to FSK increased the numbers of CRC cells with extra centrosomes compared to the control treatment, which is DMSO (Fig. 1g, h). The effect levels were comparable to those observed upon GPER1 stimulation with hormones or specific receptor agonists (Fig. 1a–d). Conversely, pre-treatment of cells with SQ 22,536 reduced centrosome amplification to control levels in the presence of estrogens or G-1 (Fig. 1i–l). Since FSK stimulates adenylyl cyclase and by this, also PKA activity, we hypothesized that PKA may present the key kinase at the end of a GPER1-Gαs-adenylyl cyclase signaling cascade that mediates centrosome amplification. To test this idea, we activated PKA by treatment of the cells with 8-bromo cAMP and quantified the proportion of cells with extra centrosomes as before. Consistent with cAMP binding to and activating PKA[48], we observed a significant increase in centrosome-amplified cells in the presence of 8-Bromo-cAMP (Fig. 1g, h). Overall, these data show that stimulation of GPER1 triggers a signaling cascade, which involves (1) Gαs-dependent activation of adenylyl cyclase, (2) an increase in cAMP production, (3) an activation of PKA, and finally (4) a numerical centrosome amplification in colon cancer cells.

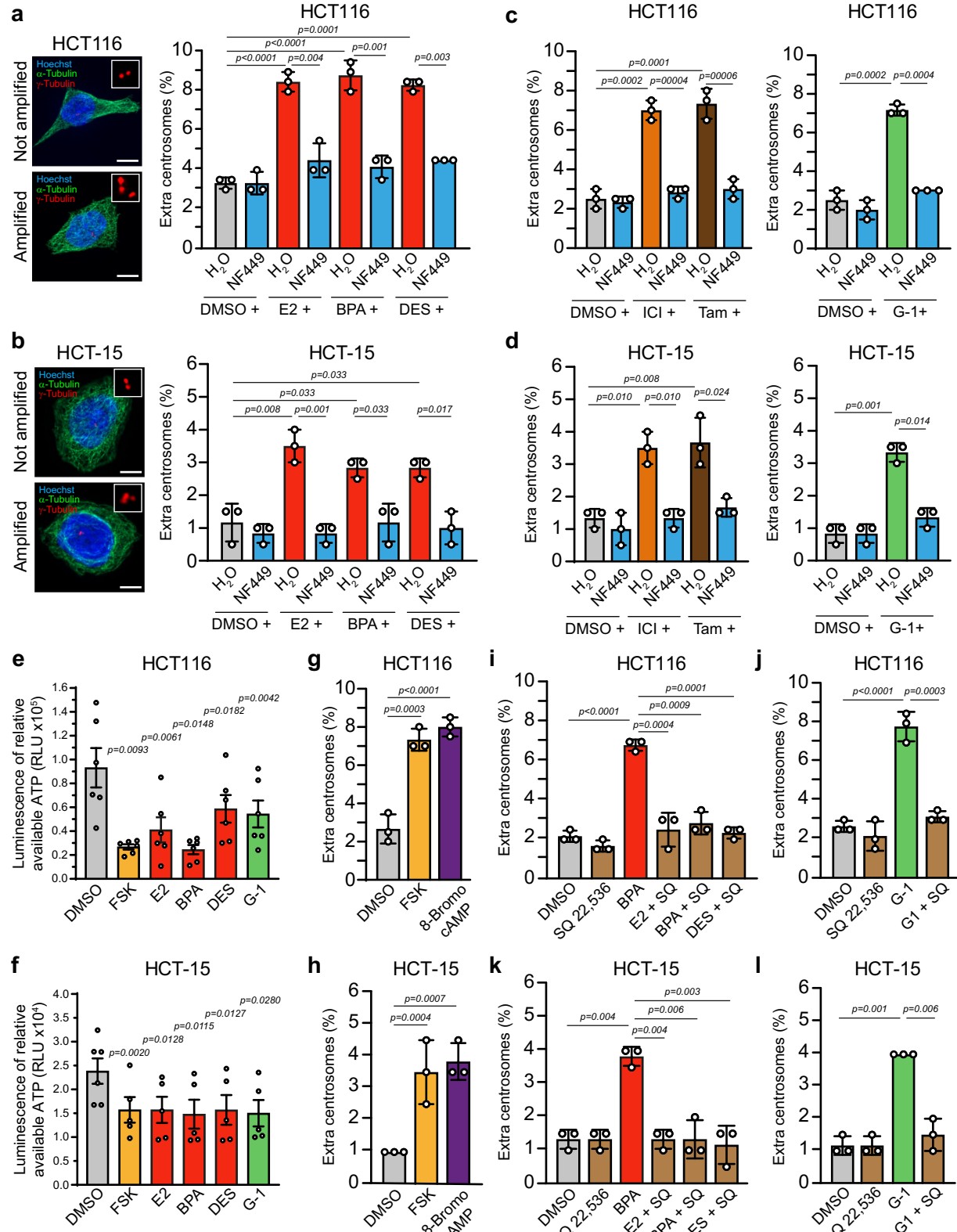

## PKA and its centrosome anchoring protein AKAP450 mediate centrosome amplification upon GPER1 activation

An important effector of cAMP at the centrosome in response to hormonal stimuli is PKA[35]. Upon activation, PKA catalyzes the phosphorylation of a variety of proteins including those associated with the centrosome[24,26–28]. To directly test whether PKA regulates centrosome numbers in response to

GPER1 activation, we partially repressed the catalytic subunit α of PKA in estrogen-treated HCT116 and HCT-15 cells and examined for centrosome amplification (Fig. 2a, b). Knockdown efficiency of *PRKACA* was confirmed in both cell lines *via* Western blot analysis (Supplementary Fig. 2c upper panels). Indeed, *PRKACA* knockdown suppressed centrosome amplification in E2, BPA, and DES treated cells. Likewise, PKA inhibition with PKI

**Fig. 1 | GPER1-triggered centrosome amplification depends on Gαs/adenylyl cyclase signaling in CRC cells.** Representative images and quantification of interphase HCT116 (**a**) and HCT-15 cells (**b**) with more than two centrosomes upon inhibition of Gαs before activation of GPER1 with endogenous or xenoestrogen ligands for 48 h. Scale bar: 10 μm. Quantification of interphase HCT116 (**c**) and HCT-15 cells (**d**) with more than two centrosomes upon inhibition of Gαs before activation of GPER1 with anti-estrogens or synthetic agonist for 48 h. Relative Luminescent Units (RLU) of relative available ATP in HCT116 (**e**) and HCT-15 cells (**f**) upon activation of GPER1 with endogenous, synthetic or xenoestrogen ligands. FSK served as a control for adenylyl cyclase-mediated cAMP production. Quantification of interphase cells with more than two centrosomes upon activation of adenylyl cyclase and PKA for 48 h in HCT116 (**g**) and HCT-15 cells (**h**). Quantification of interphase cells with more than two centrosomes upon inhibition of adenylyl cyclase before activation of GPER1 with endogenous or xenoestrogen ligands (**i**) or synthetic agonist (**j**) for 48 h. BPA (**i**) and G-1 (**j**) served as controls to induce centrosome amplification. **k, l** Same procedure as in (**i** and **j**) but in HCT-15 cells. **Data information:** All graphs show mean ± SD (**a**–**d**) and (**g**–**l**) or mean ± SEM (**e**) and (**f**), and individual data points from three (**a**–**d**), (**g**–**l**), six (**e**) or five (**f**; DMSO: six) different experiments with a total of 600 interphase cells (**a**–**d**) and (**g**–**l**). *P*-values < 0.05 are displayed. The following statistics were applied: *bootstrap* procedure for graphs in (**a**–**d**) and (**g**–**l**) as described in "Materials and Methods" section and Paired *t*test for graphs in (**e**) and (**f**).

(14–22) amide restored centrosome numbers in cells exposed to estrogens (Fig. 2c, d, left panel) or G-1 (Fig. 2c, d, right panel), notably, without affecting cell proliferation (Supplementary Fig. 2a, b). To confirm that all compounds used in this study to activate GPER1 also stimulate PKA, we measured PKA activity in cells treated with estrogens or G-1 by using a PKA activity assay[57]. Consistent with previous studies[45,46], stimulated GPER1 increased PKA activity two- to threefold compared to the solvent control, similar to the increase observed with FSK (Fig. 2e, f). Kinase specificity was confirmed by applying PKI to BPA-exposed cells, which restored PKA activity to control levels. These findings, along with our centrosome amplification assays, demonstrate that PKA drives GPER1-mediated centrosome amplification in colon cancer cells, independently of GPER's role in CRC cell proliferation[38].

PKA localizes to diverse cellular compartments by binding to different A kinase-anchoring proteins (AKAPs)[33]. AKAP450 localizes with PKA at the centrosome, suggesting a potential role as a docking platform for PKA[27,34–37]. By using immunofluorescence microscopy to analyze intracellular PKA localization, we confirmed not only the centrosomal localization of PKA in DMSO-and untreated HCT116 cells (Fig. 2g, h), but also observed a marked increase in centrosomal PKA levels upon GPER1 activation with BPA or G-1 (Fig. 2g). Notably, displaced centrosomes lacked detectable PKA signals (Supplementary Fig. 2d). Furthermore, we showed that its recruitment to the centrosome is dependent on AKAP450, as PKA is absent from centrosomes in AKAP450 knockout cells (Fig. 2h and Supplementary Fig. 2g).

These results raised the question of whether AKAP450 promotes PKA's role in centrosome amplification. To test this hypothesis, we partially repressed *AKAP9* – the gene encoding AKAP450—using gene-specific siRNAs in GPER1-activated cells and quantified the proportion of cells with extra centrosomes (Supplementary Fig. 2e). Partial repression of *AKAP9* was confirmed for both cell lines by Western blot analysis (Supplementary Fig. 2c, lower panels). Indeed, *AKAP9* knockdown suppressed centrosome amplification in CRC cells treated with E2, BPA or DES (Supplementary Fig. 2e). To further strengthen our findings, we assessed the occurrence of supernumerary centrosomes in multiple *AKAP9* knockout clones upon activation of GPER1-PKA signaling by treatment with BPA, G-1, or forskolin (Fig. 2i and Supplementary Fig. 2g). *AKAP9* knockout was confirmed by Western blot analysis (Supplementary Fig. 2f). Notably, centrosome numbers were restored to control levels in these cells supporting a role of AKAP450 in PKA-dependent centrosome amplification.

Taken together, these results support a role for AKAP450 in recruiting PKA to interphase centrosomes and suggest the existence of a GPER1-PKA-AKAP450 axis that may regulate centrosome numbers in CRC cells in response to various GPER1 activators.

## Activation of GPER1 causes enlarged Centrin *foci* at amplified centrosomes

Both size and number of centrioles (two in G1 and four from late S to M, Supplementary Fig. 3a)[58,59] are tightly controlled to ensure proper genome integrity[5]. Interestingly, previous studies describe both an increased size and number of centrioles in breast adenocarcinomas, which could be related to PKA activity[24,25]. Considering our observation that PKA regulates the number of centrosomes in response to GPER1 activation (Fig. 2), we sought to investigate centriole integrity in GPER1-activated CRC cells in more detail. To stimulate GPER1, we treated HCT116 and HCT-15 cells with estrogens, G-1, ICI182,780, or tamoxifen as before (Figs. 1, 2). Centriole integrity was monitored by evaluating the common centriole marker Centrin-2 in immunofluorescence images[11].

We first confirmed centriolar localization of Centrin-2 in CRC cells using immunofluorescence co-staining with γ-tubulin, a well-characterized pericentriolar marker[25,38], and a Centrin-2 monoclonal antibody raised against human full-length Centrin-2 recombinant protein. As expected, we detected two (typical of G1 phase) or four (typical of late S/G2 phase) distinct Centrin-2 signals, which co-localize with two γ-tubulin-positive foci per cell in DMSO-treated controls[58,59] (Fig. 3a). Of note, the occurrence of two γ-tubulin foci in G1-phase cells is consistent with previous findings showing that each centriole can recruit its own PCM (with γ-tubulin as a core marker) following Plk1-dependent modification during the preceding mitosis[59]. The two or four Centrin-2 signals had a similar size, and are therefore be referred to as 'non-enlarged' Centrin foci. Activation of GPER1 not only led to supernumerary centrosomes (Figs. 1 and 2), but one of these centrosomes also contained extraordinarily large Centrin foci compared to those in control cells (Fig. 3a). In order to determine the magnitude of Centrin enlargement in exposed cells, we measured the size of Centrin-2 foci in cells treated with DMSO ('normal') or various GPER1 activators (merged as 'enlarged') (Fig. 3b). To this end, maximum intensity projections of images from each condition capturing the entire Centrin-2 signal were used and the diameter of each Centrin-2 spot was measured. In line with previous studies, we measured around 260–270 nm in diameter for Centrin-2 in DMSO-treated cells[5,6], whereas treatment with GPER1 activators led to a more than three-fold increase in size (890 nm in HCT116, 840 nm in HCT-15). In addition, the enlarged Centrin signals were clearly separated (displaced) from the two normal-sized Centrin spots (Fig. 3a, c). Thereby, the spacing between the enlarged Centrin foci and the normal ones was strikingly asymmetrical and not uniform. The two normal-sized foci in GPER1-activated cells (#1) were separated by a distance similar to that of the paired centrioles in DMSO-treated cells (Ctr), i.e., below 1.0 μm (Fig. 3c). However, our measurements revealed that the distance between the enlarged Centrin foci and the nearest normal-sized foci (#2) increased to 1.22 μm (HCT116) or 1.27 μm (HCT-15). The distance to the more distant normal Centrin foci (#3) was even greater.

Since a recent study reported overly long centrioles marked by Centrin in some cancer cell lines including HCT116 (but not in HCT-15 cells) and centriole over-elongation triggers amplification[5], it is important to precisely determine the proportion of enlarged Centrin signals in the presence or absence of GPER1 activators in both cell lines. To this end, we categorized the cells according to their treatment with DMSO, estrogens (E2, BPA, DES), anti-estrogens (ICI182,780 and tamoxifen), or the GPER1 ligand (G-1), and determined the proportion of centrosome-amplified cells with enlarged centrioles (Fig. 3d–f). Cells with more than two centrosomal γ-tubulin signals were counted as centrosome-amplified[38], and Centrin foci were only classified as 'enlarged' if they were at least twice as large as the other foci in the same cell (see Fig. 3b). As expected, CRC cells treated with DMSO had a low proportion of cells with extra centrosomes, and the

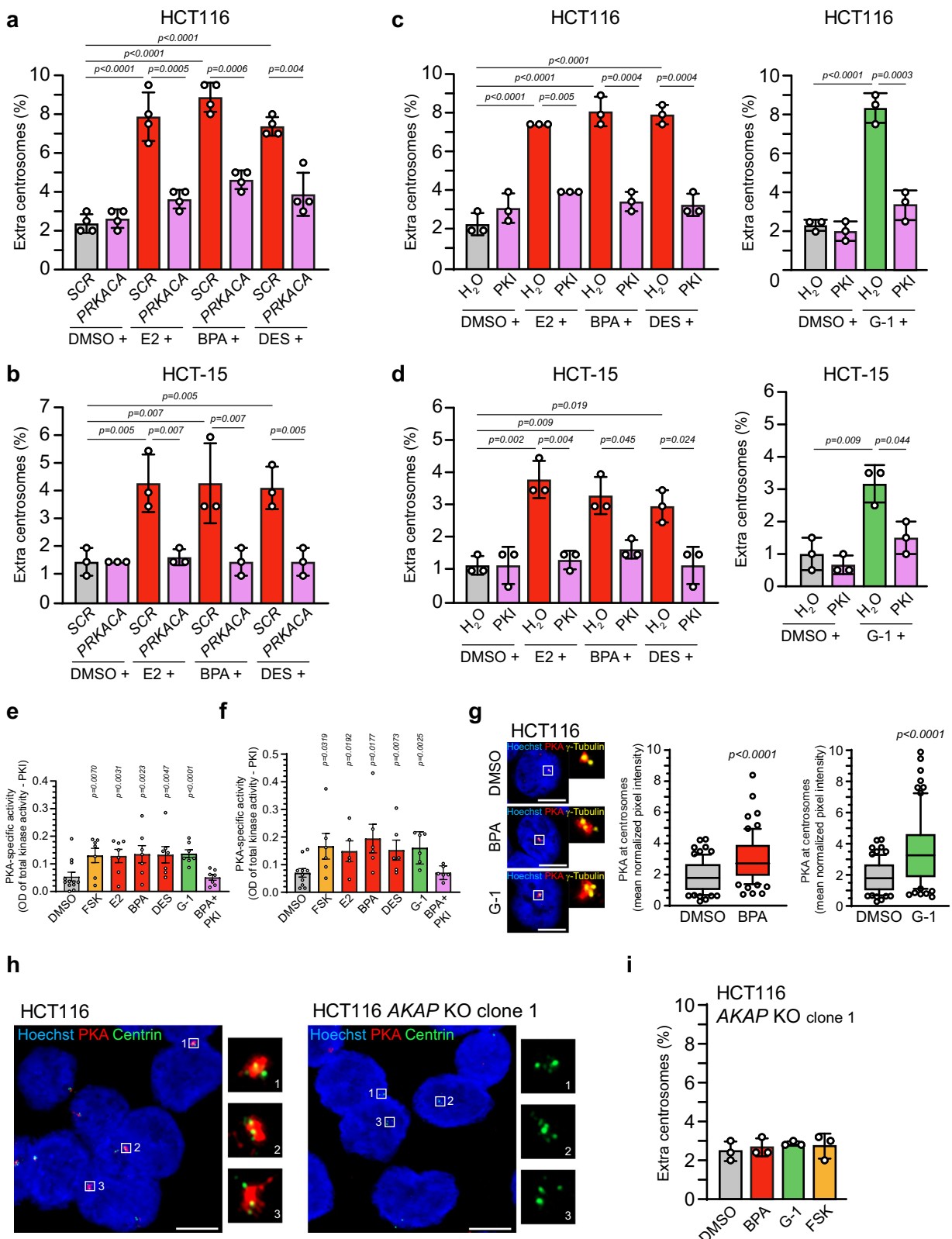

majority of cells (i.e., 97–99%) had a normal centrosome number (Fig. 3d–f). All DMSO-exposed cells with a centrosome number of two (i.e., not amplified) had normal-sized Centrin signals. In line with Marteil et al. [5], our results showed a baseline level of enlarged Centrin *foci* exclusively in DMSO-exposed HCT116 but not in HCT-15 cells[5]. The proportion of enlarged Centrin *foci* in cells with amplified centrosomes increased by up to

56% in HCT116 and 48% in HCT-15 cells in response to GPER1 stimulation (Fig. 3d–f). The type of GPER1 stimulus does not appear to have a major influence, as approximately the same proportion of centrosome-amplified cells with enlarged Centrin signals was achieved in the presence of a hormone stimulus (Fig. 3d) as after activation by GPER1 ligands (Fig. 3e) or anti-estrogens (Fig. 3f).

**Fig. 2 | Recruitment of PKA to centrosomes mediates centrosome amplification upon GPER1 activation.** Quantification of interphase cells with more than two centrosomes upon siRNA mediated knockdown of *PRKACA* in HCT116 (**a**) and HCT-15 cells (**b**) before activation of GPER1 with endogenous or xenoestrogen ligands for 48 h. Quantification of interphase cells with more than two centrosomes upon inhibition of PKA in HCT116 (**c**) and HCT-15 cells (**d**) before activation of GPER1 with endogenous or xenoestrogen ligands or synthetic agonist for 48 h. PKA activity in HCT116 (**e**) and HCT-15 cells (**f**) upon activation of GPER1 with endogenous, synthetic or xenoestrogen ligands for 48 h. FSK served as a control for PKA activation. Results are expressed as PKA-specific activity (total activity minus the activity obtained in samples treated with PKI). **g** Representative images of interphase cells treated with DMSO, BPA or G-1 for 48 h and co-immunostained with anti-PKA Cα and anti-γ-tubulin antibodies and stained with Hoechst3342. Scale bars: 10 μm. Insets show magnified PKA and γ-tubulin signals. Fluorescence intensities of PKA normalized to signals for γ-tubulin from interphase cells treated with DMSO (solvent), xenoestrogen or synthetic for 48 h were plotted as Box and whiskers with 10–90 percentile presentation and individual data points.

$n_{(DMSO)} = 75$; $n_{(BPA)} = 72$; $n_{(DMSO)} = 75$; $n_{(G-1)} = 72$ interphase cells).
**h** Representative images of interphase cells derived from HCT116 parental or HCT116 *AKAP9* knockout cells (*AKAP* KO, clone 1) co-immunostained with anti-PKA anti-Centrin antibodies and stained with Hoechst3342. Scale bars: 10 μm. Insets show magnified PKA and Centrin signals from different interphase cells. Note the absence of PKA in *AKAP9*-depleted cells. **i** Quantification of interphase cells with more than two centrosomes upon activation of GPER1 with a xenoestrogen ligand (BPA), a synthetic agonist (G-1), or PKA with forskolin (FSK) for 48 h in HCT116 *AKAP9* knockout cells (*AKAP* KO, clone 1). **Data information:** All graphs show mean ± SD (a-d) and (i), mean ± SEM (e) and (f), or Box and whiskers with 10–90 percentile (g) and individual data points from three (**b**–**d**), (**g**), and (**i**), four (a), six (**f**; E2: five) or seven (**e**) different experiments with a total of 600 (**b**–**d**), 800 (a) or 1500 interphase cells (**i**). *P* values < 0.05 are displayed. The following statistics were applied: *bootstrap* procedure for graphs in (**a**–**d**, and **i**) as described in "Materials and Methods" section, Paired *t*-test for graphs in (**e**) and (**f**), and Mann–Whitney's test for graphs in (**g**).

To strengthen our results, we included the Centrin 20H5 monoclonal antibody in our studies, which recognizes both human Centrin-2 and Centrin-3 isoforms[60–62]. Again, we observed enlarged Centrin *foci* in cells with GPER1-triggered supernumerary centrosomes (Supplementary Fig. 3b, c). Specifically, both CRC cell lines showed an increase (up to approximately 50%) in the proportion of enlarged Centrin-2/3 spots after hormone treatment, which was significantly higher compared to the DMSO control (Supplementary Fig. 3c).

To distinguish whether the Centrin enlargement generally results from centrosome amplification or is mediated by GPER1-triggered supernumerary centrosomes, we quantified the proportion of cells with enlarged Centrin signal in HCT116 and HCT-15 cells overexpressing *PLK4*, the master regulatory kinase of centrosome duplication[14,38]. According to the literature, ectopic expression of *PLK4* (Supplementary Fig. 3d) caused centrosome amplification in both cell lines (Fig. 3g) with centrioles that often formed rosette-like structures as expected[14] (Supplementary Fig. 3e). However, Plk4-elicited supernumerary centrosomes showed none at all (HCT-15) or not more enlarged Centrin-2 *foci* (HCT116) than control cells (Fig. 3g). From these results it can be deduced that it is not the amplification of the centrosome per se that leads to the Centrin enlargement in CRC cell lines, but that GPER1 activity is needed to establish this structural defect.

**Enlarged Centrin *foci* are associated with aberrant centriole-like structures**

Next, we asked whether the enlarged Centrin-labeled *foci* observed in the presence of an activated GPER-PKA signaling axis represent bona fide extra centrioles or rather ectopic Centrin *foci* resulting from mitotic defects, as previously suggested in the context of aberrant Centrobin signals[63,64]. To address this question, we included additional centriole-specific markers such as CP110, Cep135, and Centrobin[5,38,58,65] in our analysis (Fig. 4a–c). CP110 and Cep135 displayed signal patterns comparable in size and morphology to those observed in solvent-treated control cells (Fig. 4a), suggesting that not all centriole components are affected by the structural alterations, but that these changes may be specific to Centrin. Notably, colocalization of enlarged Centrin *foci* with CP110 was observed in over 50% of BPA- and more than 30% of G-1-treated cells (Fig. 4b). In contrast, Cep135 colocalized with Centrin-2 in fewer than 5% of BPA-treated cells and in none of the G-1-treated cells. These findings indicate that GPER1-activated cells harbor centrioles with enlarged Centrin signals that lack the full complements of structural proteins typically found in mature centrioles. To further clarify the nature of these centriole-like structures, we included Centrobin in our analysis. Only a small fraction (<3%) of cells exhibiting centrosome amplification along with displaced and enlarged Centrin *foci* showed detectable Centrobin staining (Fig. 4c). Thus, the formation of supernumerary centrioles with an aberrant Centrin size and morphology seems to occur largely independently of Centrobin.

To reach a higher resolution of these structural Centrin-defects uncovered by staining with Centrin-2 specific antibodies, we performed expansion microscopy (ExM) according to the protocols published[66–68] (Fig. 4d and Supplementary Fig. 4a). In contrast to conventional fluorescence microscopy (CFM) using structured illumination (Fig. 3a), ExM images (with expansion factors of 3.5x – 4.5x as indicated in the figure legend) revealed that enlarged Centrin signals in GPER1-activated, centrosome-amplified cells contain more than the expected maximum of two Centrin-labeled *foci*. These multiple Centrin *foci* appear to assemble into a "super spot" structure, enlarging at least one of the two centrioles per γ-tubulin signal. Furthermore, this super spot is again spatially separated from the remaining normal-sized Centrin *foci*.

Since our previous CFM and ExM analyses focused primarily on Centrin *foci*, we sought to determine whether the entire centriole ultrastructure is similarly affected. To this end, we performed thin section transmission electron microscopy (TEM) on HCT116 cells treated with DMSO or BPA for 48 h (Fig. 4e, f). Centriole size parameters—including diameter, length, width, and longitudinal area—were quantitatively analyzed using the Fiji software (Fig. 4f and Supplementary Fig. 4b). Consistent with previous reports[1,5,7,69], centrioles in DMSO-treated cells exhibited their characteristic longitudinal architecture (Fig. 4e, panels 3, 4), featuring the barrel-shaped arrangement of microtubule triplets composed of A-, B-, and C-tubules (Fig. 4e, panel 1), as well as the typical (sub-)distal appendages (Fig. 4e, panel 2). Measured centriole dimensions - diameter (~220 nm), length (~409 nm), and width (~198 nm) - were in agreement with published values (Fig. 4f)[5,7]. In contrast, nearly 90% of BPA-treated cells analyzed by TEM (n > 70) displayed aberrant centrioles. The microtubule wall appeared loosened, with individual microtubule bundles detaching from the triplet structure (Fig. 4e, panels 6, 8). Moreover, asymmetric protrusions of the microtubule wall (Fig. 4e, panel 8) and abnormal appendage-like structures (Fig. 4e, panels 5, 7, 9) were observed. Quantitative analysis revealed a significant reduction in centriole length from 409 nm to 322 nm upon BPA treatment, while diameter and width remained largely unchanged compared to DMSO-treated controls (Fig. 4f). The observed shortening of centrioles – with lengths as small as 114 nm—may result in very small or fragmented centriole-like structures, potentially hampering their detection by transmission electron microscopy. In line with this interpretation, among the 72 BPA-treated cells analyzed, we identified only one cell containing three centrioles. By contrast, a markedly higher proportion of BPA-treated cells contained two centrioles (24.1%) compared to DMSO-treated controls, in which the vast majority (>90%) displayed only a single centriole per cell (Supplementary Fig. 4c). Given that GPER1 activation does not appear to affect cell cycle progression or cell proliferation (Supplementary Fig. 1a)[38], the increased number of centrioles compared to control cells is unlikely to reflect cell cycle-dependent fluctuations. Rather, these findings suggest a potential role for GPER1 in promoting amplification of centriole-like structures.

Our data collectively demonstrate that GPER1 activation induces both numerical and structural abnormalities of centrioles. Structurally, this is reflected by the formation of aberrant, enlarged Centrin-positive *foci* that do

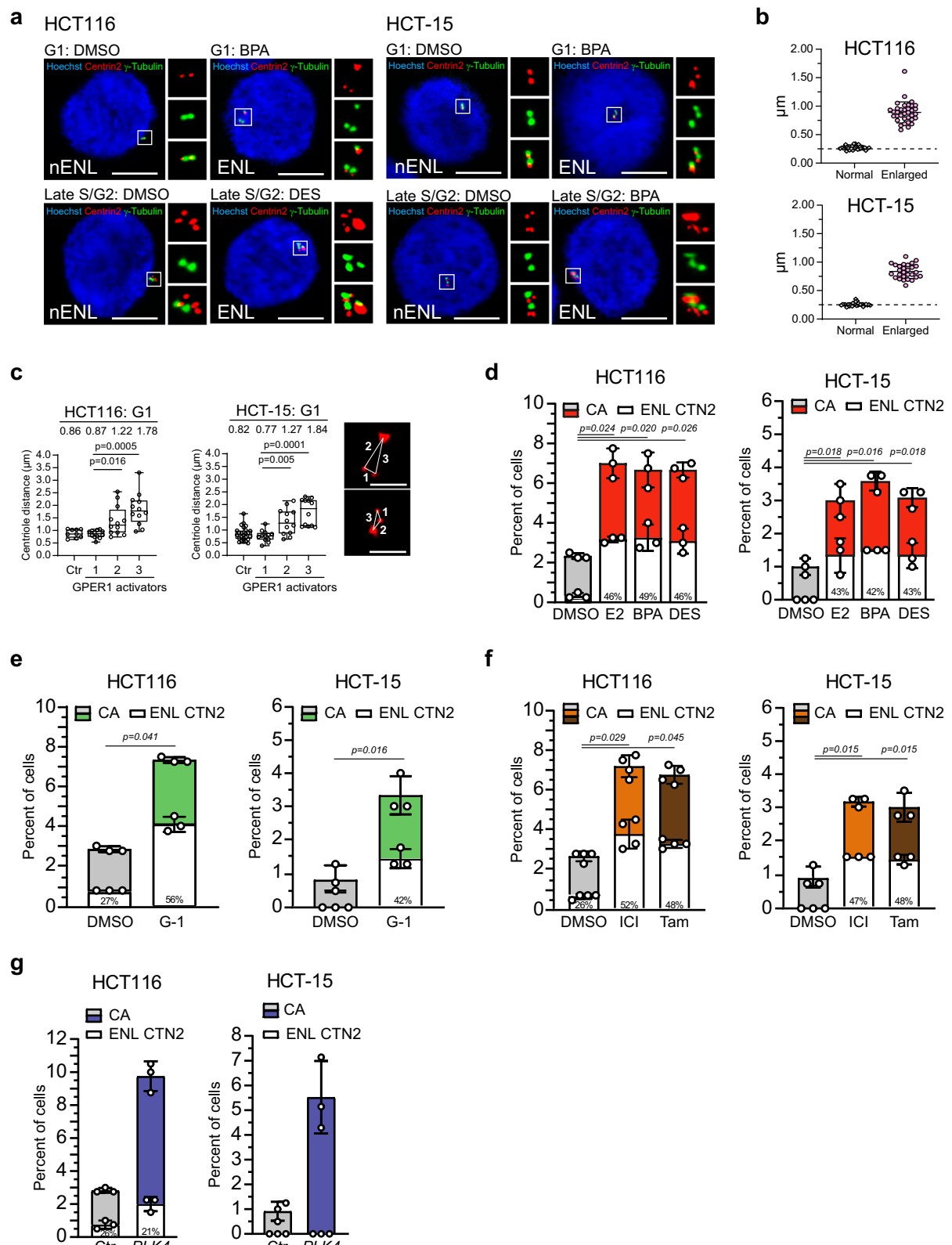

no correspond to intact centrioles but instead may represent immature or structurally compromised centriole-like assemblies. These structures exhibit a reduced centriole length and an incomplete repertoire of canonical centriole proteins, which distinguishes them from both functional centrioles and previously described ectopic Centrobin-containing structures.

## Activated PKA causes enlarged Centrin signals at amplified centrosomes

The present results demonstrate that activation of GPER1 increases intracellular cAMP levels (Fig. 1e, f and Supplementary Fig. 1c) *via* stimulation of adenylyl cyclase, resulting in elevated PKA activity[40,45,46,48,70] (Fig. 2e, f).

**Fig. 3 | Activation of GPER1 causes enlarged Centrin *foci* at amplified centrosomes. a** Representative images of HCT116 and HCT-15 cells upon activation of GPER1 with xenoestrogen ligands for 48 h, with or without enlarged Centrin-2 *foci* (Insets) at G1 or late S/G2 phase. Scale bar: 10 μm. **b** Determination of the diameter of enlarged Centrin-2 *foci* in centrosome amplified cells upon activation of GPER1 with endogenous, synthetic, xenoestrogen or anti-estrogen ligands (merged as 'Enlarged') compared to DMSO ('Normal') for 48 h ($n_{(HCT116, Normal)} = 35$; $n_{(HCT116, Enlarged)} = 35$; $n_{(HCT-15, Normal)} = 28$ cells; $n_{(HCT-15, Enlarged)} = 28$ cells). **c** Centriole distances measured in G1-phase, derived from images used in (**c**) and stained for Centrin-2. Ctr: Distances between paired normal-sized Centrin-2 *foci* of non-amplified, DMSO-exposed control cells; #1: Distances between normal-sized Centrin-2 *foci* of centrosome amplified cells exposed to GPER1 activators; #2 and #3: Distances between the enlarged and the closer (#2) or more distant (#3) normal-sized Centrin-2 *foci* ($n_{(HCT116, DMSO)} = 9$; $n_{(HCT116, #1-3)} = 12$; $n_{(HCT-15, DMSO)} = 25$; $n_{(HCT-15, #1-3)} = 12$). Representative images are given (centrioles, Centrin-2, red; 10 nM BPA). Scale bar: 2 μm. **d–f** HCT116 and HCT-15 cells were treated with endogenous, synthetic, xenoestrogen or anti-estrogen GPER1 ligands for 48 h to induce GPER1 mediated centrosome amplification. The bar graphs show quantification of interphase cells with more than two centrosomes (CA) and the proportion of centrosome amplified cells with enlarged Centrin-2 *foci* (Enlarged, ENL). Percentages of enlarged Centrin-2 *foci* are given. **g** Quantification of interphase cells with more than two centrosomes upon overexpression of *PLK4* for 48 h. Bar graphs show the same as in (**f–i**). Percentages of enlarged Centrin-2 *foci* are given. **Data information:** All graphs show mean ± SD (**b**) and (**d–g**), or median, box and whiskers with min to max presentation (**c**) and individual data points from three (**d**, **e**), (**f**, HCT-15), for four (**f**, HCT116) different experiments with a total of 1200 (**d**), (**e**), (**f**, HCT-15), and (**g**) or 1600 amplified cells (**f**, HCT116). *P*-values for enlarged Centrin-2 < 0.05 are displayed. *P*-values for centrosome amplification are given in the Supplementary Table 1. The following statistics were applied: Paired *t*-test for graph in (**c**) and adapted *bootstrap* procedure for graphs in (**d–g**) as described in "Materials and Methods" section. n = number of cells analyzed.

---

Ultimately, the GPER1-triggered signaling cascade results in centrosome amplification (Figs. 1, 2). Notably, approximately half of the cells with supernumerary centrosomes exhibit enlarged Centrin *foci* accompanied by structurally abnormal centrioles (Figs. 3, 4). All these observations lead us to the question of whether PKA activity is the linchpin of this GPER1-mediated centriole defect. To address this issue, we treated HCT116 and HCT-15 cells with FSK or 8-Bromo cAMP to activate PKA and quantified the proportion of enlarged centriolar Centrin-2 in cells with amplified centrosomes as before. Indeed, similar to GPER1 stimulation (Fig. 3d–f), activation of PKA caused enlarged Centrin-2-labeled centrioles in around 50% of treated cells (Fig. 5a). Thus, our data demonstrate a role of PKA within a GPER1-PKA signaling axis to regulate centrosome numbers and centriole size in CRC cell lines.

A key question that arises from our findings is the role of Centrin in GPER1-PKA-mediated centrosome amplification. If Centrin actively contributes to this process, its deletion should prevent the formation of supernumerary centrosomes despite activation of the GPER1-PKA signaling axis. To test this hypothesis, we generated Centrin-2 knockout cell clones and assessed centrosome amplification by immunofluorescence microscopy under conditions of GPER1-PKA activation (Fig. 5b, c and Supplementary Fig. 5b). Successful knockout of Centrin-2 was confirmed by Western blotting (Supplementary Fig. 5a) and immunostaining for Centrin (Fig. 5b). Quantification of cells with supernumerary centrosomes revealed that GPER1-PKA-induced centrosome amplification was effectively reduced to baseline levels in Centrin-2-deficient cells (Fig. 5c and Supplementary Fig. 5b). These findings suggest that Centrin plays a functional role in the generation of supernumerary centrosomes following activation of the GPER1-PKA signaling cascade.

### Activated PKA phosphorylates Centrin at serine-170
Centrin-2/3 share a highly conserved C-terminal consensus motif of $KKX(pS^{170})X$ for PKA phosphorylation[24,71], and more, phosphorylation of Centrins at position 170 through PKA was associated with elevated PKA activity and abnormal centriole movements in HeLa cells[24]. We thus investigated whether the enlarged Centrin-2-labeled centrioles observed in GPER1-stimulated centrosome amplified CRC cells are also phosphorylated at serine 170. To address this question, given the unavailability of a commercial antibody, we generated antibodies specifically targeting phosphorylated serine-170. We characterized the Centrin-2-pS170 antibody and tested its specificity in our CRC cell systems by Western blot and immunofluorescence analyses.

Given that Centrin-2 is phosphorylated during mitosis[11,24], we initially compared asynchronous and mitotically arrested whole cell lysates of HCT116 and HCT-15 with regard to serine 170-phosphorylated Centrin-2 using immunoblot assays. We revealed a clear band for phosphorylated Centrin-2 around 25 kDa exclusively in mitotically arrested cells using the anti-Centrin-2-pS170 antibody (Fig. 5d). As expected, the molecular weight of phosphorylated Centrin-2 shifted by a few kDa compared to total Centrin, for which we were able to detect bands around 20 kDa using the monoclonal anti-Centrin 20H5 antibody[24]. *CETN2*-specific knockdown experiments verified the specificity of both anti-Centrin (Fig. 5e) and anti-Centrin-2-pS170 antibodies (Fig. 5f) using Western blot analysis. After treatment with λ-phosphatase, the phosphorylation-specific antibody was no longer able to detect its cognate form of phosphorylated Centrin-2 (Fig. 5g). To exclude cross-reactivity against non-phosphorylated Centrin, we performed Centrin-pS170 antibody neutralization experiments by Western blot and immunofluorescence analysis (Fig. 5h, i). We used a phosphopeptide covering the region around S170 in the human Centrin-2 sequence equally used for phospho-Centrin antibody production and the corresponding non-phosphopeptide as a control. Immunoblot analysis of HCT116 whole cell lysates showed a band around 25 kDa exclusively in mitotically arrested cells when using the anti-Centrin-pS170 antibody pre-incubated with the non-phosphopeptide (Fig. 5h). No phospho-Centrin band was detected in mitotic lysates with the neutralized Centrin-pS170 antibody.

Since these results demonstrate the specificity of anti-Centrin-pS170 for phosphorylated Centrin at the protein level, we investigated its specificity with respect to centriole localization using co-immunostainings with total Centrin antibodies (Fig. 5i). To this end, we pre-incubated anti-Centrin-pS170 with (non)-phosphopeptides as before and monitored centriole localization in mitotically arrested HCT116 cells. When the non-phosphopeptide was used, the cells stained intensely for Centrin-2-pS170 in prometaphase and to a lesser extent in anaphase, as reported elsewhere[24]. phospho-Centrin spots co-localize with total Centrin at mitotic centrioles. However, prometaphase cells stained with the neutralized Centrin-pS170 antibody displayed no centriole signal, while total Centrin was still detectable (Fig. 5i). Careful analysis of timely Centrin-2 phosphorylation during mitotic progression revealed interphase centrioles did not stain with the Centrin-pS170 antibody, while mitotic cells (i.e., prometaphase and metaphase cells) stained intensively for phosphorylated Centrin, as expected[24] (Fig. 5j). In contrast to phosphorylated Centrin-2, total Centrin was detectable throughout all stages of the cell cycle. Notably, inhibition of PKA led to a reduced proportion of phospho-Centrin-2 at mitotic centrosomes (Supplementary Fig. 5c, d).

Combined immunoblot and immunofluorescence studies demonstrate that the Centrin-pS170 antibody specifically detects centriolar Centrin phosphorylated at early stages of mitosis by PKA or a kinase with similar substrate specificity[23,24].

### Enlarged Centrin signals are atypically phosphorylated at amplified centrosomes during interphase
Next, we addressed the crucial question of whether the abnormally sized Centrin at centrioles, observed in GPER1-stimulated amplified interphase centrosomes, is phosphorylated at serine 170. To this end, we stimulated GPER1 or PKA activity in CRC cells and quantified the proportion of enlarged centrioles containing phosphorylated Centrin-2.

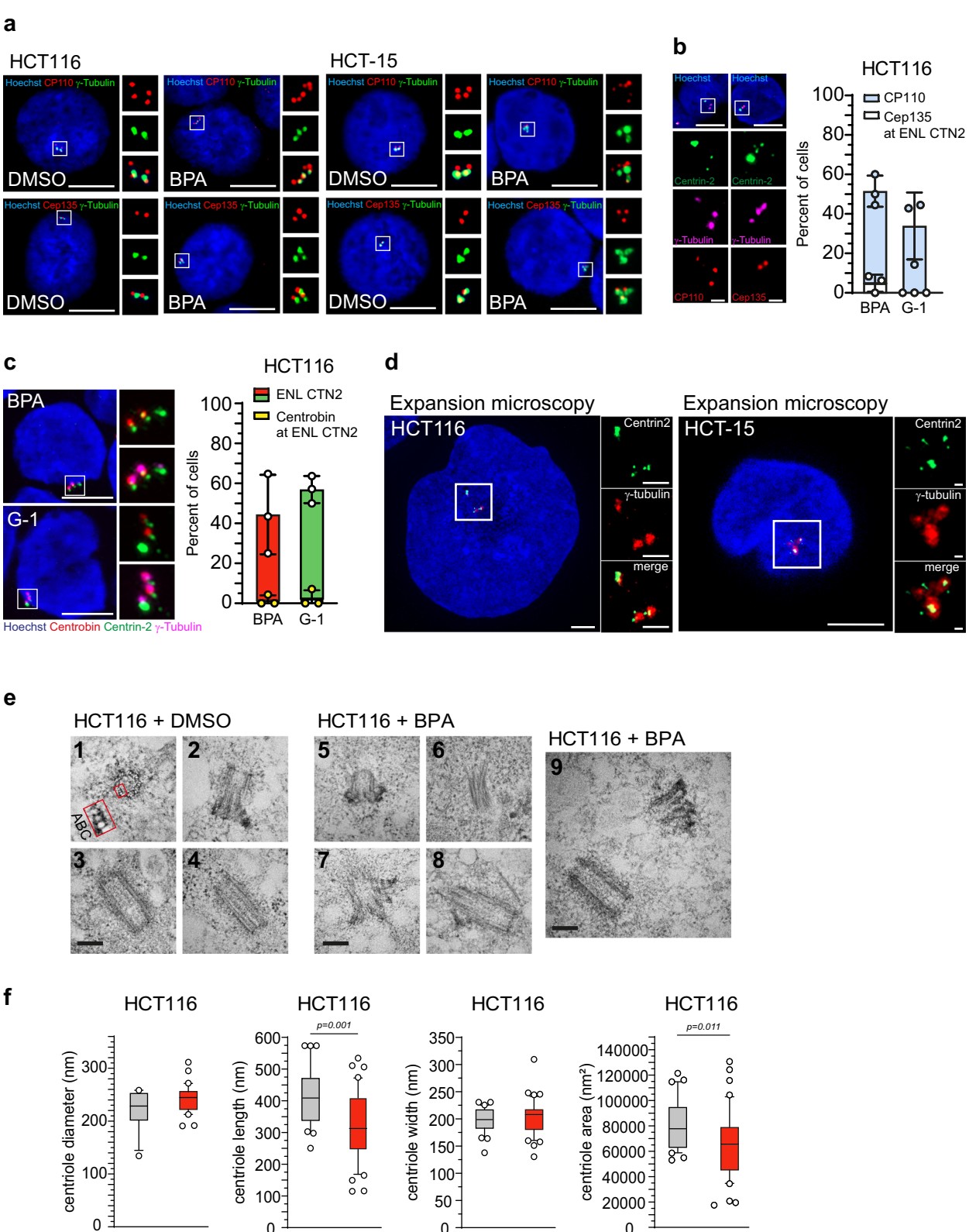

Co-immunostainings with γ-tubulin, total Centrin-2, and anti-Centrin-pS170 antibodies revealed that phosphorylated Centrin-2 was almost absent from interphase centrosomes of DMSO-treated CRC cells, as expected[11,24], whereas total Centrin co-localized with γ-tubulin (Fig. 6a, b). Remarkably, activation of GPER1 by estrogens not only resulted in supernumerary centrosomes with enlarged Centrin-2 *foci* as seen before (Fig. 3 and

Supplementary Fig. 3) but also in intensive phosphorylation of enlarged Centrin-2-labeled centrioles at serine 170 (Fig. 6a, b and Supplementary Fig. 6a).

In light of these observations, we sought to ascertain whether phosphorylation of Centrin at the PKA site is associated with activation of the GPER1-PKA axis, which we demonstrated to be implicated in the regulation

**Fig. 4 | GPER1-activation leads to aberrant centrioles with reduced length and lacking the characteristic repertoire of key centriole-associated proteins.**
**a** Representative images of interphase cells upon activation of GPER1 with xenoestrogen ligand for 48 h and co-immunostained with anti-CP110 or anti-Cep135 and anti-γ-tubulin antibodies (insets), and stained with Hoechst 33342. Scale bar: 10 μm. Note four specific CP110 *foci* characteristic of G2 cells.
**b** Representative images of interphase HCT116 cells upon activation of GPER1 with Bisphenol A for 48 h and co-immunostained with anti-CP110 or anti-Cep135, anti-Centrin-2, and anti-γ-tubulin antibodies (insets), and stained with Hoechst 33342. Scale bar: 10 μm. Scale bar of insets: 2 μm. Note the absence of Cep135 at enlarged Centrin-2 *foci*. Quantification of interphase HCT116 cells with CP110- or Cep135-positive immunostaining at enlarged Centrin-2 *foci* (Enlarged, ENL) following 48 h of GPER1 activation. **c** Representative images of interphase HCT116 cells upon activation of GPER1 with xenoestrogen or synthetic ligand for 48 h and co-immunostained with anti-Centrobin, anti-Centrin-2, and anti-γ-tubulin antibodies (insets), and stained with Hoechst 33342. Scale bar: 10 μm. Note the absence of Centrobin at enlarged Centrin-2 *foci*. Quantification of interphase HCT116 cells with enlarged Centrin-2 *foci* (Enlarged, ENL), analysis of the proportion of Centrobin-positive immunostaining at these *foci* following 48 h of GPER1 activation. **d** Representative expansion microscopy image of cells treated with 10 nM BPA with enlarged Centrin-2 *foci* at interphase centrioles. Scale bar: 10 μm. Expansion

factor of 4.5x. Scale bar of insets: 2 μm. **e** Representative transmission electron microscopy images of HCT116 cells treated with either solvent (DMSO) or Bisphenol A (BPA) to activate GPER1. Panels 1-4 show normal centrioles, and panels 5-9 depict aberrant centrioles, as described in the text. Cells shown were imaged at 60k x magnification. Scale bar: 200 nm. **f** Centriole dimensions (centriole diameter, length, width, and area) of interphase HCT116 cells treated with DMSO (solvent) or Bisphenol A (BPA) for 48 h were plotted as Box and whiskers with 10–90 percentile presentation and individual data points. **Data information:** Graphs in (**b** and **c**) show mean ± SD and individual data points from three different experiments with a total of 63 (CP110, BPA), 53 (CP110, G-1), 39 (Cep135, BPA, G-1), 56 (Centrobin, BPA), and 43 (Centrobin, G-1) amplified cells. Graphs in (**f**) show box and whiskers and individual data points from one (DMSO) or three (BPA) different experiments with a total of 37 (DMSO) and 58 (BPA) cells. The number (n) of cells evaluated for the proportion of CP110 or Cep135 positive cells at centrosome amplified cells with enlarged Centrin-2 are as followed: (**b**, CP110) $n_{(BPA)} = 33/45$; $n_{(G-1)} = 17/31$; (**b**, Cep135) $n_{(BPA)} = 2/27$; $n_{(G-1)} = 0/22$; (**c**, Centrobin) $n_{(BPA)} = 1/25$; $n_{(G-1)} = 1/24$. *P*-values < 0.05 are displayed. *P*-values for centrosome amplification are given in the Supplementary Table 1. The following statistics were applied: Adapted *bootstrap* procedure for graphs in (**a**) and Mann–Whitney's test for graphs in (**f**) as described in "Materials and Methods" section.

of centrosome numbers (Figs. 1, 2), Centrin enlargement (Fig. 3, Supplementary Fig. 3, Fig. 4d, Supplementary Fig. 4a, and Fig. 5a) and centriole size (Fig. 4e, f). To this end, we re-analyzed HCT116 and HCT-15, previously treated with different GPER1 and PKA activators (see Fig. 3d–f, Fig. 5a, and Supplementary Fig. 3c) to determine the proportion of phosphorylated Centrin at the enlarged centrioles in centrosome-amplified interphase cells (Fig. 6c–h and Supplementary Fig. 6b). Our quantifications revealed an increase for phosphorylated Centrin at enlarged centrioles to approximately 40–50% in both cell lines for all conditions with activated GPER1. HCT116 cells appeared to respond slightly more to G-1, Tamoxifen, and 8-Bromo cAMP, showing a generally higher effect level compared to HCT-15 cells, at least when using the Centrin 20H5 antibody (Fig. 6c–h and Supplementary Fig. 6b). However, this might be due to their overall higher baseline level and degree of variation. Taken together, these results show that GPER1-PKA signaling induces phosphorylation of Centrin-2 at enlarged centrioles within amplified centrosomes in interphase cells. This suggests that phosphorylation of Centrin-2 at serine 170 may contribute to the regulation of both the numerical and structural centriole integrity prior to mitosis in an in vitro colon cancer model under conditions of experimentally enhanced GPER-PKA activity.

Establishing a direct causal link between Centrin phosphorylation and Centrin enlargement remains challenging, as phospho-Centrin is detectable only at amplified centrosomes, which are largely absent upon PKA inhibition (Fig. 2c, d). To assess whether PKA activity is required to maintain already established enlarged Centrin *foci*, we applied delayed PKA inhibition during the final 5 h of G-1 treatment (Fig. 6i). In GPER1-activated cells, this led to a decrease in serine 170 phosphorylation from approximately 50% to ~10% (Supplementary Fig. 6d). However, the size of Centrin *foci* remained unchanged, indicating that their maintenance seems to be independent of sustained PKA activity (Fig. 6i).

Consistent with these findings, we re-expressed either wild-type human Centrin (WT) or a non PKA-phosphorylatable Centrin mutant (S170A) in Centrin-2 knockout cells already used before (Supplementary Fig. 5a). Protein expression and phospho-status were confirmed by Western blot and immunofluorescence microscopy, respectively (Fig. 6j and Supplementary Fig. 6c). Wild-type Centrin induced centrosome amplification upon BPA treatment, but the S170A mutant failed to do so (Fig. 6l). Notably, however, the frequency of enlarged Centrin *foci* was similar in both wild-type and mutant-expressing cells (Fig. 6k), indicating that phosphorylation at S170 seems to be required for numerical centrosome integrity, but is not required for Centrin enlargement per se. Interestingly, Centrin displacement, which is commonly observed in GPER1-activated cells (Fig. 3c), was markedly reduced in cells expressing the non-phosphorylatable Centrin

mutant (Supplementary Fig. 6e). This is consistent with previous work by Lutz et al. [24], showing that PKA activation promotes centriole separation in interphase HeLa cells *via* centrosomal phosphorylation[24].

In summary, our data support a model in which PKA-mediated phosphorylation of Centrin-2 is critical for centrosome amplification and for initiating, but not maintaining, structural changes at the centrosome. This points toward a two-step mechanism in which PKA activity serves as a priming signal for centrosomal remodeling, followed by stabilization through additional, PKA-independent pathways.

**Phosphorylated Centrin-2 associates with the catalytic subunit of PKA during interphase**

We hypothesize that Centrin-2 and PKA physically interact at centrosomes, leading to the phosphorylation of Centrin-2 at amplified centrosomes upon stimulation of GPER1 and PKA (Fig. 6). To test this, we first investigated whether GPER1 activation affects the total protein content of PKA. Western blot studies of whole cell lysates revealed that GPER1 had no apparent effect on PKA protein levels, as PKA-Cα (see also Supplementary Fig. 2c) and PKA-RIIα levels in estrogen-exposed HCT116 and HCT-15 cells were comparable to the DMSO control (Fig. 7a). Although the total Centrin-2 protein level appeared to increase slightly following E2, DES and DME exposure, quantification of three independent replicates showed that these differences were not statistically significant (Fig. S7a). Again, phosphorylation of Centrin-2 at the PKA site was detectable in mitotically arrested cells but not in interphase cells exposed to DMSO or estrogens[11,24] (Fig. 5d and Fig. 7a).

We next investigated the potential kinase-substrate interaction between PKA and Centrin-2 during interphase, and assessed whether this interaction is dependent on GPER1. First, we verified that PKA catalytic subunits assemble with regulatory subunits into a holoenzyme complex, as evidenced by PKA-Cα bound to PKA-RIIα in mitotically arrested HCT116 cells (Fig. 7b). Of note, Hsp90, a well-known binding partner of PKA[72], co-immunoprecipitated with PKA-RIIα, confirming the functionality and specificity of the immunoprecipitation protocol. In contrast, Centrin-pS170, as expected[24], did not associate with PKA-RIIα under these conditions. Importantly, PKA-Cα and PKA-RIIα similarly physically interact during interphase as confirmed by PKA-RIIα immunoprecipitation from asynchronously growing HCT116 cells exposed to DMSO (Fig. 7c). However, it was found that the amount of PKA-Cα co-immunoprecipitated with PKA-RIIα was reduced to 33% after BPA exposure (Fig. 7c). This fits with the current hypothesis that stimulation by GPER1 causes dissociation of the PKA holoenzyme into its catalytic and regulatory subunits, which is an important prerequisite for PKA activation[73]. As demonstrated in mitotic

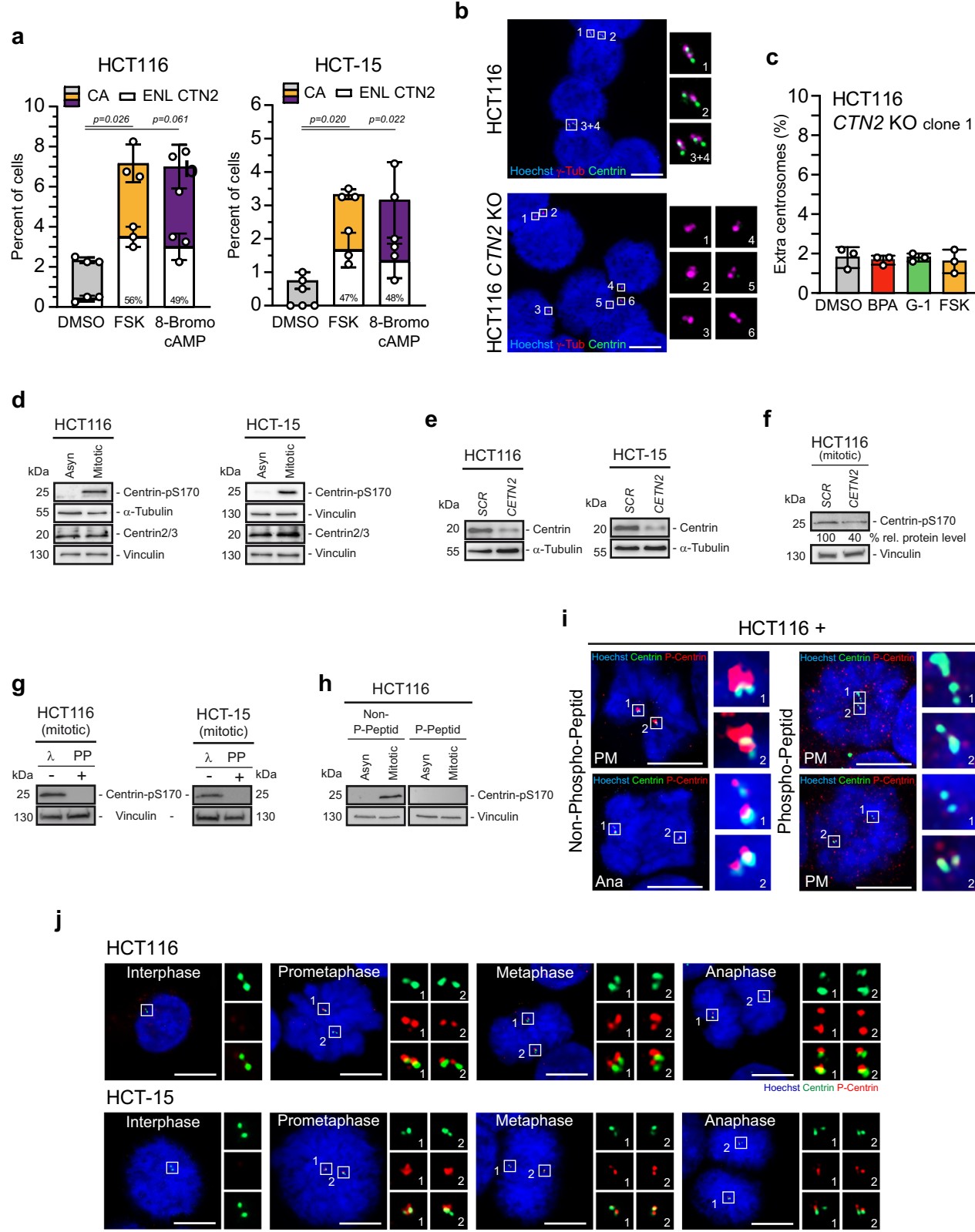

cells (Fig. 7b), Centrin-pS170 also did not physically interact with PKA RIIα during interphase, neither in the absence (DMSO exposure) nor in the presence (BPA exposure) of activated GPER1 (Fig. 7c). However, it appeared that phosphorylated Centrin-2 interacts with PKA-Cα during mitosis (Fig. 7d), although detection of the specific signal was initially hampered by overlap with the immunoglobulin light chain of the Centrin-

pS170 antibody used for Western blotting (see also Supplementary Fig. 7b). To resolve this issue, we adapted the SDS-PAGE conditions by expanding gel run time, which allowed clear separation of the light chain from the phospho-Centrin signal (Supplementary Fig. 7c). This enabled us to examine, whether phosphorylated Centrin-2 binds to PKA catalytic sub-units in non-mitotic cells following GPER1 activation. Indeed, we were able

**Fig. 5 | Activated PKA causes enlarged Centrin signals at amplified centrosomes and phosphorylation of Centrin at serine-170. a** Quantification of interphase HCT116 and HCT-15 cells with more than two centrosomes (centrosome amplification, CA) and the proportion of centrosome amplified cells with enlarged Centrin-2 foci (Enlarged, ENL) upon activation of PKA for 48 h. Percentages of enlarged Centrin-2 foci are given. **b** Representative images of interphase HCT116 parental and Centrin-2 (CTN2) knockout cells co-immunostained with anti-Centrin-2 and anti-γ-tubulin antibodies (insets), and stained with Hoechst 33342. Scale bar: 10 μm. Note the absence of Centrin-2 signals in CTN2-deleted cells. **c** Quantification of HCT116 interphase cells (clone 1) depleted of Centrin-2 (CTN2 knockout, KO) with more than two centrosomes upon activation of GPER1 and PKA for 48 h. Whole-cell lysates were immunoblotted from cells treated with DMSO (Asyn) or 2 μM dimethylenastron (Mitotic) for 16 h (**d**) and (**f**), or were left untreated (**e**) and, where indicated, transfected with SCRAMBLED (SCR) or Centrin-2 specific siRNA (CETN2). α-tubulin and Vinculin served as loading

controls. Relative protein levels were normalized to the loading controls and expressed relative to SCR. **g** Whole-cell lysates exposed or not exposed to λ-phosphatase were immunoblotted from cells treated with 2 μM dimethylenastron (mitotic). **h** Whole-cell lysates were immunoblotted from cells exposed to Centrin-2 phospho- and non-phosphopeptide for 2 h. Vinculin served as a loading control. **i, j** Cells were co-immunostained with anti-Centrin-2 and anti-Centrin-pS170 antibodies (insets), and stained with Hoechst 33342. **i** Representative images of cells in prometa- (PM) or anaphase (Ana) of mitosis treated as in (**h**). Scale bar: 10 μm. **j** Representative images of cells in different cell cycle phases as indicated. Scale bar: 10 μm. **Data information:** Graphs in (**a**) and (**c**) show mean ± SD and individual data points from three different experiments with a total of 1200 (**a**) and 1500 (**c**) amplified cells. P-values < 0.05 are displayed. P-values for centrosome amplification are given in the Supplementary Table 1. The following statistics were applied: adapted bootstrap procedure for graphs in (**a**) and bootstrap procedure for graphs in (**c**) and as described in "Materials and Methods" section.

to co-immunoprecipitate Centrin-2-pS170 with PKA-Cα in asynchronously growing HCT116 and HCT-15 cells treated with DMSO (Fig. 7e, f). Intriguingly, phosphorylated Centrin-2 physically complexed even more with PKA-Cα in the presence of GPER1 activating estrogens such as E2, BPA, and DES. Normalization of phospho-Centrin to the PKA-Cα signal showed that BPA and DES induce a similar fold increase of 1.8 and 1.6, respectively in HCT116 cells, while E2 induces complex formation slightly less strongly with a 1.4-fold increase. We verified these results using HCT-15 cells and showed that PKA-Cα was again efficiently immunoprecipitated for all conditions as was seen for HCT116, with an 1.2 to 1.6-fold increase upon treatment with BPA, DES or E2 compared to the DMSO control (Fig. 7f). To complement our immunoprecipitation experiments, which indicate protein binding, we additionally employed a sensitive proximity ligation assay (PLA) to directly visualize and localize the interaction between Centrin and PKA-Cα in CRC cells. This in situ approach uses antibody probes conjugated to DNA oligonucleotides that, upon close proximity (40-60 nm), undergo enzymatic ligation and rolling circle amplification[74]. The resulting fluorescent signals indicate the location and abundance of protein interactions at single molecule resolution. We complemented the PLA method with subsequent immunofluorescence staining against γ-tubulin to visualize the pericentriolar material, enabling us to specifically detect PKA-Centrin interactions at interphase centrosomes. The combined PLA and immunofluorescence approach confirmed the centrosomal interaction between Centrin and PKA-Cα in both DMSO- and BPA-treated HCT116 cells (Fig. 7g). Notably, displaced centrosomes lack PKA-Centrin interaction, which is consistent with previous immunofluorescence analyses showing that displaced interphase centrosomes lack detectable PKA signals following GPER1/PKA activation (Supplementary Fig. 2d).

To investigate whether the physical interaction of PKA-Cα with Centrin-pS170 was GPER1-specific, we repeated the immunoprecipitation studies and included the GPER1-specific agonist G-1 in our studies (Fig. 7h, i). We were able to verify our results obtained by exposure to estrogens and again observed more phosphorylated Centrin-2 bound to PKA-Cα in the presence of G-1-stimulated GPER1.

Taken together, these results support the proposed kinase-substrate interaction between PKA and Centrin-2 at interphase centrosomes in the presence of hormonally or agonist-stimulated GPER1.

## Discussion

This study makes an exciting contribution to our understanding of how centrosome integrity is regulated in a colon cancer-derived in vitro model. Our results demonstrate the existence of a previously uncharacterized signaling cascade initiated by GPER1 activation by endogenous, synthetic, and xenoestrogen ligands. This cascade ultimately leads to the abnormal phosphorylation of Centrin-2 during interphase, resulting in the disruption of both centrosome number and centriole integrity (Fig. 8). In particular, we showed that GPER1 triggers a Gαs-mediated and adenylyl cyclase-dependent pathway that leads to AKAP450-mediated recruitment of PKA

to centrosomes, where the kinase physically interacts with and atypically phosphorylates centriolar Centrin-2 during interphase of the cell cycle. Excessive activation of this GPER1-PKA-Centrin-2 axis outside of mitosis leads to amplified centrosomes and the formation of morphologically aberrant and shortened centrioles, which contain enlarged Centrin-labeled foci that are spatially separated from their normal-sized counterparts. Our findings shed new light on the crucial role of GPER1-mediated PKA activation in CRC cells, revealing how this process drives numerical and structural centrosome defects.

GPER1 signals through multiple G proteins, initiating several downstream cascades, including cAMP production with PKA activation, activation of Src and downstream EGFR signaling, or mobilization of calcium (Ca²⁺), which in turn activates protein kinase C (PKC)[48]. The former is mediated by the binding of GPER1 to G protein αs subunit, which leads to stimulation of adenylyl cyclase and cAMP-dependent signaling, which contributes to activation of PKA[41,48]. Our data now reveal that Centrin-2, a component of the centriole, is a target of GPER1-adenylyl cyclase-transactivated PKA in a previously unconsidered in vitro CRC model system. This target is affected in the size, motility and phosphorylation status in centrosome-amplified interphase cells. These new findings address the question that remained unanswered from our previous study, in which we demonstrated that the activation of GPER1 by estrogens or receptor-specific agonists results in centrosome amplification in a non-classical hormone-derived CRC cell system[38]. The connection between the G protein-coupled, membrane-bound estrogen receptor (i.e., GPER1) and the centrosome, has been largely unclear, at least in the CRC model. In this study, we address this gap by demonstrating that activated GPER1 promotes the AKAP450-dependent recruitment of PKA to the centrosome, thereby, modulating centriole size, composition, and structure of centriole-associated proteins, and increasing centrosome number.

The centriole defect is characterized by structural abnormalities, including a general reduction in length, incomplete incorporation of canonical proteins, and the appearance of enlarged Centrin-positive foci (Figs. 3 and 4), although their molecular composition remains elusive. Our expansion microscopy analyses provide initial evidence that multiple Centrin-2 molecules may self-assemble into a "super spot" structure (Fig. 4d and Supplementary Fig. 4a). This hypothesis finds support in the fact that Centrins are Ca²⁺-binding proteins capable of forming multimers upon calcium binding in yeast, algae, and humans[75]. The biological significance of Centrin oligomerization is not clear, but it may be important for Centrins to perform their specific cellular function[11]. Notably, when GPER1 is bound to the Gαq subunit, it activates phospholipase C (PLC), leading to an elevation of cytosolic Ca²⁺ levels, which could, in turn, trigger Centrin self-assembly[76]. Given that not all but half of the centrosome-amplified cells studied exhibit Centrin-enlargement upon GPER1 activation (Fig. 3), it is plausible to speculate that additional to GPER1/Gαs-dependent mechanisms, G protein αq-triggered Ca²⁺ release (or even GPER-independent mechanisms) may (synergistically) influence the size of Centrin-labeled centrioles. Notably, the

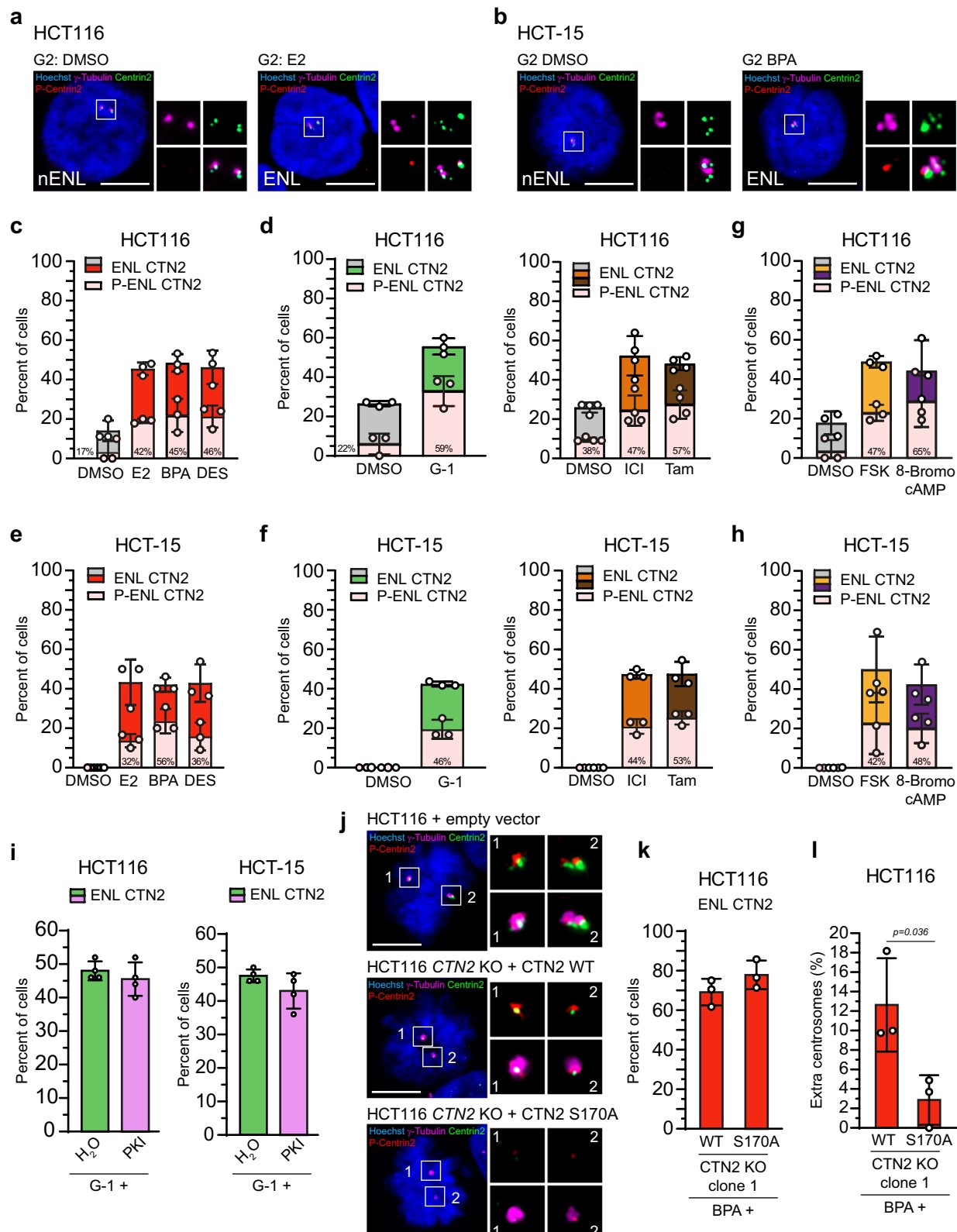

appearance of enlarged Centrin *foci* seems to be specific to Centrin, as the size of other centriole markers such as CP110 and Cep135 remains unchanged upon GPER1 activation (Fig. 4a). It is striking, however, that these extra Centrin *foci* partially colocalize with CP110 but show little overlap with Cep135 (Fig. 4b), suggesting possible displacement or incomplete recruitment of specific centriole components in the presence of

enlarged Centrin *foci*. Together, these findings suggest that GPER1 activation induces centriole remodeling, supporting the idea that centriole-like structures may be immature, misassembled, or aberrant rather than fully functional centrioles. Whether these abnormal structures serve as a functional scaffold, or contribute to whole chromosomal instability observed following GPER1 activation[38] remains to be further investigated.

**Fig. 6 | Enlarged Centrin signals are atypically phosphorylated at amplified centrosomes during interphase.** Representative images of HCT116 (**a**) and HCT-15 cells (**b**) upon activation of GPER1 with endogenous or xenoestrogen ligands for 48 h, with or without enlarged Centrin-2 *foci* and phosphorylated Centrin-2 at Ser170 (Insets) at G1 or G2 phase. Scale bar, 10 µm. **c–f** Interphase cells from Fig. 3d–f upon activation of GPER1 with endogenous, synthetic, xenoestrogen or anti-estrogen ligands for 48 h. The bar graphs show quantification of centrosome amplified interphase cells with enlarged Centrin-2 *foci* (Enlarged, ENL) and the proportion of enlarged Centrin-2 phosphorylated at Ser170 (P-ENL). Percentages of enlarged and concomitantly phosphorylated Centrin-2 *foci* are given. The number (n) of cells evaluated for the proportion of phosphorylated Centrin-2 from centrosome amplified cells with enlarged Centrin-2 are as followed: (**c**) $n_{(DMSO)} = 1/4$; $n_{(E2)} = 16/38$; $n_{(BPA)} = 18/39$; $n_{(DES)} = 18/37$; (**d**) $n_{(DMSO)} = 2/9$; $n_{(G-1)} = 29/49$; $n_{(DMSO)} = 4/11$; $n_{(ICI)} = 28/60$; $n_{(Tam)} = 30/52$; (**e**) $n_{(DMSO)} = 0/0$; $n_{(E2)} = 5/16$; $n_{(BPA)} = 10/18$; $n_{(DES)} = 6/16$; (**f**) $n_{(DMSO)} = 0/0$; $n_{(G-1)} = 8/17$; $n_{(DMSO)} = 0/0$; $n_{(ICI)} = 8/18$; $n_{(Tam)} = 9/17$. **g, h** Interphase cells from Fig. 5a upon activation of PKA. The number (n) of cells evaluated for the proportion of phosphorylated Centrin-2 from centrosome amplified cells with enlarged Centrin-2 are as followed: (**g**) $n_{(DMSO)} = 1/5$; $n_{(FSK)} = 20/42$; $n_{(8-Bromo\ cAMP)} = 23/36$; (**h**) $n_{(DMSO)} = 0/0$; $n_{(FSK)} = 9/20$; $n_{(8-Bromo\ cAMP)} = 7/16$. **i** HCT116 and HCT-15 cells were treated with a synthetic GPER1 ligand for 43 h to induce GPER1-mediated centrosome amplification, followed by concurrent inhibition of PKA using PKI (14–22) amide (PKI) during the final 5 h. The bar graphs show quantification of interphase cells with enlarged

Centrin-2 *foci* (Enlarged, ENL) upon 48 h of treatment. **j** Representative images of HCT116 parental cells expressing an empty vector (control), and HCT116 cells depleted of Centrin-2 (*CTN2* knockout, KO) re-expressing either wild-type Centrin (*CTN2* WT) or a PKA-non-phosphorylatable Centrin mutant (*CTN2* S170A). Cells were co-immunostained with anti-Centrin-2, anti-Centrin-pS170, and anti-γ-tubulin antibodies (insets), and stained with Hoechst 33342. Scale bar, 10 µm. **k, l** HCT116 cells depleted of Centrin-2 (*CTN2* knockout, KO) and re-expressing either wild-type Centrin (WT) or a PKA-non- phosphorylatable Centrin mutant (S170A) were treated with Bisphenol A (BPA) for 48 h to induce GPER1-mediated centrosome amplification. The bar graphs show quantification of interphase cells with (**k**) enlarged Centrin-2 *foci* (Enlarged, ENL), and (**l**) quantification of interphase cells with more than two centrosomes, corrected for background amplification observed in the respective DMSO-treated controls. **Data information:** All graphs in (**c–i**), (**k**), and (**l**) show mean ± SD and individual data points from three (**c**), (**d**, left), (**g**), (**e–h**), (**k**), and (**l**) or four (**d**, right), and (**i**) different experiments with a total of 1200 (**c**), (**d**, left), (**g**), (**e–h**), 1600 (**d**, right) or 200 (**i**) amplified cells. The number (n) of cells evaluated for the proportion of centrosome amplified cells with enlarged Centrin-2 in (**k**) are as followed: $n_{(WT)} = 66/96$; $n_{(S170A)} = 75/97$. The number (n) of cells evaluated for the proportion of centrosome amplified cells with enlarged Centrin-2 in (**l**) are as followed: $n_{(WT)} = 169/524$; $n_{(S170A)} = 110/501$. *P*-values for centrosome amplification are given in the Supplementary Table 1. *P*-values for enlarged Centrin-2 are given in the related Fig. 3f–i and Fig. 4a. *P*-values for phosphorylated, enlarged Centrin-2 cannot be provided due to sample sizes.

Interestingly, Marteil et al. [5] previously reported that excessive centriole elongation can promote centrosome amplification, specifically through centriole fragmentation and ectopic procentriole formation[5]. These findings are particularly relevant to our observations, as the enlargement of Centrin-labeled *foci* may contribute to centrosome amplification in a similar manner. Notably, cells with overly long centrioles typically harbor only one elongated centriole[5]. Consistent with this, we observed that centrosome-amplified cells exhibited only one enlarged Centrin *foci* per centrosome (Fig. 3a, b, and Supplementary Fig. 3b). It is conceivable that centrioles with enlarged Centrin *foci* may also undergo fragmentation, thereby contributing to centrosome amplification. However, a larger proportion of GPER1-activated cells in our study displayed abnormally short centrioles rather than enlarged Centrin *foci*. This suggests that centrosome amplification can also occur in the presence of short centrioles, potentially through alternative or additional mechanisms, such as centriole fragmentation. Supporting a role for Centrin in centrosome amplification, gene-specific knockout restored normal centrosome numbers in the presence of activated GPER1 or PKA (Fig. 5c and Supplementary Fig. 5b). Moreover, phosphorylation of Centrin at the PKA target site appears relevant for centrosome amplification, although it remains to be clarified whether this modification directly contributes to the formation of enlarged Centrin *foci* (Fig. 6k, l). Notably, our data show that phosphorylation of Centrin at serine 170 by PKA during interphase is associated with distinct centriole separation (Supplementary Fig. 6e). These findings are in line with a study by Lutz et al. [24], who demonstrated that Centrin-2/3 is phosphorylated by PKA at serine residue 170 during mitosis and, under conditions of experimentally elevated PKA activity, also at interphase centrosomes in HeLa cells. This phosphorylation has been shown to trigger centriole separation prior to centrosome duplication[24].

It is of particular significance that our findings present a comprehensive insight into the PKA-Centrin signaling axis in a CRC-derived in vitro system. This involves GPER1-mediated activation of adenylyl cyclase, resulting in the production of cAMP and, subsequently, augmented PKA activation at interphase centrosomes. By using a different cell system than previous studies[24] (i.e., colon cancer-derived cells), our findings reinforce the established link between PKA and Centrin-2 to a broader, more universally valid level, independent of cell lines derived from specific tissues. Of note, similar to the enlargement phenotype, not all amplified centrosomes have phosphorylated Centrin-labeled *foci* after GPER1 activation. About 50% of enlarged *foci* in centrosome-amplified cells are phosphorylated at serine 170. This suggests that Centrin enlargement and phosphorylation may also be mediated by other protein kinases sharing substrate specificity with PKA,

in response to GPER1 activation. One such kinase could be PKC, particularly isoforms ε and δ, known to localize at the centrosome[77,78], likely in close proximity to Centrin proteins capable of phosphorylating them[24]. Notably, Centrin-2 physically binds to and interacts with both PKA (Fig. 7e–i) and Aurora A[23], and is phosphorylated at the same site by both kinases to regulate Centrin stability, centrosome amplification mediated by Aurora A, and centriole separation[23,24]. An intriguing question for future consideration is whether Aurora A contributes to the numerical centrosome defect observed in our study following GPER1 activation.

Supernumerary centrosomes induce w-CIN and aneuploidy, contributing to invasiveness through increased microtubule nucleation and elevated RAC1 activity[38,79,80]. Enlarged Centrin-labeled structures may also trigger invasiveness, as they may induce the formation of over-active centrosomes, with enhanced microtubule nucleation capacities triggering chromosome missegregation[5]. It remains to be investigated whether the GPER-PKA-Centrin-2 axis also contributes to these defects. Lingle et al. [25] and Lukasiewicz et al. [23] showed that Centrins are highly phosphorylated in human breast tumors that have extra centrosomes[23,25]. Further investigation is needed to elucidate the precise mechanisms by which GPER1 signaling regulates centrosome dynamics and to assess its relevance in vivo.

In conclusion, the findings of this study offer compelling support for a signaling pathway, involving GPER1-PKA-Centrin, which plays a role in regulating centrosome/centriole integrity in CRC cells (Fig. 8). By elucidating the molecular basis of centrosome defects, our study enhances the mechanistic understanding of colorectal carcinogenesis. Furthermore, our findings provide crucial mechanistic insight into the potential effects of industrial synthetic chemicals (such as bisphenol A), which are suspected to have estrogenic effects and to be endocrine disruptors. A significant concern in health risk assessment is that these chemicals have so far been studied only for their potential effects *via* the estrogen receptor alpha and beta, overlooking the toxicological relevance of GPER1-mediated effects. Our findings highlight the importance of GPER1-mediated effects of industrial synthetic chemicals, emphasizing their relevance to health protection at the mechanistic level. The development of new in vitro test methods based on GPER1-PKA-mediated signaling pathways is recommended to enhance our understanding of industrial chemicals and improve the prediction of their potential carcinogenic effects.

## Materials and methods
### Cell lines and growth conditions
HCT15 (male) were purchased from the American Type Culture Collection (ATCC). HCT116 (male) were purchased from the German Collection of

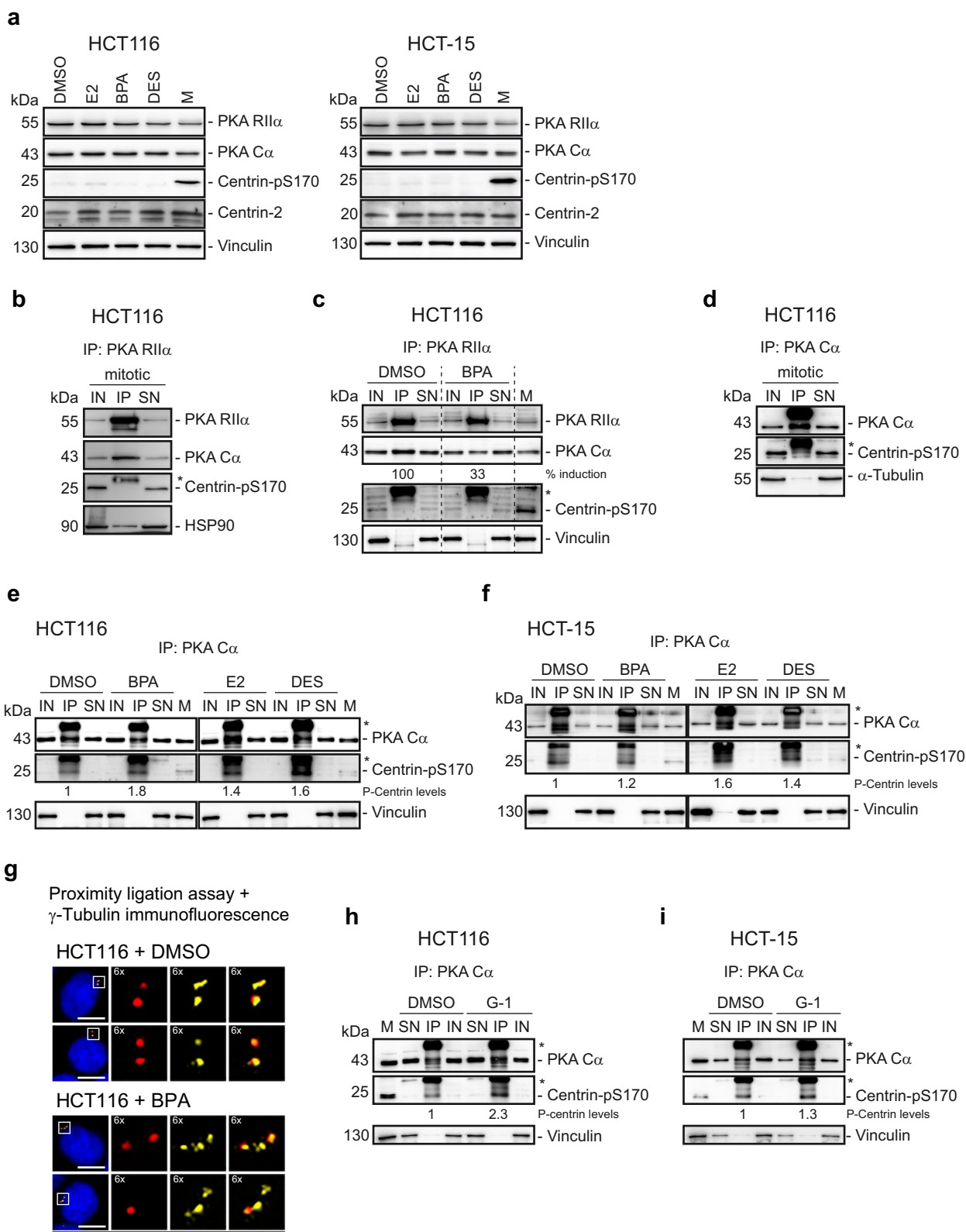

Microorganisms and Cell Cultures GmbH (DSMZ). The cell lines were routinely checked for mycoplasma contamination tested using the Eurofins Genomics mycoplasma test service (Eurofins Genomics, Ebersberg, Germany). All cell lines used were tested negative. CRC cell lines were cultured in phenol red-free RPMI 1640 (PanBiotech), supplemented with 10% charcoal stripped fetal calf serum (Th. Geyer GmbH) and 1% penicillin/

streptomycin (Merck Millipore) at 37 °C and 95% humidity and 5% $CO_2$. Culture medium conditions were used for all experiments.

**Reagents and cell treatments**

All chemicals were purchased from Sigma unless otherwise stated, dissolved in dimethyl sulfoxide (DMSO) or water as indicated and stored at −20 °C.

**Fig. 7 | Phosphorylated Centrin-2 associates with the catalytic subunit of PKA during interphase. a** Whole-cell lysates were immunoblotted from cells treated with endogenous or xenoestrogen ligands. 2 μM dimethylenastron for 16 h (M) served as a control for phospho-Centrin. Vinculin served as a loading control. **b** Cells were treated with 2 μM dimethylenastron for 16 h (mitotic). Cell lysates were immuno-precipitated with anti-PKA RIIα antibody. HSP90 served as a control for PKA-Cα-binding[72]. **c** Cells were treated with xenoestrogen GPER1 ligand for 48 h. Cell lysates were immunoprecipitated with anti-PKA RIIα antibody. Band intensities of immunoprecipitated PKA Cα were normalized to immunoprecipitated PKA RIIα and expressed relative to DMSO. Vinculin served as a loading control. **d** Cells were treated with 2 μM dimethylenastron for 16 h (mitotic). Cell lysates were immuno-precipitated with anti-PKA Cα antibody. α-tubulin served as a loading control. **e, f** Cells were treated with endogenous or xenoestrogen GPER1 ligands for 48 h. Cell lysates were immunoprecipitated with anti-PKA Cα antibody. Band intensities of immunoprecipitated phospho-Centrin (Centrin-pS170) were normalized to immunoprecipitated PKA Cα, which was in turn normalized to Vinculin from the IP supernatant control (SN). Values are expressed relative to the DMSO control. Vinculin served as a loading control. Note: All samples were derived from the same

experiment. Independent gels/membranes (DMSO/BPA and E2/DES) were processed in parallel under identical conditions (i.e., blotting and development in the same device, using the same washing buffers, blocking and antibody solutions and incubations, and exposure times) to ensure signal comparability across membranes. **g** Representative images of HCT116 cells treated with DMSO (solvent control) or Bisphenol A (BPA) for 48 h to induce GPER1 activity and subjected to proximity ligation assay. Red puncta represent interaction between Centrin and PKA-Cα. Cells were co-immunostained with anti-γ-tubulin antibodies to confirm Centrin/PKA-interaction at interphase centrosomes, and stained with Hoechst 33342. Scale bar, 100 μm. **h, i** Cells were treated with synthetic GPER1 ligand for 48 h. Cell lysates were immuno-precipitated with anti-PKA Cα antibody. Band intensities of immuno-precipitated Centrin-pS170 were normalized to immunoprecipitated PKA Cα, which in turn was normalized to Vinculin from the supernatant control. Values are expressed relative to the DMSO control. Vinculin served as a loading control. **Data information:** IP immunoprecipitates, IN input control (50 μg of lysate), SN = IP supernatant control (50 μg equivalent); M mitotic cells. Asterisks (*) mark light/heavy chains of the IP capture antibody.

**Fig. 8 | Proposed model of a GPER1-PKA-Centrin signaling axis regulating centriole integrity in colon cancer cells.** GPER1 activation triggers Gαs-dependent stimulation of adenylyl cyclase, which converts ATP into cAMP. The tetrameric PKA holoenzyme, composed of regulatory and catalytic subunit dimers, remains sequestered in an inactive state in the absence of cAMP. Upon binding of cAMP to PKA regulatory subunits, the holoenzyme becomes active and is recruited to the interphase centrosome by AKAP450. At the centrosome, PKA dissociates, releasing its catalytic subunits, which are then free to phosphorylate Centrin at the centrioles. In addition to atypical phosphorylation of Centrin outside of mitosis, the centriolar protein becomes enlarged compared to its normal-sized counterparts in centrosome-amplified cells. Moreover, amplified centrioles display structural abnormalities and reduced length, as described in the main text. C, PKA-Cα; CTN centrin PCM, pericentriolar material, R, PKA-Rα.

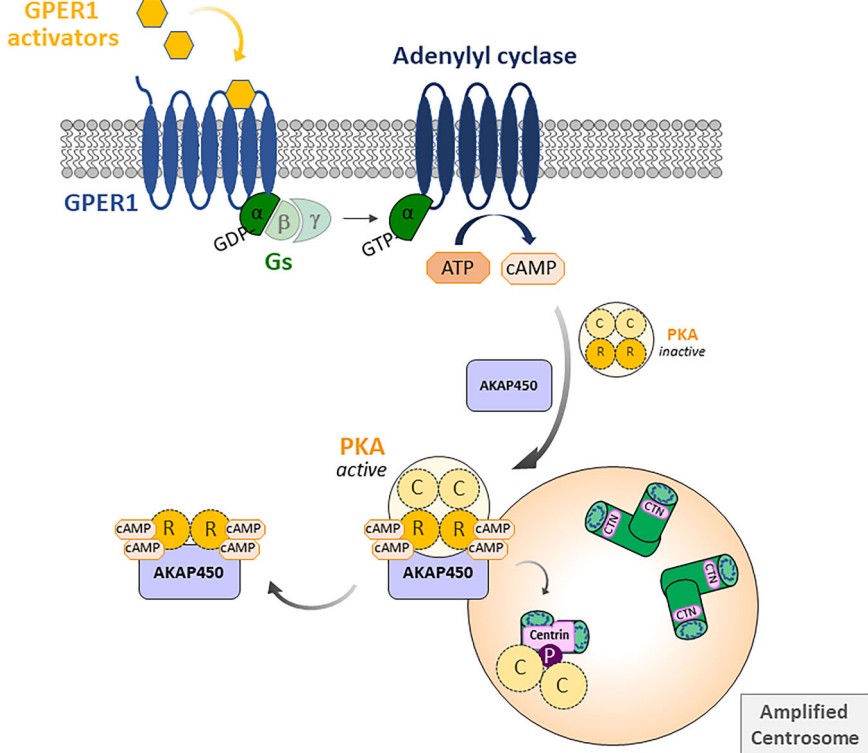

The cells were treated with 10 nM 17β-estradiol (E2), 10 nM bisphenol A (BPA), 10 nM diethylstilbestrol (DES), 100 nM G-1 (Tocris), 100 nM ICI182,780 (ICI), 10 nM tamoxifen (Tam), 10 nM NF449, 800 ng/ml Pertussis toxin (PXT; Enzo, storage 4 °C, in H₂O), 10 μM Forskolin (FSK), 10 μM SQ 22,536, 100 μM (HCT116) or 10 μM (HCT-15) 8-Bromo-cAMP, and 5 μM PKI (14–22) amide (myristoylated, in H₂O), for 48 h in stripped medium. To arrest cells in mitosis, cells were treated with 2 μM Dimethy-lenastrone (Sigma Aldrich) for 16 h. For protein dephosphorylation, mitotically arrested cells or lysates were treated with λ-phosphatase (New England Biolabs, Ipswich, United Kingdom) according to the manufactures' instructions. Corresponding volumes of DMSO or H₂O were used as controls.

### Transfections and generation of stable cell lines
Cells were transfected with siRNAs using INTERFERin® siRNA transfection reagent (Polyplus Transfection) according to the manufacturer's

instructions 60 pmol siRNA; (Eurofins genomics, Germany) according to the manufacturer protocols. The used siRNA sequences (Eurofins, custom made) are listed in the Supplementary Table 2. Cells were analyzed 48 h after transfection and Western blotting was used to confirm transfection efficiency. pcDNA 3.1 and pCMVflag-Plk4 plasmids were kindly provided by Ingrid Hoffmann (DKFZ, Heidelberg, Germany) and used for transient DNA transfection using Torpedo DNA Transfection Reagent (Ibidi) according to the manufacturer's protocol. To generate HCT116 cells depleted of Centrin-2 or AKAP450, stable transfections of CRISPR-Cas9 plasmids (sc-400867-NIC-2, sc-405430-NIC-2, Santa Cruz) were performed using Metafectene (Biontex) according to the manufacturer's protocol using 2 μg of plasmids. 29 h post transfection, transfection efficiency was confirmed by monitoring GFP signals via fluorescence microscopy analysis. 48 h post transfection, 1 μg/ml (HCT116) puromycin was added and cells were selected for two days. At four days post puromycin-selection, transfected and selected cells were diluted (1:1000-1:10,000), and single-cell

clones were isolated and expanded in normal growth medium. Centrin-2 and AKAP450 knockout of selected cell clones was verified using Western blot analysis. To re-express either wild-type Centrin (*CTN2* WT) or a PKA-non-phosphorylatable Centrin mutant (*CTN2* S170A) in Centrin-depleted HCT116 cells, cells were transiently transfected with 10 µg plasmids by electroporation using an electroporator (BioRad) at 300 V and 500 mF. The Centrin wild-type (NM_004344.3) and S170A vectors (NM_004344.3, codons 508-510 were changed to gcc) were obtained from, VectorBuilder. Codons 331-351 were changed to aagctgttcgacgacgatgaa to receive a Centrin-2 siRNA resistant version of both vectors.

## Immunoprecipitation
To investigate protein-protein interactions, and co-immunoprecipitation of different proteins with PKA, cells were treated with GPER1 activators for 48 h and subsequently harvested with PBS/0.5 mM EDTA. Mitotically arrested cells served as a control for phosphorylated Centrin. The cell pellets were collected by centrifugation at 130 x *g* for 5 min at RT. Cells were lysed for 30 min in ice-cold IP lysis buffer (50 mM tris/HCl pH 7.4, 150 mM NaCl, 10% (v/v) Glycerin. 0.1 mM EDTA, 0,25% (v/v) NP-40 Igepal, supplemented with Protease Complete Inhibitor and Phosphatase Complete Inhibitor cocktails (Roche), 0.5 µM Microcystin-LR, and 1 mM DTT). Lysates were additionally sonicated (Hielscher Ultrasonics GmbH) 3 times for 0.5 ms at 80% amplitude. Lysates were centrifuged and protein concentration was determined as described for Western blot procedure. An aliquot of the lysate (Input, 50 µg) served as a control. Two milligrams of the lysate (adjusted to a concentration of 7 mg/mL) were incubated with 0.6 µg anti-PKA Cα (4782, Cell Signaling) or 2.0 µg anti-PKA-RIIα (sc-137220) on a rotating wheel for 2 h at 4 °C (IP sample), and immunocomplexes were subsequently precipitated using protein-G Sepharose beads (Cytiva). An aliquot of the supernatant (SN, 50 µg) served as a control for IP efficiency. The immunocomplexes were washed five times with lysis buffer, and finally, eluted with 15 µL of 5 x SDS loading buffer.

## Protein extract preparation and western blotting
To prepare whole-cell extracts, cells were washed once with PBS, detached from the plate with 0.5 mM EDTA (Sigma) in PBS and the cell pellets were collected by centrifugation at 130 x *g* for 5 min at RT. Cells were lysed for 20 min in ice-cold lysis buffer (50 mM Tris·HCl [pH 7.4], 150 mM NaCl, 5 mM EDTA, 5 mM EGTA, 1% (v/v) Nonidet P-40, 0.1% (w/v) SDS, 0.1% sodium desoxycholate) supplemented with protease inhibitor, and phosphatase inhibitor mixtures (Millipore Sigma). Lysates were centrifuged at 20,000 x *g* at 4 °C for 20 min. The supernatant was transferred into a new tube, and protein concentration was measured using the DC Protein Assay Reagents Package (Bio-Rad) according to manufacturer's instruction. The concentration was determined photometrically at 750 nm using TECAN Infinite M200 plate reader. 50-80 µg of protein was boiled with 5x sample buffer (250 mM Tris-HCl [ph 6.8], 50% (v/v) Glycerol, 15% (v/v) SDS, 25% (v/v) ß-Mercaptoethanol, 0,25% (w/v) Bromphenolblue) and loaded on an 12% (IP samples), 10% or 5–7.5% SDS polyacrylamide gel with Tris running buffer. Whole-cell extracts were transferred to a PVDF membrane (GE Healthcare) using tank-blot procedure. IP samples were blotted onto nitrocellulose membranes (Trans-Blot Turbo Mini 0.2 µm Nitrocellulose Transfer Packs) using Fast-Blot procedure (BioRad). Membranes were washed with 0.1% Tween in Tris-buffered saline (TBST), blocked with 5% nonfat powdered milk in 1% TBST or Intercept blocking buffer (Licor), and then incubated with primary antibodies overnight at 4 °C. Proteins were detected using the SuperSignalTM West Pico PLUS Chemiluminescent Substrate Kit (Thermo Scientific) according to manufacturer instructions, and documented with the Fusion Solo S. Images were further processed and Western blot bands were quantified using Adobe Photoshop (version 25.7.0) and ImageJ (version 1.53a).

## Antibodies for western blot
Primary antibodies and dilutions used were as follows (see Supplementary Table 2):

mouse anti-flag (1:700, clone M2, F3165, Millipore Sigma);
mouse anti-α-tubulin (WB 1:2000, clone B-5-1-2, sc-23948, Santa Cruz);
mouse anti-actin (1:60,000, clone AC-15, F3022, Sigma);
mouse anti-HSP90 alpha/beta (1:500, clone F-8, sc-13119, Santa Cruz);
mouse anti-PKA Cα (1:500, clone A-2, sc-28315, Santa Cruz);
rabbit anti-PKA Cα (1:1000, 4782, Cell Signaling);
mouse anti-PKA IIα reg (1:500, H-12, sc-137220, Santa Cruz);
rabbit anti-PKA IIα reg (1:500, # A301-670A, ThermoFisher);
rabbit anti-AKAP9 (1:500, ab237752, Abcam);
rabbit anti-Centrin-2 (1:1000, ABE480, Merk Millipore);
mouse anti-Centrobin (1:1000, ab70448, abcam);
rabbit anti-phospho-Ser170-centrin-2 (1:430, 7TM Antibodies GmbH);
mouse anti-Vinculin (1:2000, 7F9, sc-73614, Santa Cruz);
rabbit anti-Plk4 (1:1000, 12952-1-AP, Proteintech).

Secondary anti-rabbit or anti-mouse antibodies conjugated to HRP or fluorophores were used at 1:10000 (111-035-146, 111-035-144, Jackson Immuno-Research, or 926-32210, 926-68071, Licor).

## Production of phosphorylated Ser170-Centrin-2–specific antibodies
The Centrin-2-pS170 antibody was provided by 7TM Antibodies. In detail, three rabbits were immunized by using a synthetic phosphopeptide spanning the region around S170 in the human Centrin-2 sequence [Cys-LRIMKKT(pS)LY$_{172}$]. The peptides were coupled to keyhole limpet hemocyanin (KLH). Antisera were used for affinity purification against their immunizing peptides, and specificity was then verified with synthetic phosphopeptides and corresponding non-phosphopeptides using dot-blot assays.

## Centrin-2-pS170 antibody neutralization with synthetic peptides
For neutralization of phospho-Centrin-antibodies, antibodies were pre-incubated with synthetic phosphorylated or non-phosphorylated peptides used for antibody production and kindly provided by 7TM Antibodies GmbH. For Western blotting, peptides were solved in water (1 mg/ml). Peptides (1:125) and antibody (1:250) were incubated in TBST following careful shaking for 2 h at RT. Antibody neutralization solution was centrifuged at 20.000 x *g* at 4 °C for 15 min. Bovine serum albumin in TBST was added to a final albumin concentration of 3%. Western blot membranes were incubated with the phospho-Centrin/phosphopeptide or phospho-Centrin/nonphosphopeptide solution at 4 °C over night. Membranes were washed with TBST before proceeding with incubation of secondary antibody.

## Detection of centrosome amplification
For quantification of amplified centrosomes, asynchronously growing cells were treated with GPER1 activators, PKA activating substances or PTX (Gαi inhibition) for 48 h. For *PKA* and *AKAP-9* repression, cells were first transfected with control or gene-specific siRNAs following further treatment with estrogens for 48 h. To block PKA, Gαs or adenylyl cyclase activity, cells were preincubated with PKI (14–22) amide, NF449 or SQ 22,536 respectively for 30 min before additional treatment with estrogens or GPER1 agonists. As a positive control, cells were transiently transfected with pCMVflag-*PLK4* (kindly provided by I. Hoffmann, DKFZ, Heidelberg, Germany). To visualize γ-tubulin, interphase cells were fixed with 2% p-formaldehyde (PFA) in PBS for 5 min at RT, followed with extraction with methanol at –20 °C for 5 min. Cells were washed once with PBS and blocked with 5% FCS for 20 min. Subsequently, the cells were stained for γ-tubulin (1:650, T3559, Sigma) and α-tubulin (1:650, clone B-5-1-2, sc-23948, Santa Cruz) to visualize centrosomes and microtubules, respectively, and with Hoechst (Hoechst 33342, 1:15,000, H3570, Thermo Fisher Scientific) to identify nuclei. Secondary antibodies conjugated to Alexa Fluor-488/-555 (1:1000, A-11029, A-21428, Invitrogen) were used. The amount of interphase cells with more than two centrosomes was quantified, i.e., more than two γ-tubulin signals localized at centrosomes, respectively[38].

## Detection of enlarged and phosphorylated Centrin

For quantification of enlarged and phosphorylated Centrin, asynchronously growing cells were treated for 48 h with GPER1 activators or PKA activating substances. To visualize γ-tubulin, Centrin, phospho-Ser170 Centrin-2, Cep135, CP110, and Centrobin (additional centriole proteins to specifically assess Centrin-related centrioles.), cells were pre-extracted with 0.5% Triton X-100 in BRB80 (80 mM K-Pipes, 1 mM MgCl2, and 1 mM EGTA) for 40 s followed by extraction with methanol at –20 °C for 7 min. For staining with Centrobin antibodies, cells were fixed with MeOH for 7 min at -20 °C, washed once with PBS, followed by extraction with 0.05% Triton-X-100 in PBS, three times for 5 min. Cells were blocked with sterile-filtered 5% BSA/PBS (Carl Roth) or 1% BSA and 0.05%Triton-X-100 in PBS (Centrobin) for 30 min at RT. Subsequently, cells were stained with antibodies against mouse anti-γ-tubulin (1:650, T6557, clone GTU88), mouse anti-Centrin-2/3 (1:300, clone 20H5, 04-1624, Sigma), rat anti-Centrin-2 (1:150, clone W16110A, 698602, BioLegend), rabbit Centrin-pSer170 (1:25, 7TM Antibodies GmbH), Cep135 (1:300, ab75005, Abcam), CP110 (1:100, ab243696, Abcam) or Centrobin (1:1000, ab70448, abcam) diluted in 1% BSA/PBS and 0.2% Triton-X-100 or 1% BSA/PBS and 0.05% Triton-X-100 (Centrobin) for 30 min at RT. Secondary antibodies conjugated to Alexa Fluor 488 or 555 (1:1000 or 1:500, A-11029, A-11034, A-11006, A-21424, A-21428, Invitrogen) were used. Where indicated, cells were stained with γ-tubulin conjugated to Alexa Fluor 647 (1:50, TU30, ab191114, abcam) in 1% BSA/PBS and 0.2% Triton-X-100 for 1.5 h at RT. To identify nuclei, cells were finally stained with Hoechst 33342 1:15000 (Invitrogen, Cat no. H3570). The amount of interphase cells with enlarged centriolar *foci* (i.e., signals with more than twice the diameter as in control-treated cells [see Fig. 3c]) was quantified.

## Detection of PKA at centrosomes

For detection and quantification of PKA at interphase centrosomes, asynchronously growing HCT116 or HCT116 cells depleted for AKAP450 were treated for 48 h with GPER1 activators or left untreated. To visualize PKA and γ-tubulin, cells were fixed, blocked and stained as described for Centrobin (see material and methods section '*Detection of enlarged and phosphorylated Centrin*'). Antibodies against rabbit anti-PKA Cα (1:1000, 4782, Cell Signaling), mouse anti γ-tubulin (1:650, GTU88, T6557, Sigma), and rat anti-Centrin-2 (1:150, clone W16110A, 698602, BioLegend) were used. Secondary antibodies conjugated to Alexa Fluor 488 or 555 (1:1000, A-11029, A-21428, Invitrogen) were used. To identify nuclei, cells were finally stained with Hoechst 33342 1:15000 (Invitrogen, Cat no. H3570). The amount of interphase cells with PKA colocalized to pericentriolar γ-tubulin was quantified from maximum intensity projections of 16 z-planes using the ZEN 3.1 blue edition software (Carl Zeiss Microscopy GmbH).

## Expansion microscopy

Expansion microscopy (ExM) was carried out according to Tillberg et al., Chozinski et al. and Gao et al. [66–68] with some modifications. In detail, asynchronously growing cells were treated for 48 h with 10 nM BPA and fixed as previously described. Cells were stained with antibodies against Centrin-2 1:150, clone W16110A, 698602 (BioLegend) and γ-tubulin (1:500, GTU88, T6557, Sigma) diluted in 1% BSA/PBS and 0.2% Triton-X-100 for 30 min at RT. Cells were subsequently stained with secondary antibodies conjugated to Alexa Fluor 488 or 555 (1:500, A-11006, A-21424, Invitrogen) for 1.5 h. To identify nuclei, cells were stained with Hoechst 33342 (1:15000, H3570 Thermo Fisher Scientific). Then, cells were post fixated using 0.25% Glutaraldehyde (freshly prepared in PBS, Sigma) for 10 min at RT and in the following step linked to an expandable gel matrix (sodium acrylate 33%, acrylamide 40%, N,N′-Methylenebisacrylamide 1%, sodium chloride 5 M) by adding the solution on top of the cells for 1 h at 37 °C to allow the gel to polymerize. The gel was then incubated in digestion buffer containing Proteinase K for 1 h at 37 °C. Subsequently, the gel was expanded in A. dest. for 2 h and stained again with Hoechst 33342 for 15 min at RT. Finally, 80% of sucrose diluted in H2O was added on top of the gel and incubated overnight at RT. To avoid bleaching, the gel was incubated

in 5% DABCO for 30 min at RT before being placed on a high precision coverslip coated with poly-L-lysine und sealed using eco-sil (Picodent). Magnification index was determined by measuring the size of the gel before and after expansion.

## Immunofluorescence microscopy

Microscopy of fixed cells was performed on a Zeiss AxioObserver Z1 microscope (Zeiss) equipped with an apotome 2.0 module, a heated chamber, and an AxioCam MRm camera (Zeiss). Images were recorded with an 63x, 1.42 oil immersion objective and a z-optical spacing of 0.28 μm, processed and analyzed with ZEN 3.1 blue edition software (Carl Zeiss Microscopy GmbH) and shown as maximum intensity projection. Image sections had a size in μm of $25 \times 25$, $35 \times 35$ or $50 \times 50$ or $120 \times 120$ for ExM. Images for signal intensities of phosphorylated Centrin-2 or PKA were shown as maximum intensity projection from 16 z-planes. Image sections had a size in pixels of $50 \times 50$ (Centrin-2) or $22 \times 22$ (PKA). For pixel quantification of phospho-Centrin-2, signals were normalized to signal intensities derived from centriolar Centrin-2. For pixel quantification of PKA, signals were normalized to signal intensities derived from pericentriolar γ-tubulin. Images related to the proximity ligation assay were recorded on a Zeiss LSM880/Airyscan microscope with a 40x dry objective and a z-optical spacing of 0.50 μm (14 slices), processed with ZEN 2.1 black edition software (Carl Zeiss Microscopy GmbH) and shown as maximum intensity projection. Images were further processed using Adobe Photoshop (version 25.7.0).

## Transmission electron microscopy
### Sample Preparation

Chemical fixation. HCT116 cells treated with DMSO or 10 nM Bisphenol A were detached from the culture plate using Trypsin/EDTA (0,05%/0,02%) w/o: Ca and Mg (Sigma). Cell pellets were collected by centrifugation at 130 x *g* for 5 min at RT, washed once with 1x PBS, and resuspended in 1 ml of ice-cold fixative solution 25% glutaraldehyde (agar scientific) in 1x PBS (Carl Roth). The suspension was incubated for 2 min on ice. After centrifugation at 2400 x *g* for 10 min at RT, the pellet was resuspended in 1 ml of fixative over night at 4 °C before being pre-embedded in agarose.

Agarose pre-embedding. Fixed cells were centrifuged at 2400 x *g* for 10 min at RT. The supernatant was gently removed and discarded, then 1 ml of 1x PBS (Carl Roth) was added. The cell pellet was transferred into a 38 °C preheated ThermoMixerC (Eppendorf) equipped with a 2.0 ml centrifuge tube top piece with lid. Separately, 1 ml of freshly prepared 2% agarose (w/v) QA-Agarose™ Low Melting (MP Biomedicals) was added to another 2.0 ml tube and also incubated at 38 °C in the ThermoMixerC. The samples were gently rocked at 300 rpm. Once both the cell pellets and the agarose reached 38 °C (incubated at least for 30 min), the residual PBS above the cell pellets was carefully removed and discarded. The cell pellet was resuspended in 50 μl of liquid agarose and incubated for 30 min at 38 °C. Subsequently, the samples were gently rocked at 300 rpm for 2 h while maintaining the temperature at 38 °C (the thermomixer acted hereby as a centrifuge and a cell pellet was formed). Finally, the tubes were transferred to 4 °C to allow the agarose to solidify.

Post-fixation, dehydration and Epon embedding. The agarose pre-embedded cell pellet was carefully removed from the centrifuge tube. Areas with pure agarose were cut off. The cell pellet was cut into suitable pieces (ca. 0.5 mm cubes) which were transferred to a centrifuge tube containing 1 ml osmium tetroxide (1% (v/v), Electron Microscopy Sciences). The sample was incubated on a rocking plate (horizontal, 300 rpm) for 1 h. The post-fixated cell pellets were transferred to an EM embedding machine (Lynx II, Electron Microscopy Sciences) which contained the following chemicals and reagents, respectively: Tannic acid (0.1% (w/v) in aqua bidest., 30 min incubation time, Serva), sodium sulfate (1% (w/v) in aqua bidest., 2 ×30 min incubation time, Merck), aqua bidest. (3 ×30 min incubation time), uranyl acetate (2% (w/v) in aqua bidest., 1 h incubation

time, Electron Microscopy Sciences) and ethanol in several concentrations (30, 50, 70, 95, 100% (v/v), each concentration for 1 h incubation time, Carl Roth). Finally, the cell pellets were transferred into 100% (v/v) absolute ethanol (Carl Roth) where they resided until Epon embedding. All vials were filled with 20 ml of chemicals and reagents. Then the Epon embedding was carried out in the EM embedding machine (Lynx II), which contained propylene oxide and Epon or mixture of both, respectively. The Epon contained the following chemicals and reagents, respectively: agar 100 resin (24 g, Agar 100 resin Kit, Agar Scientific), [3-(2-dodecenyl)succinic anhydride] (9 g, DDSA, Agar 100 resin Kit, Agar Scientific), (Methyl-5-norbonene)-2,3-dicarboxylic anhydride (15 g, MNA, Agar 100 resin Kit, Agar Scientific) and bethyldimethylamine (1,4 g, BDMA, Agar 100 resin Kit, Agar Scientific). The Epon was freshly prepared and allowed to stir on a magnetic stirring plate for 20 min before use. The dehydrated cell pellets were transferred into 20 ml propylene oxide where they incubated for 2×1 h. Then the Epon was introduced into the sample by increasing the Epon content stepwise using the following ratios of propylene oxide and Epon (v/v): 2/1, 1/1 and 2/1, respectively. Finally, the cell pellets were transferred into pure Epon (incubation for a minimum of 4 h) where they resided until further use. All vials were filled with 20 ml of chemicals and reagents.

Serial sectioning. The resin block was trimmed (Leica EM Trim, Leica), then serial sections of 70 nm thickness (diamond knife, ultra 35°, 2.00 mm DiATOME) were made using an ultramicrotom (Leica Ultracut UC T, Leica). Two to four consecutive sections were placed on a formvar coated grid (Cu, 200 mesh, Plano GmbH). Several grids were prepared per sample and allowed to air dry on a filter paper in a petri dish.

Post staining. The post-staining was mainly conducted according the protocol as outlined in Köhrer et al.[81]. Throughout all following steps, the grid floated on the surface of the drops during incubation, with the sections facing downward. Briefly, a parafilm was prepared with all reagents, whereby the size depended on the number of grids to be stained as per grid one row of reagents was used (one drop was ca. 25 μl). First, the grid was placed for 1 min onto the methanol (50% v/v, Morphisto) drop, then transferred to uranyl acetate (2% (v/v) aqueous uranyl acetate (Electron Microscopy Sciences) in 70% (v/v) methanol (Morphisto)) and incubated for 5 min. The parafilm was covered with a lid. The grid was rinsed in all 4 methanol 50% v/v (Morphisto) drops, then placed in the first water drop. The grid was incubated for 1 min with the lid on, then the process was repeated two more times before moving over to the lead citrate (Science Services) staining. Here the lead citrate drop is placed next to a sodium hydroxide pellet (Carl Roth) to avoid oxidation of the lead(II). The grid was incubated for 3 min in lead(II) citrate (Science Services) before it was washed four times with aqua bidest. Each washing step was carried out for 1 min. The grids were allowed to air dry on a petri dish on filter paper. Then the individual grids were subjected to transmission electron microscopy (TEM) imaging.

## TEM imaging and image analysis
The images were acquired using a Jeol TEM the JEM 1400 Plus. The TEM was being operated at 120 keV and 64 mA. Individual cells were imaged at 10k x magnification. This magnification was sufficient to identify centriole(s) within a single cell. If the cell contained centriole(s) then these were imaged at 60k x magnification. The images were acquired with an Olympus camera using the software ITEM (Olympus, Version ITEM-E-23082007). Further image analysis such as the measurement of dimensions (length, width and diameter) was carried out using Image J (Version 1.53a).

## Proliferation assay
$5 \times 10^4$ HCT116 or HCT15 cells were seeded in 6-well or 12-well plates to quantify cell proliferation. Cells were washed once they had attached to the surface and pre-treated with $H_2O$, 5 μM PKI (14–22) amide (myristoylated)

or 10 nM NF449 for 30 min. Subsequently, cells were exposed to 10 nM Bisphenol A or 100 nM G-1 for 48 h. To exclude dead cells, trypan blue was added in a ratio of 1:2 before manual counting using a hemacytometer (purchased from Fein-OPTIK).

## PKA activity assay
To determine the PKA activity, asynchronously growing cells were treated with GPER1 activators or the adenylyl cyclase activator forskolin (FSK) as a positive control for 48 h. Subsequently, cells were harvested, lysed, and the protein concentration was determined as described above. PKA activity was assayed in each sample using a commercially available competitive ELISA–PKA Kinase Activity Kit (Enzo Life Science) according to the manufacturer's instructions. For each assay reaction, 50 ng of the lysate was added. Absorbance in the PKA assay was measured at 450 nm with the TECAN Infinite M200 plate reader. To determine the PKA-specific activity per sample, one well was treated with 5 μM of the PKI inhibitor, PKI (14–22) amide (myristoylated). PKA activity was then assessed using the PKA activity assay kit, and the activity obtained in samples treated with PKI was subtracted from the total activity for each sample as published[57].

## Proximity ligation assay
To visualize interaction of Centrin-2 and PKA-Cα, a Duolink® Proximity Ligation Assay (PLA) was performed. In detail, Bisphenol A-treated cells (48 h) were processed as described in the materials and methods section '*Detection of enlarged and phosphorylated Centrin*' and stained with antibodies against mouse anti-Centrin-2/3 (1:300, clone 20H5, 04-1624, Sigma) and rabbit anti-PKA Cα (1:50, 4782, Cell Signaling) diluted in 1% BSA/PBS and 0.2% Triton-X-100 for 30 min at RT. Subsequently, slides were washed once with PBS and then further processed according to the manufacturer's protocol. After the final wash step with wash buffer B for 1 min at RT, slides were washed once with PBS before additional staining with antibodies against γ-tubulin-647 (1:50, clone TU-30, ab191114, abcam) diluted in 1% BSA in PBS + 0.2% Triton X-100 for 1.5 h at RT. To identify nuclei, cells were finally stained with Hoechst 33342 1:15000 (Invitrogen, Cat no. H3570).

## Intracellular cAMP level measurement
Intracellular cAMP levels were determined in asynchronously growing cells treated with GPER1 activators, or the adenylyl cyclase activator forskolin (FSK, positive control) using cAMP-Glo™ Assay (Promega) according to the manufacturer's instructions. In detail, 3.1 Mio cells were seeded in 6-well plates (Faust Lab Science) over night. One extra well per cell line served for determination of final cell numbers before treatments. Cells were treated with the Induction Buffer containing a ratio of 1:200 phenol red-free RPMI 1640 and 500 μM of the PDE-inhibitor 3-Isobutyl-1-methylxanthin, IBMX (Sigma) for 20 min at 37 °C and 5% $CO_2$ to prevent degradation of available intracellular cAMP. Subsequently, cells were additionally treated with the above mentioned GPER1 activators or FSK for 30 min at 37 °C and 5% $CO_2$. Where indicated, cells were pretreated with DMSO, 10 nM NF449 or 10 μM SQ 22,536 for 30 min, before additional exposure with 100 nM G-1 for additional 30 min. Then, the medium was aspirated and the cells were detached with 1 mL of 1:100 PBS/EDTA for each well and centrifuged for 5 min at 130 x g. The cell pellets were resuspended in 25 μL Lysis Buffer of the Glo-cAMP Assay kit plus 25 μL Induction Buffer and shaken for 30 min. Subsequently, 40 μL of each sample was transferred into a 96 Well Cell Carrier Plate (Perkin Elmer), and the intracellular cAMP level was assayed with cAMP-Glo™ Assay (Promega) using the protocol provided by the manufacturer. In detail, 40 μL of 1:400 Detection Buffer and PKA was added to each well, carefully shaken on the orbital shaker for 60 s and incubated for 20 min at RT. Afterwards, 80 μL of the Kinase Glow Buffer was added to each well and the lysate was carefully shaken on the orbital shaker for 60 s and incubated for 10 min at RT. Finally, luminescence was read using the Synergy Neo2 Reader (BioTek) with the program Gen5, with an integration time of 1 s per well, and a gain of 135.

## Statistics and reproducibility

All graphs were created with PRISM 10.1.2 and individual data points were plotted. Statistical analyses were performed with R (v4.0.2 or v4.3.2) or GraphPad PRISM 10.1.2 software. All quantifications are based at least on $n = 3$ biological replicates if not indicated otherwise. All statistical tests used for analysis of specific experiments, exact value of n, and dispersion and precision measures are indicated in the figure legends. P-values were indicated as numbers in the figure legend and/or Supplementary Table 1. $P \geq 0.05$ was defined as not significant.

All quantifications of centrosome amplification are based on 3-4 independent experiments, in which 600–1600 interphase cells (centrosome amplification) were evaluated, and mean values with standard deviation (SD) were calculated. Bootstrap procedure using R.4.0.2 was used to calculate pvalues. Determination of cAMP level and PKA activity is based on the evaluation of 5–6 (cAMP) or 6-7 (PKA) independent experiments. PKA assays were performed in two technical replicates. Mean values with standard error of the mean (SEM) were calculated. Paired t-test (two-tailed) was used to calculate P-values. Distances of centrioles were plotted as Box and whiskers with min to max presentation and individual data points using GraphPad PRISM 10.1.2. Numbers of analyzed cells were given in the respective figure legend. Paired t-test (two-tailed) was used to calculate Pvalues. Quantification of the relative protein levels shown in Supplementary Fig. 7a were based on 3 different experiments, and mean values with SEM were calculated. Data were analyzed using a two-way ANOVA with post hoc Tukey's multiple comparisons test to compare the control to each of the conditions. Quantification of signal intensities for Centrin-pS170 and PKA normalized to total Centrin-2 or γ-tubulin were based on 3 different experiments, and geometric mean values with 95% CI (Centrin-2) or box and whiskers with 10–90 percentile (PKA), were calculated. Data were analyzed using a Mann–Whitney's test to calculate P-values. The proportion of centrosome amplified cells with enlarged Centrin-2 foci was determined from 3-4 different experiments, and mean values with SD were determined. P-values were calculated with R.4.3.2 using an adapted bootstrap procedure. To quantify enlarged phospho-Centrin, the same immunofluorescence slides used to assess the proportion of enlarged Centrin were analyzed for phospho-Centrin. Cell proliferation assays were performed in 3 different experiments (biological replicates). Representative examples of immunofluorescence experiments and Western blots that are shown in the figures were repeated at least 3 times.

## Sample size

For all quantitative data described in the article, the sample size (n) was reported as an exact number in the corresponding figure legends. Sample sizes to determine centrosome amplification upon different conditions are related to previous studies[38]. Initial pilot experiments for enlarged/phospho-Centrin were performed to estimate intra-variability of experimental conditions and to determine optimal sample size for statistical analysis.

## Data exclusions

No data were excluded with exception for cAMP assays. Here, a cut-off was set derived from the respective positive control-treatment that was set to 50% for HCT116 or 25% for HCT-15 related to FSK (Fig. 1e, f) and 30% for HCT116 or 25% for HCT-15 related to G-1 (Supplementary Fig. 1c). Values below these cut-offs were excluded and values above these cut-offs were considered as valid. The different cut-offs were chosen related to different base-levels of RLU values between both cell lines and treatments with the respective positive control. One experiment for HCT116 was excluded due to abnormally low RLU DMSO treatment, and one for HCT-15 due to abnormally high RLU, likely caused by cell seeding issues (Supplementary Fig. 1c).

## Randomization

This in vitro study is based on the analyses of single cells (centrosome amplification; enlarged and phospho-Centrin), whole cell populations (cell proliferation), and whole cell extracts (Western blot, Immunoprecipitation, and PKA/cAMP assays), and randomization did not apply. As described in the Results and Materials and Methods sections, cells were seeded and then treated with the different substances for the indicated time periods.

## Blinding

The investigators who collected these data were not blinded to the substance treatments. Unbiased analysis of data was carried out wherever possible.

All scientists were trained to identify relevant phenotypes. Two independent scientists quantified centrosome amplification and enlarged/phospho-Centrin phenotypes, producing consistent results. These quantifications were repeated using the same treatments (E2, BPA., DES, ICI, Tam, G-1) across multiple (sub-)figures and verified in two different cell lines, both yielding consistent outcomes. To analyze centrosome amplification and enlarged/phospho-Centrin, various substances, including endogenous estrogens, xenoestrogens, and synthetic ligands, were used, all producing consistent effects. Additionally, two different Centrin-antibodies were employed to quantify centrosome amplification and enlarged Centrin-labeled centrioles, as described in the results section. Quantitative data for cAMP and PKA assays were automatically analyzed using the Synergy Neo2 Reader (BioTek) or the TECAN Infinite M200 plate reader.

## Bootstrap procedure

Data from most experiments are (i) count data dealing with (ii) fixed effects (substance treatments) and (iii) random effects (biological replicates). The following *bootstrap* procedure considers all three conditions (i-iii) and was used for statistics if not stated otherwise.

For the control and each treatment i probabilities of centrosome amplification were estimated by $\pi_{ij} =$ (number of cells with centrosome amplification)/(number of all observed cells) (j = 1,..,nb). From this nb x 100,000 probabilities were drawn with replacement for each treatment and the control. With these probabilities, the number of cells with centrosomes ($n_{ij}$) were randomly chosen from the corresponding binomial distribution. The number of cases, where the sum over the nb numbers of cells with centrosome amplification for a treatment was equal or smaller than the corresponding number for the control, were counted. This number divided by 100,000 is then the P-value.

## Adapted bootstrap procedure

The following *bootstrap* procedure was applied to test whether the relative frequency of enlarged Centrin-labeled centrioles in centrosome amplified cells is increased after GPER1 activation compared to the DMSO control: The null hypothesis is that the relative frequencies of enlarged Centrin *foci* are equal while the relative frequencies of centrosome amplified cells is greater with GPER1 activation. For GPER activation as well as for the control we had a minimum of three biological replicates. For each biological replicate there is a relative frequency of enlarged Centrin *foci* among amplified centrosomes. These frequencies were randomly assigned to control and substance treatments. The frequencies regarded as probabilities were used to sample the number of enlarged Centrins from binomial distributions. This sampling in turn results in relative frequencies for control and treatment, respectively. The difference between the two was calculated. The procedure was repeated $N = 100,000$ times. The number of the resulting differences greater than the observed divided by N was then regarded as P-value.

## Reporting summary

Further information on research design is available in the Nature Portfolio Reporting Summary linked to this article.

## Data availability

All data supporting the findings of this study are available within the article and its Supplementary Information. The source data behind all graphs in the main and Supplementary Figs. are available as Supplementary Data 1. The original images of the immunoblots presented in this study are included in

Supplementary Figs. 8-10. All other data are available from the corresponding author on reasonable request.

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

## Acknowledgements

We thank Ingrid Hoffmann (DKFZ, Heidelberg, Germany) for plasmids, Beate Döring, and Birgitta Slawik for technical assistance, Matthias Steinfath for help with statistics, Marianne Grafe (Potsdam University) for help with Expansion Microscopy, and all colleagues from the BfR/Bf3R, especially Marta Barenys, Jose Muino Acuna, Norman Ertych, Chris Tina Höfer, and

Ralph Gräf (Potsdam University) for scientific input and/or comments on the manuscript. This work was funded by the Deutsche Forschungsgemeinschaft (DFG, German Research Foundation) – Project No. 465732234; STO 1306/2-1 to A.S.

## Author contributions

J.F.: Data curation; investigation; methodology; writing—contributed to Methods and Protocols; writing—review and editing. M.Bü.: Data curation; investigation; methodology; writing—review and editing. J.M.S.: Data curation; investigation; methodology; formal analysis; writing—contributed to Methods and Protocols; writing—review and editing. C.Z.: Data curation; investigation; methodology; writing—contributed to Methods and Protocols; writing—review and editing. M.Bu.: Data curation; investigation; methodology; writing methods—review and editing. M.Be.: Data curation; investigation; writing—review and editing. G.S.: Resources, writing—review and editing. A.S.: Conceptualization; funding acquisition; data curation; investigation; methodology; visualization; formal analysis; supervision; writing—original draft; project administration; writing—review and editing.

## Funding

## Competing interests

The authors declare no competing interests.
