## [Transparent Peer Review file · Communications Biology]

A GPER-PKA-Centrin axis regulates centrosome numbers and centriole integrity in colon cancer cells

Corresponding Author: Dr Ailine Stolz

Version 0:

Decision Letter:

**** Please ensure you delete the link to your author homepage in this email if you wish to forward it to your coauthors ****

Dear Dr Stolz,

Your manuscript entitled "A GPER-PKA-Centrin axis regulates centrosome numbers and centriole integrity in colon cancer cells" has now been seen by 3 referees, whose comments are appended below. You will see from their comments copied below that while they find your work of potential interest, they have raised quite substantial concerns that must be addressed. In light of these comments, we cannot accept the manuscript for publication, but would be interested in considering a revised version that addresses these serious concerns.

We hope you will find the referees' comments useful as you decide how to proceed. Should further experimental data or analysis allow you to address these criticisms, we would be happy to look at a substantially revised manuscript.

In particular, please note that the following revisions would be necessary for us to contact our referees again: the referees focus on the question whether the extra centrin dots observed are indeed extra centrioles or rather extra centrin foci, which should be addressed accordingly - ideally by EM. We acknowledge this is a challenging technique, but this should be attempted as it's a major concern shared by all 3 reviewers.

We are committed to providing a fair and constructive peer-review process. Do not hesitate to contact us if you wish to discuss the revision or if there are specific requests from the reviewers that you believe are technically impossible or unlikely to yield a meaningful outcome.

If you decide to submit a revised version, we ask that you ensure your manuscript complies with our editorial policies. Please see [our revision checklist](https://www.nature.com/documents/CommsBio-file-checklist-revision.pdf) for guidance on formatting the manuscript and complying with our policies. A comprehensive guide to our formatting requirements for final submissions is also available for your reference [here](https://www.nature.com/documents/commsj-life-style-formatting-guide-accept.pdf).

Please use the following link to submit your revised manuscript, point-by-point response to the referees' comments (which should be in a separate document to any cover letter) and any completed checklist:

Link Redacted

****This url links to your confidential home page and associated information about manuscripts you may have submitted or be reviewing for us. If you wish to forward this email to co-authors, please delete the link to your homepage first****

We hope to receive your revised manuscript within 3 months, but appreciate that every situation is unique. Please take as long as necessary to address these concerns in full, including performing any additional experimental work required. We look forward to receiving your revised manuscript when it is ready and will not enforce any specific deadline. However, please bear in mind that if the revision process takes significantly longer than six months, we may take into account whether anything similar has been accepted for publication at Communications Biology or published elsewhere in the meantime.

Please do not hesitate to contact me if you have any questions or would like to discuss the required revisions further. Thank

you for the opportunity to review your work.

Best regards,

Manuel Breuer, PhD
Deputy Editor, Communications Biology
4 Crinan Street
London N1 9XW, UK
orcid.org/0000-0003-1982-2517

on behalf of

Patrick Meraldi, PhD
Editorial Board Member
Communications Biology
orcid.org/0000-0001-9742-8756

Referee expertise:

Referee #1: centrosome biology

Referee #2: cell division, centrosome biology

Referee #3: centrosome biology

Reviewers' comments:

Reviewer #1 (Remarks to the Author):

In the present manuscript Fahrländer and colleagues describe a role of the G protein-coupled estrogen receptor GPER1 in regulating numerical and structural centrosome aberrations. They demonstrate that GPER1 activation by estrogens or environmental estrogenic compounds can lead to centrosome amplification (CA) in colorectal cancer cell lines. They show that CA is mediated by elevated cAMP levels and protein kinase a (PKA) activation downstream of GPER1 ligation. Half of the cells harbouring extra centrosomes show enlarged and displaced Centrin-2 foci indicating alterations in centriole structure. They further seek to establish a causal link between centrosomal PKA recruitment, Centrin-2 phosphorylation and lack of centriole integrity.

Collectively, the authors provide evidence for a GPER-PKA-Centrin signalling axis regulating centriole numbers and structure in colorectal cancer cells.

Major comments:

The manuscript is well written and easy to follow and presents novel insights into the causes of centrosome amplification in cancer cells. The experiments are technically well performed and well-presented. However, they do not fully support the authors conclusions. Moreover, the overall degree of CA in the chosen cell lines is very low with less than 10% of cells showing extra centrosomes. This raises concerns regarding the biological significance and whether their findings would also apply to cell lines that show a high proportion of cells with CA.

Specific comments:

1. The authors convincingly show that PKA activation is involved in CA in colorectal cancer cell lines. Yet, the conclusion that PKA is recruited to centrosomes via AKAP-450 is not well supported by the data presented (reduced CA after AKAP-450 knock-down). The authors should show localization of PKA in the presence or absence of GPER1 activation.
2. Centriole integrity was determined by immunofluorescence using antibodies against Centrin-2. The authors detect pronounced Centrin-2 staining after GPER1 activation referred to as 'enlarged centrioles'. The pictures shown do not allow to precisely measure centriole length (even after expansion microscopy; what is the degree of expansion?). To determine centriole size and conclude about structural centriole defects, the authors should present higher magnification images and/or carry out transmission electron microscopy (TEM).
3. In the present study CA is induced only after GPER1 activation (in the presence of estrogens). They refer several times to the work from the Bettencourt-Dias lab where they catalogue centriole numbers in human cancer cell lines and demonstrate CA in HCT116 cells in the absence of any stimulus (Marteil et al). Why is CA only induced after GPER1 activation in one study but not the other?
4. Why does PLK4 overexpression (OE) not result in a larger number of cells with extra centrioles (Figure 3k, l)? The authors should demonstrate efficient PLK4 OE resulting in CA in their cell lines.

5. The authors claim that PKA activation is responsible for CA in colorectal cancer cells by phosphorylating Centrin-2. While the generated phospho-specific Centrin-2 antibody seems to work very well, only a minor reduction in phospho-Centrin-2 was detected after PKA inhibition (Figure S4a,b). This strongly suggests, that other kinases are involved in phosphorylating Centrin-2 at Serine 170.

6. Moreover, data supporting the model that phosphorylation of Centrin-2 causes structural centriole aberrations is weak. The authors should generate phospho-mutants of Centrin-2 to demonstrate its impact for centriole structure.

Reviewer #2 (Remarks to the Author):

In this manuscript, Fahrländer and colleagues propose a mechanism by which estrogen-mediated signaling regulates centrosome number in colorectal cancer-derived cell lines. They demonstrate that activation of the estrogen receptor GPER1 triggers PKA-dependent phosphorylation of centrin at the centrosomes, which, in turn, leads to the formation of additional centrin foci. The authors conclude that these extra centrin foci are responsible for centrosome amplification.

The finding that estrogen promotes centrosome amplification via PKA is an important finding with potential implications. Understanding the underlying molecular mechanism is a valuable contribution to the field. However, the claims of the paper are not sufficiently supported by the presented data. The initial experiments, which explore the role of GPER1 and PKA in centrosome amplification through chemical inhibition and activation, are well designed and appropriately controlled (Fig. 1–2). The authors also show that PKA activation leads to ectopic centrin foci. Unfortunately, the authors do not provide any conclusive evidence that ectopic centrin foci are related to centrosome amplification. Thus, I think it remains unclear how estrogen-dependent PKA activity leads to centrosome amplification. I am worried that the authors claim throughout the manuscript that PKA enlarges centrioles based on their observation of ectopic centrin foci. This is a dangerous assumption which could be wrong. The existence of centrin foci that are independent of centrioles has been observed (PMID: 33983387). Thus, I think that at this point it is unclear how PKA activation leads to centrosome amplification. However, I think that this point could be addressed by a few additional experiments outlined below.

Major issues:

1. Throughout the manuscript, the authors consider centrin foci as centrioles. However, it is very likely that most centrin foci don't belong to centrioles. The authors even provide strong evidence that the centrin foci are not centrioles in figures 3a-b, where they show that most centrin foci do not colocalize with gamma-tubulin. Gamma-tubulin was used in the first place to count the amplified centrosomes. They also do not show the co-localization of any other centriole marker with the centrin foci. Thus, there is no evidence that support the claim that all centrin foci are centrioles.

The authors should pay attention to a recent publication, which describes ectopic centrin foci in TRIM37-deleted cells (PMID: 33983387). Using CLEM this paper showed that ectopic centrin foci do not reassemble centrioles. They also found a small increase in centriole number. However, this effect was independent of centrin and was explained by mitotic defects that were caused by ectopic centrin foci, which were independent of centrin.

In the light of this work, the authors must rigorously test the relationship between centrin foci and centrioles. They also have to address the possibility that other factors such as ectopic Centrin foci and/or mitotic defects could lead to centriole amplification through the described PKA-dependent mechanism.

2. The authors provide evidence that estrogen signaling leads to PKA-dependent centrosome amplification. They use gamma-tubulin as marker to count centrosomes. However, the connection between centrin and extra gamma-tubulin foci remains unclear as discussed above. To conclude that centrin induces centrosome amplification, the authors must show that centrosome amplification (gamma tubulin staining) upon PKA activation does not happen in centrin-deleted cells. Centrin is not essential and can be deleted in HCT116 cells using standard CRISPR-Cas9 gene editing. This is a straightforward experiment, which is necessary to determine the role of centrin in the described mechanism.

3. The authors claim that Centrin associates with PKA using immunoprecipitation experiments. However, all IP experiments were performed without negative control (IgG control, empty beads). Therefore, these experiments are inconclusive and do not provide any evidence that Centrin associates with PKA.

Other Points:

1. The authors quantify throughout the paper centrosome amplification as shown in figure 1a. They classify cells with 2 centrosomes as "Not amplified". This represents an inaccuracy, which should be avoided or explained. In G1-phase just one centrosome should be present. Two centrosomes in G1-phase indicate a prior defect, possibly centrosome amplification or cell division failure. Since there is no discrimination between G1 and S/G2-phase, there is no way to determine whether cells with 2 centrosomes have the normal number of centrosomes.

2. Cells are exposed to NF449 and estrogens for 48h prior to quantification. This seems to be quite long considering that the cell division time for HCT116 cells is less than 20h. The authors should address the possibility that mitotic failure rather than centrosome amplification leads to the increase in centrosome number. One possibility is to show that centrosome amplification happens in cells during interphase.

3. Does NF449 treatment impair cell cycle progression? If yes, then the observed effect on centrosome number could be independent of centrosome amplification per se.
4. In Figure S1C-D the significance of the difference between H₂O+DMSO and DMSO+G1 is shown. However, the relevant difference, for which the significance is missing, is between DMSO+G1 and SQ22,536+G1. These measurements have to be significantly different to support the related claims made in the manuscript.
5. The authors claim that PKA phosphorylates proteins that are involved in centrosome function. However, they never mention which proteins are supposed to be phosphorylated by PKA. I think this is important to clarify.
6. The connection between AKAP9 and AKAP450 should be clarified. Readers most likely do not know that they are the same protein/gene.
7. AKAP9 is not essential. Considering potential siRNA-related side effects, the authors should confirm their results with CRISPR-Cas9-mediated gene knockout.
8. The authors mention several times that the observed effect is related to PKA at centrosomes. However, they never provide any evidence for this. To make these claims they must address the effect of estrogen signaling on centrosome number and PKA localization in AKAP9-deleted cells.
9. Does GPER activation or cAMP signaling influence PKA activity specifically at the centrosome or global? If this question cannot be addressed, the authors must discuss this point in the discussion and tone down related claims.
10. It is unclear to me how G1 and G2 cells were distinguished in Figure 3a/b?
11. The conclusions based on PLK4 overexpression are not clear. Instead of showing rosette-like structure, which is not clear at all (Fig. 3J), the authors must quantify centrosome number as they did before (e.g., gamma-tubulin foci after 48h of PLK4 overexpression).
12. From the data shown in Figure S3f-g, I'm not convinced that all extra centrin foci are centrioles. There is no quantification for S3f-g. CP110 and CEP135 are co-stained with gamma-tubulin, but not with Centrin. In the main figures, it is shown that Centrin foci do not always co-localize with gamma tubulin. Thus, they are most likely not centrioles. The authors have to show co-localization of centrin with other centriole markers to claim that all centrin foci are centrioles.
13. The PKA inhibitor PKI only had a marginal effect on Centrin phosphorylation (Fig. S4a,b), yet the authors also show that PKI completely rescues centrosome amplification upon estrogen treatment (Fig. 2). To support the claim that PKA-mediated phosphorylation of centrin leads to extra centrin foci, the authors have to show the effect of PKA inhibition (PKI) on the phosphorylation of centrin in interphase cells.
14. The authors "hypothesize that Centrin-2 and PKA physically interact at centrosomes" (line 378). However, they never test this idea with appropriate methods that can determine the location of interaction (e.g.; Proximity ligation assay).
15. I don't understand this sentence "Of note, while Hsp90, a well-known binding partner of PKA62, co-immunoprecipitated with PKA regulatory subunits, Centrin-pS170 did not interact with PKA-R11, as expected" (lines 392,393).
16. What does the asterisk show in Figure 6b-h?
17. Line 449/450: "In this study, we have now filled this gap by showing that activated GPER1 triggers the recruitment of PKA to the centrosome, where the kinase affects the centriole size and centrosome number." The authors do not show that activated GPER1 triggers recruitment of PKA to centrosomes. They also do not show that GPER1 influences centriole size.
18. Line 86: References 24 and 27 do not show that PKA localizes at centrosomes. Reference 26 is a 22-year-old review that cites one paper from 1999 (PMID: 10328961) that claims that PKA localizes at centrosomes with the resources that were available at that time. Considering the importance of PKA for the conclusion of this paper the authors should provide the most current and most solid references that show the localization of PKA at centrosomes.
19. Line 96: References 24, 27 and 32 do not show that AKAP450 recruits PKA to centrosomes. Please provide references that support the provided information that is used for the argumentation.
20. Line 111-112: The authors write "Upon activation, PKA is recruited to centrosomes via AKAP-450, where it orchestrates the phosphorylation of Centrin-2 at serine 170". However, the authors don't provide any evidence for their claim that PKA is recruited to centrosomes following GPER1 activation.
21. Line 113: The authors write that centrin phosphorylation by PKA "occurs inappropriately at non-mitotic amplified centrosomes". As described above, there is no evidence that shows that the centrin foci belong to amplified centrosomes.
22. Line 125-127: The authors write "GPER1 binding to G protein Gas triggers the activation of

PKA, which is recruited to centrosomes". However, none of the provided references (26, 43, 44, 45) provides any evidence that support this claim.

23. Line 176: The authors write that "upon activation, PKA catalyzes the phosphorylation of a variety of proteins including those involved in centrosome function". Reference 19 does not mention PKA. Reference 24 shows centrin phosphorylation. Reference 26 mention only centrin as relevant centrosomal target of PKA. Thus, I could only find centrin as centrosomal target of PKA. Are there other publications that support the claim that PKA phosphorylates a variety of centrosomal proteins?

24. Line 192-193: None of the references 26, 32 and 44 shows that AKAP450 recruits PKA to centrosomes.

25. Line 342-343: The authors create the impression that PKA is the major kinase phosphorylating Centrin. However, Figures S4a-b indicate that the majority of phosphorylation is independent of PKA. Please rephrase to make this point clear.

26. Line 361 and 372: There is no evidence that centriole size is altered by PKA.

27. Line 428-431: The authors don't provide any evidence that GPER1 triggers the recruitment of PKA by AKAP450 to centrosomes.

28. Line 218-220, 203-268, 432-433, 443, 450, 451, 460, 466-467, 476, 486-488, 498-499: The size of the centrin foci cannot be used to conclude the length or size of centrioles. The authors do not provide any conclusive evidence that centrioles are enlarged.

Reviewer #3 (Remarks to the Author):

A GPEER-PKA-Centrin axis regulates centrosome numbers and centriole integrity in colon cancer cells

Centrosome number control is a critical aspect of cell division and an intriguing area of research. During normal cell division, centrioles duplicate once per cell cycle during the S phase and subsequently mature into centrosomes during G2 and mitosis. Aberrations in centrosome number, frequently observed in cancer cells, contribute to chromosome missegregation and tumor progression. Mechanisms underlying centrosome amplification include the upregulation of PLK4 kinase, defects in cytokinesis, centrosome fragmentation, or de novo centriole formation.

This manuscript by Fahländer et al. explores the role of GPER1 and its interaction with protein kinase A (PKA), facilitated by the adaptor protein AKAP450 localized at centrosomes, in regulating centrosome number and integrity. The authors report that PKA phosphorylates Centrin-2, which may contribute to centrosome abnormalities observed in colon cancer cell lines. The study employs a range of approaches, including the use of antagonists, agonists, and GPER1-activating hormones, with centrosomes analyzed 48 hours after treatment initiation. Statistical analyses robustly support the findings. Centrosome amplification during the 48-hour period increases moderately, from 2% to 8%. The data convincingly demonstrate that GPER1 activation via PKA (mediated by cAMP) and its binding to AKAP450 induces the appearance of extra gamma-tubulin (at the pericentriolar material and within centrioles) and centrin-2 (components of SF11 and POC5 complexes) signals. Intriguingly, these signals do not always co-localize upon centrosome amplification, unlike the control cells. This observation suggests centriole fragmentation or protein accumulation, akin to phenomena seen with overexpression of CEP250 or centriolin. Furthermore, amplified centrin-2 signals frequently appear as "super spot structures," interpreted as signs of centriole enlargement. This conclusion, supported by conventional immunofluorescence imaging (Fig. 3c), would benefit from confirmation via electron microscopy (EM) analysis using conventional serial sections. Such analysis could also reveal additional aberrations in centriole structure following GPER1 activation.

The authors also demonstrate that centrin-2 phosphorylation at Ser170 is mediated by the GPER1-PKA pathway. In cells exhibiting additional centrin-2 signals, a significant fraction shows phosphorylation at Ser170, further linking this modification to centrosome abnormalities.

Conclusion and Recommendations

This manuscript offers important insights into centrosome regulation via the GPER1-PKA-AKAP450 pathway. While the findings are compelling, further validation using EM and improved ultrastructure expansion microscopy (u-ExM) to analyze additional centrin-2 and gamma-tubulin signals is essential. These additional analyses will strengthen the evidence and provide deeper structural understanding of the observed centrosome anomalies.

Overall, this study provides a significant contribution to the field and merits publication following major revision.

Major points

1. The authors use HCT116 and HCT-15 cells as models. How general are the findings of this manuscript?
2. The authors have to confirm centrosome amplification using electron microscopy and improved u-ExM. U-ExM in Fig. 2e is of insufficient quality.
3. AKAP450 is indicated as a 40 kDa protein in Fig. 2j. AKAP450 has a molecular weight of 450 kDa. Depletion efficiency is moderate. On p. 5 the authors state that AKAP450 knockdown is more efficient in HCT-15 cells. I cannot see this in Fig. 2j.
4. The P-centrin2 IP with PKA α in Fig. 6 is not so clear mainly because of the IgG light chain. The authors should repeat the IB using TrueBlot (specific to whole IgG).
5. What happens to centrosomes when a phospho-inhibitory centrin2-Ser170Ala is expressed in combination with GPER1 activation?

Minor points

1. The authors use AKAP450 and AKAP9. They should use AKAP450 throughout the manuscript.
2. The p-centrin2 signal in Fig. 5b is weak.
3. AKAP450 is not associated with centrosomes in mitosis. How is centrin2 phosphorylated in mitosis? AuroraA?

** See the Nature Portfolio author and referees' website at www.nature.com/authors for information about policies, services and author benefits

Communications Biology is committed to improving transparency in authorship. As part of our efforts in this direction, we are now requesting that all authors identified as 'corresponding author' create and link their Open Researcher and Contributor Identifier (ORCID) with their account on the Manuscript Tracking System prior to acceptance. ORCID helps the scientific community achieve unambiguous attribution of all scholarly contributions. You can create and link your ORCID from the home page of the Manuscript Tracking System by clicking on 'Modify my Springer Nature account' and following the instructions in the link below. Please also inform all co-authors that they can add their ORCIDs to their accounts and that they must do so prior to acceptance.

Version 1:

Decision Letter:

** Please ensure you delete the link to your author homepage in this email if you wish to forward it to your coauthors **

Dear Dr Stolz,

Your revised manuscript entitled "A GPER-PKA-Centrin axis regulates centrosome numbers and centriole integrity in colon cancer cells" has now been seen by 3 referees, whose comments are appended below. You will see from their comments below that while reviewers 1 and support a publication at this stage, reviewer 2 has some remaining concerns. We are interested in the possibility of publishing your study in Communications Biology, but would like to consider your response to these concerns in the form of a revised manuscript before we make a final decision on publication.

We therefore invite you to revise and resubmit your manuscript, taking into account the points raised.

1) Need for additional Centrin-KO clones: given that in your present manuscript you show that WT-centrin phenotypically complements the knock-out phenotype, we feel that this point has already been addressed in a satisfactory manner.

2) representative images for the quantification shown in Figure 4b: we feel that this is an excellent point that should be addressed with your current data

3) controls and quantification of immunoprecipitations: this point should be improved to strengthen these results and put the quantification on a more solid footing. While we appreciate that it is not always possible to analyze all the samples on the same blot, this point can be nevertheless improved.

Please highlight all changes in the manuscript text file.

We are committed to providing a fair and constructive peer-review process. Do not hesitate to contact us if you wish to discuss the revision in more detail or if there are specific requests from the reviewers that you believe are technically impossible or unlikely to yield a meaningful outcome.

At the same time, we ask that you ensure your manuscript complies with our editorial policies. Specifically:

For all graphs depicting a single point value (e.g., mean) with error bars, you must add individual data points or convert the graph to a boxplot or dot-plot to show data distribution.

It's mandatory to provide access to the numerical source data for graphs and charts either through a repository or by

providing the data in a Supplementary Data file (in excel format).

All blots/gels must be accompanied by size markers in every figure panel. Uncropped and unedited blot/gel images must be included as Supplementary Figure(s) in the Supplementary Information pdf.

Please ensure that you have complied with the data deposition policies at the Nature Portfolio, please see [here](http://www.nature.com/authors/policies/availability.html#data).

Please ensure that you have complied with our policies on research involving animals and humans, see [here](https://www.nature.com/commsbio/editorial-policies/ethics-and-biosecurity#studies-involving-animals-and-human-research-participants).

Please follow the ARRIVE guidelines for reporting animal experiments. Please fully complete an [ARRIVE checklist](https://arriveguidelines.org/sites/arrive/files/documents/Author%20Checklist%20-%20Full.pdf) including both the essential and recommended set of items (adding information to the manuscript where needed) and upload this with your revised manuscript.

Please also see [our revision checklist](https://www.nature.com/documents/CommsBio-file-checklist-revision.pdf) for guidance on formatting the manuscript and complying with our policies. A comprehensive guide to our formatting requirements for final submissions is also available for your reference [here](https://www.nature.com/documents/commsj-life-style-formatting-guide-accept.pdf).

Please use the following link to submit your revised manuscript, point-by-point response to the referees' comments (which should be in a separate document to the cover letter) and any additional files:

Link Redacted

When submitting the revised version of your manuscript, please pay close attention to our [Digital Image Integrity Guidelines](https://www.nature.com/commsbio/editorial-policies/image-integrity).

We would like to receive your revision within 10 weeks, but appreciate that every situation is unique. We look forward to receiving your revised manuscript when it is ready, and will not enforce a hard deadline on this revision.

Please do not hesitate to contact me if you have any questions or would like to discuss these revisions further. We look forward to seeing the revised manuscript and thank you for the opportunity to review your work.

Best regards,

Patrick Meraldi, PhD
Editorial Board Member
Communications Biology
orcid.org/0000-0001-9742-8756

Reviewers' comments:

Reviewer #1 (Remarks to the Author):

I want to congratulate the authors for their revised manuscript. They answered all my questions in great detail and significantly improved their manuscript.

Reviewer #2 (Remarks to the Author):

The finding that estrogen promotes centrosome amplification via PKA is intriguing, and the authors have made progress in elucidating the underlying mechanisms. The work addresses an important question that will be of interest to the field. However, I believe the manuscript still requires improvements before it is suitable for publication. Several experiments lack appropriate controls, and some figures would benefit from clearer presentation and more rigorous data representation.

1. One of the central questions of this study is whether PKA-mediated phosphorylation of Centrin drives centrosome amplification. Two of my main points focused on clarifying this aspect.

First, as suggested, the authors generated a Centrin knockout cell line. They show that PKA activation does not induce centrosome amplification in this knockout (Fig. 5b-c), which is an important finding and strongly supports the role of Centrin in centrosome amplification. This result would be even more compelling if confirmed in two independent knockout clones and the experiment includes a positive control (HCT116 cells).

2. Second, I had raised the question of whether the observed Centrin foci represent centriole fragments or independent foci. The authors addressed this by quantifying co-staining with other centriole markers (Fig. 4b). However, no representative images are shown, and the quantification alone could be misleading. I strongly recommend including representative images demonstrating that CEP135 and CP110 co-localize with the enlarged Centrin foci. For example, if there are three Centrin foci (two corresponding to centrioles and one enlarged), there should also be three corresponding CEP135 or CP110 foci in the same cell.

3. Third, I was concerned that the IP experiments shown in Fig. 7 lacked negative controls. The authors attempted to address this with an independent IP, but this does not suffice. Each IP experiment must include a negative control performed in parallel on the same blot. In my view, all IP experiments should be repeated with appropriate negative controls.

Additionally, I noticed that Centrin-pS170 signals are quantified and compared across independent blots (Fig. 7e-f). This is problematic, as many factors—including blotting conditions, exposure times, and developing methods—can influence signal intensity. Direct comparison between samples from different blots is therefore unreliable. For accurate comparisons, all relevant samples must be run on the same gel. For example, even a visual inspection shows strong differences in IgG intensities between the DMSO/BPA and E2/DES blots, indicating that differences in Centrin-pS170 cannot be confidently interpreted across blots.

Other points:

The PLA data are inconclusive without proper controls and quantification. Suggested controls include: a) antibody 1 only; b) antibody 2 only). Furthermore, the centrin knockout mutant should be included to demonstrate that the PLA signal is specific.

Reviewer #3 (Remarks to the Author):

The authors have addressed most of my suggestions from the first round of review. In particular, they now provide a more thorough characterization of the supernumerary centrioles/centrosomes. They also show some EM analysis of centrioles in Fig. 4 although the quality is not too high (no serial sections). It remains unclear how frequent the defective centrioles in Fig. 4e are. The authors demonstrate that Centrin-2 is important to mediate GPER1-PAK activity on amplified centrioles and show that Centrin-2 is phosphorylated by PKA at serine-170 and this may be the basis for the enlarged Centrin signals associated with centrioles. Based on these data, they can suggest the model that is outlined in Fig. 8.

Overall, the manuscript is sufficiently improved to justify publication in Communications Biology.

** See the Nature Portfolio author and referees' website at www.nature.com/authors for information about policies, services and author benefits

Communications Biology is committed to improving transparency in authorship. As part of our efforts in this direction, we are now requesting that all authors identified as 'corresponding author' create and link their Open Researcher and Contributor Identifier (ORCID) with their account on the Manuscript Tracking System prior to acceptance. ORCID helps the scientific community achieve unambiguous attribution of all scholarly contributions. You can create and link your ORCID from the home page of the Manuscript Tracking System by clicking on 'Modify my Springer Nature account' and following the instructions in the link below. Please also inform all co-authors that they can add their ORCIDs to their accounts and that they must do so prior to acceptance.

Version 2:

Decision Letter:

** Please ensure you delete the link to your author homepage in this email if you wish to forward it to your coauthors **

Dear Dr Stolz,

Your manuscript entitled "A GPER-PKA-Centrin axis regulates centrosome numbers and centriole integrity in colon cancer cells" has now been seen again by our editors, who found your rebuttal and revisions satisfactory. In light of their advice I am delighted to say that we are happy, in principle, to publish a suitably revised version in Communications Biology.

We ask that you edit your manuscript to comply with our format requirements and to maximise the accessibility and therefore the impact of your work.

* Please see the attached document for editorial requests for the final version (.docx file). Please ensure a completed version of this file is uploaded as a Related Manuscript with your final submission.

* Please review our [final submission file checklist](https://www.nature.com/documents/commsj-file-checklist.pdf) to ensure all necessary files are present with your final submission and to avoid delays in accepting your manuscript. For your reference, a style and formatting guide is available [here](https://www.nature.com/documents/commsj-life-style-formatting-guide-accept.pdf) and includes all of our style requirements.

It is important that you pay careful attention to the requests in these documents to avoid a delay in formal acceptance of the article.

Open access

Communications Biology is a fully open access journal. Articles are made freely accessible on publication. For further information about article processing charges, open access funding, and advice and support from Nature Research, please visit <https://www.nature.com/commsbio/open-access>

Please use the following link to upload your revised files:

Link Redacted

We hope to hear from you within two weeks. If you expect the process to take longer than one month, please let us know.

Congratulations on an excellent paper!

Best regards,

Kaliya Georgieva, PhD
Associate Editor
Communications Biology
<https://orcid.org/0009-0006-2251-2578>

PS: At acceptance, the corresponding author will be provided with instructions for completing the license on behalf of all authors. This grants us the necessary permissions to publish your paper. Additionally, you will be asked to declare that all required third party permissions have been obtained, and to provide billing information in order to pay the article-processing charge (APC).

** See the Nature Portfolio author and referees' website at www.nature.com/authors for information about policies, services and author benefits

Point-by-point response letter to the referees

Fahrländer et al., manuscript COMMSBIO-24-7871-T

A GPER-PKA-Centrin axis regulates centrosome numbers and centriole integrity in colon cancer cells

We cordially thank all reviewers for an excellent review of our manuscript and their constructive feedback, which highlights the significance of our work. We are confident that the revised version, which now includes 8 Figures with a total of 26 new panels, significantly improves the quality of the manuscript. Please find below changes of the revised figures, the detailed responses point-by-point along with the corresponding changes in the manuscript.

Changes of revised figures:

Revised Figure 1:

Revised Fig. 1 is identical to original Fig. 1.

Revised Figure 2:

New sub-panels include: Revised Fig. 2g, 2h, 2i

Revised Fig. 2a-f is identical to original Fig. 2a-f

Revised Supplemental Fig. 2e is identical to original Fig. 2g, h

Revised Supplemental Fig. 2c is identical to original Fig. 2i, j

Revised Figure 3:

Revised Fig. 3a is identical to original Fig. 3a, b

Revised Fig. 3b is identical to original Fig. 3c

Revised Fig. 3c is identical to original Fig. 3d

Revised Fig. 3d is identical to original Fig. 3f

Revised Fig. 3e, f is identical to original Fig. 3g

Revised Fig. 3g is identical to original Fig. 3k, l

Revised Figure 4:

New sub-panels include: Revised Fig. 4b, 4c, 4e, 4f

Revised Fig. 4a is identical to original Supplemental Fig. 3f, g

Revised Fig. 4d is identical to original Supplemental Fig. 3c

Revised Figure 5:

New sub-panels include: Revised Fig. 5b, 5c

Revised Fig. 5a is identical to original Fig. 4a

Revised Fig. 5d is identical to original Fig. 4b

Revised Fig. 5e is identical to original Fig. 4c

Revised Fig. 5f is identical to original Fig. 4d

Revised Fig. 5g is identical to original Fig. 4e

Revised Fig. 5h is identical to original Fig. 4f

Revised Fig. 5i is identical to original Fig. 4g
Revised Fig. 5j is identical to original Fig. 4h

Revised Figure 6:

New sub-panels include: Revised Fig. 6i, 6j, 6k, 6l
Revised Fig. 6a, b is identical to original Fig. 5a, b
Revised Fig. 6c-h is identical to original Fig. 5c-h

Revised Figure 7:

New sub-panels include: Revised Fig. 7g
Revised Fig. 7a is identical to original Fig. 6a
Revised Fig. 7b is identical to original Fig. 6b
Revised Fig. 7c is identical to original Fig. 6c
Revised Fig. 7d is identical to original Fig. 6d
Revised Fig. 7e is identical to original Fig. 6e
Revised Fig. 7f is identical to original Fig. 6f
Revised Fig. 7h is identical to original Fig. 6g
Revised Fig. 7i is identical to original Fig. 6h

Revised Figure 8:

Revised Fig. 8 is identical to original Fig. 7 with adaption to our new findings that centrioles are reduced in length and harbor enlarged Centrin foci.

Revised Supplemental Figure 1:

New sub-panels include: Revised Fig. S1a
Revised Fig. S1b is identical to original Fig. S1a, b
Revised Fig. S1c is identical to original Fig. S1c

Revised Supplemental Figure 2:

New sub-panels include: Revised Fig. S2d, S2f, S2g
Revised Fig. S2a, b is identical to original Fig. S2a, b
Revised Fig. S2c is adapted from original Fig. 2i, j.
Revised Fig. S2e is identical to original Fig. 2g, h

Revised Supplemental Figure 3:

Revised Fig. S3a is identical to original Fig. S3a
Revised Fig. S3b is identical to original Fig. S3b
Revised Fig. S3c is identical to original Fig. S3d
Revised Fig. S3d is identical to original Fig. S3e
Revised Fig. S3e is identical to original Fig. 3j

Revised Supplemental Figure 4:

New sub-panels include: Revised Fig. S4b, c
Revised Fig. S4a is identical to original Fig. 3e

Revised Supplemental Figure 5:

New sub-panels include: Revised Fig. S5a, b
Revised Fig. S5c is identical to original Fig. S4a
Revised Fig. S5d is identical to original Fig. S4a

Revised Supplemental Figure 6:

New sub-panels include: Revised Fig. S6c, d, e
Revised Fig. S6a is identical to original Fig. S5a
Revised Fig. S6b is identical to original Fig. S5b, c

Revised Supplemental Figure 7:

New sub-panels include: Revised Fig. S7c
Revised Fig. S7a is identical to original Fig. S6a
Revised Fig. S7b is identical to original Fig. S6b

Point-by-point responses to the referees:

Reviewer #1:

Remarks to the Author:

In the present manuscript Fahrländer and colleagues describe a role of the G protein-coupled estrogen receptor GPER1 in regulating numerical and structural centrosome aberrations. They demonstrate that GPER1 activation by estrogens or environmental estrogenic compounds can lead to centrosome amplification (CA) in colorectal cancer cell lines. They show that CA is mediated by elevated cAMP levels and protein kinase a (PKA) activation downstream of GPER1 ligation. Half of the cells harbouring extra centrosomes show enlarged and displaced Centrin-2 foci indicating alterations in centriole structure. They further seek to establish a causal link between centrosomal PKA recruitment, Centrin-2 phosphorylation and lack of centriole integrity. Collectively, the authors provide evidence for a GPER-PKA-Centrin signalling axis regulating centriole numbers and structure in colorectal cancer cells.

Major comments:

The manuscript is well written and easy to follow and presents novel insights into the causes of centrosome amplification in cancer cells. The experiments are technically well performed and well-presented. However, they do not fully support the authors conclusions. Moreover, the overall degree of CA in the chosen cell lines is very low with less than 10% of cells showing extra centrosomes. This raises concerns regarding the biological significance and whether their findings would also apply to cell lines that show a high proportion of cells with CA.

We appreciate the reviewer's valuable comments and performed additional experiments, which support the conclusions presented in the revised manuscript.

We agree that the overall frequency of centrosome amplification (CA) in our model is relatively low (<10%, see revised **Figures 1 and 2**). However, this level is consistent with previous studies reporting ~10-15% of CA following treatment with low concentrations of estrogen (Kim et al., 2009; Tarapore et al., 2014; Ho et al., 2017; Bühler et al., 2023). Importantly, even strong inducers of CA, such as PLK4 overexpression, typically lead to a modest increase in CA levels (e.g., Bühler et al., 2023; Ganem et al., 2009), with higher levels (>85–95%) being associated with p53-dependent proliferation arrest and reduced cell viability (Holland et al., 2012). Moreover, even long-term estrogen treatment over 30 cell generations causes a stable CA level below 10% (Bühler et al., 2023). This suggests that there may be a threshold of CA above which cells experience a growth disadvantage, thus limiting the accumulation of cells with multiple centrosomes.

The use of nanomolar hormone concentrations is also relevant in this context. Higher (micromolar) doses have been shown to disrupt microtubule dynamics and arrest cells in mitosis - effects that resemble spindle poisons (Metzler et al., 1995; Pfeiffer et al., 1997; Jurasek et al., 2018). In contrast, the nanomolar doses used in our study did not induce mitotic arrest or impair proliferation (see new **Figure S1a** and Bühler et al., 2023).

Importantly, we have previously shown that low concentrations of (xeno)estrogens - as used in our current study - induce a low level of supernumerary centrosomes, which in turn, cause the generation of lagging chromosomes, whole chromosomal instability and aneuploidy in colon (cancer) cell lines (Bühler et al., 2023). We therefore propose that while higher levels of CA may compromise cell survival, low-level CA is tolerated and cause characteristic cancer-prone lesions.

[...] whether their findings would also apply to cell lines that show a high proportion of cells with CA.

It is an excellent suggestion to examine whether other cell lines with a high level of CA also exhibit structurally defective centriole-like structures characterized by enlarged and phosphorylated Centrin foci. Ghadimi et al. (2000) identified several colorectal cancer cell lines with a high proportion of cells displaying CA, such as Caco-2 (35%), HT-29 (25%), and SW837 (50%).

However, our findings indicate that centrosome amplification alone (e.g., by overexpression of PLK4) is not sufficient to induce Centrin enlargement (see revised **Figure 3g**). Instead, increased activity of GPER1 appears to be required to trigger this structural phenotype. Therefore, it would be important to assess endogenous GPER1 expression in these cell lines and to further investigate the potential role of the GPER1-PKA-Centrin axis. These represent exciting directions for future research.

Specific comments:

1. The authors convincingly show that PKA activation is involved in CA in colorectal cancer cell lines. Yet, the conclusion that PKA is recruited to centrosomes via AKAP-450 is not well supported by the data presented (reduced CA after AKAP-450 knock-down). The authors should show localization of PKA in the presence or absence of GPER1

activation.

We appreciate the reviewer's insightful comment and have addressed it by investigating PKA localization at interphase centrosomes both in the presence or absence of GPER1 activation and in AKAP450-deficient cells, generated by CRISPR-Cas9 technology. Using immunofluorescence microscopy, we confirmed PKA localization at centrosomes in DMSO-treated HCT116 cells (see new **Figures 2g and h**). Upon GPER1 activation by BPA or G-1, centrosomal PKA signal intensity increased significantly, consistent with recruitment of PKA to the centrosome (see new **Figure 2g**). In AKAP450 knockout cells, PKA was absent from centrosomes, supporting a model in which AKAP450 mediates GPER1-dependent PKA recruitment. The data are shown in the new **Figures 2h and S2f and g**. Accordingly, we updated the Results section (**line 216 onwards**) to state:

“PKA localizes to diverse cellular compartments by binding to different A kinase-anchoring proteins (AKAPs)³³. AKAP450 localizes with PKA at the centrosome, suggesting a potential role as a docking platform for PKA^{27, 34, 35, 36, 37}. By using immunofluorescence microscopy to analyze intracellular PKA localization, we confirmed not only the centrosomal localization of PKA in DMSO-treated HCT116 cells (**Fig. 2g, h**), but also observed a marked increase in centrosomal PKA levels upon GPER1 activation with BPA or G-1 (**Fig. 2g**). [...] Furthermore, we showed that its recruitment to the centrosome is dependent on AKAP450, as PKA is absent from centrosomes in AKAP450 knockout cells (**Fig. 2h and Fig. S2g**).”

We hope these additions comprehensively address the reviewer's concerns.

2. Centriole integrity was determined by immunofluorescence using antibodies against Centrin-2. The authors detect pronounced Centrin-2 staining after GPER1 activation referred to as 'enlarged centrioles'. The pictures shown do not allow to precisely measure centriole length (even after expansion microscopy; what is the degree of expansion?). To determine centriole size and conclude about structural centriole defects, the authors should present higher magnification images and/or carry out transmission electron microscopy (TEM).

We thank the reviewer for this important point and fully agree that Centrin foci size alone cannot be used to infer centriole length or size. To address this, we performed transmission electron microscopy (TEM) with GPER1 activated colon cancer cells. Although technically challenging within the limited revision timeframe, we successfully performed TEM analysis to examine centriole ultrastructure following GPER1 activation.

Our new data include quantitative measurements of centriole size in HCT116 cells treated with either DMSO or BPA (see new **Figure 4e, f** and new **Figure S4b**). In DMSO-treated controls, centrioles displayed their typical barrel-shaped morphology composed of triplet microtubules, as well as (sub)distal appendages, in agreement with previous reports (Marteil et al., 2018, LeGuennec et al., 2021, Guichard P et al., 2023). Measured dimensions - diameter (~220 nm), length (~409 nm), and width (~198 nm) - closely matched published values (revised **Figure 4f**, and Marteil et al., 2018, LeGuennec et al., 2020).

By contrast, ~90% of BPA-treated cells (n > 70) showed structural centriole abnormalities, including loosened microtubule walls, detachment of microtubule bundles, asymmetric protrusions, and irregular appendage-like extensions (see new **Figure 4e**, panels 5–9). Quantification revealed a significant reduction in centriole length (mean ~322 nm; minimum

114 nm), while diameter and width remained unchanged (see new **Figure 4f**). This was somewhat unexpected, as we identified *enlarged* Centrin-foci upon GPER1 activation (see revised **Figure 3**). However, together with our new immunofluorescence analyses, which demonstrate that GPER1-activation leads to an incomplete composition of centriole-specific proteins at those centriole-like structures containing enlarged Centrin foci (new **Figures 4b and c**), these data suggest that the observed enlarged Centrin signals do not correspond to intact centrioles, but rather reflect structurally aberrant or immature centriole-like assemblies.

This supports our revised terminology, referring to these structures as “enlarged Centrin-2,” “enlarged Centrin-2 foci” or “enlarged Centrin-labeled foci” rather than using less precise terms such as “enlarged Centrin-labeled centrioles” or “enlarged centrioles”, and highlights the importance of using ultrastructural and molecular markers in parallel to define centriole integrity.

Accordingly, we updated the Results (line 389 onwards) and Discussion section (line 783 onwards). We kindly refer the reviewer to our responses to reviewer reviewer 2’s comments #1, #12, and #17 for further details regarding the TEM analyses.

[...] what is the degree of expansion?

Thank you for pointing this out. The degree of expansion used in our experiments is 4.5x (see revised **Figure 4d**) and 3.5x (see revised **Figure S4a**), as indicated in the respective figure legends. For clarity, we have now also included this information in the main text (see line 384).

3. In the present study CA is induced only after GPER1 activation (in the presence of estrogens). They refer several times to the work from the Bettencourt-Dias lab where they catalogue centriole numbers in human cancer cell lines and demonstrate CA in HCT116 cells in the absence of any stimulus (Marteil et al). Why is CA only induced after GPER1 activation in one study but not the other?

The reviewer raises a valid and important point. The study by Marteil et al. (2018) reports that approximately 7% of untreated HCT116 cells contain more than four centrioles, and 10.5% exhibit overly long centrioles. In line with this observation, we also detect a baseline level of centriole amplification (CA) and enlarged Centrin foci in DMSO-treated HCT116 cells in our experiments (see revised **Figures 1, 2, 3d–f**, and Bühler et al., 2023). Specifically, we observe a background level of CA of 2–3%, which increases to approximately 8% upon GPER1 activation. Similarly, the proportion of cells with enlarged Centrin foci rises from 25% to 56% following GPER1 stimulation (see revised **Figure 3d–f**). In contrast - and consistent with Marteil et al. - we do not observe similar background levels of CA or Centrin enlargement in DMSO-treated HCT-15 cells.

Thus, CA and Centrin enlargement are indeed present in DMSO-treated HCT116 control cells, as mentioned in the original (see line 253 onwards) and in the revised manuscript (see line 311 onwards) but importantly, both phenotypes are increased by GPER1 activation. We agree with the reviewer, however, that the baseline CA levels we detected are lower than those reported by Marteil et al. This discrepancy may stem from methodological differences in the quantification and classification of centriolar abnormalities across studies.

4. Why does PLK4 overexpression (OE) not result in a larger number of cells with extra centrioles (Figure 3k, l)? The authors should demonstrate efficient PLK4 OE resulting in CA in their cell lines.

We apologize for the misunderstanding. PLK4 overexpression does in fact lead to centrosome amplification, as shown in the original **Figures 3k, l** (revised **Figure 3g**) for both HCT116 and HCT-15 cells. We observe an increase in CA from approximately 3% to 10% in HCT116 cells, and from about 1% to 6% in HCT-15 cells following PLK4 overexpression. Furthermore, immunofluorescence analysis in the original **Figure 3j** (revised **Figure S3e**) shows the characteristic rosette-like arrangement of Centrin (shown in red), which is a hallmark of PLK4 overexpression (Habedanck et al., 2005). Efficient PLK4 overexpression is shown in the western blot shown in **Figure S3e** (revised **Figure S3d**).

5. The authors claim that PKA activation is responsible for CA in colorectal cancer cells by phosphorylating Centrin-2. While the generated phospho-specific Centrin-2 antibody seems to work very well, only a minor reduction in phospho-Centrin-2 was detected after PKA inhibition (Figure S4a,b). This strongly suggests, that other kinases are involved in phosphorylating Centrin-2 at Serine 170.

It is true that PKA inhibition leads to only a modest reduction in Centrin-2 phosphorylation overall (see revised **Figure S5c and d**). However, when examining interphase cells specifically, we observe a marked decrease in phospho-Centrin-2 following GPER1 stimulation upon PKA inhibition - from approximately down to 5-10% (see new **Figure S6d**). These data suggest that the GPER1-PKA-Centrin axis plays a dominant role during interphase, which is also the cellular context where we observe Centrin enlargement and centrosome amplification. However, the reviewer is absolutely correct in pointing out that Centrin-2 is likely phosphorylated by multiple kinases. In addition to PKA, Aurora A has been shown to phosphorylate Centrin-2 at Serine 170, contributing to Centrin stability, centrosome amplification, and centriole separation (Lukasiewicz et al., 2011; Lutz et al., 2001; see **line 79ff, line 822ff**).

6. Moreover, data supporting the model that phosphorylation of Centrin-2 causes structural centriole aberrations is weak. The authors should generate phospho-mutants of Centrin-2 to demonstrate its impact for centriole structure.

This is an excellent suggestion, which we have directly implemented in the revised manuscript. We generated HCT116 Centrin-2 knockout cells and re-expressed either wild-type Centrin-2 (CTN2 WT) or a PKA-non-phosphorylatable Centrin-2 mutant (CTN2 S170A), and assessed whether centrosome amplification, Centrin enlargement, or centriole separation could still be induced upon GPER1 activation (see new **Figures 6j-l** and **Figure S6e**). Our results show that the phospho-mutant CTN2 S170A effectively rescues both centrosome amplification and the displacement of enlarged Centrin foci, but not the Centrin enlargement itself.

These findings suggest that phosphorylation of Centrin-2 at Ser170 is critical for the numerical and spatial centrosome defects, while structural Centrin enlargement might be regulated through a different mechanism.

Reviewer #2:

Comments to the Author:

In this manuscript, Fahrlechner and colleagues propose a mechanism by which estrogen-mediated signaling regulates centrosome number in colorectal cancer-derived cell lines. They demonstrate that activation of the estrogen receptor GPER1 triggers PKA-dependent phosphorylation of centrin at the centrosomes, which, in turn, leads to the formation of additional centrin foci. The authors conclude that these extra centrin foci are responsible for centrosome amplification. The finding that estrogen promotes centrosome amplification via PKA is an important finding with potential implications. Understanding the underlying molecular mechanism is a valuable contribution to the field. However, the claims of the paper are not sufficiently supported by the presented data. The initial experiments, which explore the role of GPER1 and PKA in centrosome amplification through chemical inhibition and activation, are well designed and appropriately controlled (Fig. 1–2). The authors also show that PKA activation leads to ectopic centrin foci. Unfortunately, the authors do not provide any conclusive evidence that ectopic centrin foci are related to centrosome amplification. Thus, I think it remains unclear how estrogen-dependent PKA activity leads to centrosome amplification. I am worried that the authors claim throughout the manuscript that PKA enlarges centrioles based on their observation of ectopic centrin foci. This is a dangerous assumption which could be wrong. The existence of centrin foci that are independent of centrioles has been observed (PMID: 33983387). Thus, I think that at this point it is unclear how PKA activation leads to centrosome amplification. However, I think that this point could be addressed by a few additional experiments outlined below.

We thank the reviewer for the thoughtful and constructive comments. We appreciate the concerns raised regarding the interpretation of Centrin-positive foci and their relationship to PKA and true centrosome amplification. As suggested, we have carefully addressed this point by performing additional experiments and revising the relevant sections of the manuscript accordingly. A detailed response to this important issue is provided in our point-by-point reply below.

Major issues:

1. Throughout the manuscript, the authors consider centrin foci as centrioles. However, it is very likely that most centrin foci don't belong to centrioles. The authors even provide strong evidence that the centrin foci are not centrioles in figures 3a-b, where they show that most centrin foci do not colocalize with gamma-tubulin. Gamma-tubulin was used in the first place to count the amplified centrosomes. They also do not show the co-localization of any other centriole marker with the centrin foci. Thus, there is no evidence that support the claim that all centrin foci are centrioles. The authors should pay attention to a recent publication, which describes ectopic centrin foci in TRIM37-deleted cells (PMID: 33983387). Using CLEM this paper showed that ectopic centrin foci do not reassemble centrioles. They also found a small

increase in centriole number. However, this effect was independent of centrin and was explained by mitotic defects that were caused by ectopic centrobins foci, which were independent of centrin. In the light of this work, the authors must rigorously test the relationship between centrin foci and centrioles. They also have to address the possibility that other factors such as ectopic Centrobins foci and/or mitotic defects could lead to centriole amplification through the described PKA-dependent mechanism.

12. From the data shown in Figure S3f-g, I'm not convinced that all extra centrin foci are centrioles. There is no quantification for S3f-g. CP110 and CEP135 are co-stained with gamma-tubulin, but not with Centrin. In the main figures, it is shown that Centrin foci do not always co-localize with gamma tubulin. Thus, they are most likely not centrioles. The authors have to show co-localization of centrin with other centriole markers to claim that all centrin foci are centrioles.

We thank the reviewer for this important and insightful comment and for drawing our attention to the recent publication on ectopic Centrin foci in TRIM37-deleted cells (PMID: 33983387). We fully agree that it is crucial to rigorously evaluate whether the Centrin foci observed in our experiments correspond to bona fide centrioles, and to consider the possibility that they may instead represent Centrin-positive structures that are not true centrioles.

Since the concerns raised in comments #1 and #12 overlap, we address them here jointly:

a. The reviewer is correct that in **Figures 1 and 2** we used γ -tubulin staining as the primary marker to quantify centrosome amplification. In our previous study (Bühler et al., 2023), we confirmed bona fide centrosome amplification in HCT116 cells upon GPER1 activation by co-immunofluorescence staining of the PCM marker γ -tubulin together with two centriole-specific proteins, Cep135 and CP110. These data confirmed γ -tubulin as a reliable marker for detecting supernumerary centrosomes under the experimental conditions used here.

Building on these findings, and motivated by reports linking PKA activity to changes in centriole number and size in other cancers (Lutz et al., 2001; Lingle et al., 1998) as well as our own observation that PKA modulates centrosome number downstream of GPER1 (see revised **Figure 2**), we investigated centriole integrity more closely in GPER1-activated CRC cells using Centrin as a well-established centriole marker. We detected enlarged Centrin foci in cells with amplified centrosomes (see revised **Figure 3**).

b. To rigorously test whether these enlarged Centrin foci represent genuine centrioles as suggested by the reviewer, we performed co-immunostaining with additional established centriole markers, including CP110, Cep135, and Centrobins (Martel et al., 2018; Bühler et al., 2023; Wang et al., 2019; Zou et al., 2005), and analyzed their co-localization with Centrin (see revised **Figure 4a** and new **Figures 4b and 4c**).

CP110 and Cep135 signals in control cells exhibited expected morphology and size, indicating intact centriole components. More than 50% of BPA-treated and over 30% of G-1-treated cells showed co-localization of enlarged Centrin foci with CP110. Co-localization with Cep135 was much less frequent (<5% in BPA-treated cells) and absent in G-1-treated cells, suggesting that GPER1 activation may induce formation of centriole-

like structures lacking full canonical protein composition, especially Cep135, in the presence of enlarged Centrin.

- c. We also investigated the potential role of ectopic Centrobin foci, as described in TRIM37-deficient cells, by co-staining Centrobin with Centrin. Less than 3% of cells with amplified centrosomes and enlarged Centrin foci showed Centrobin signal (new **Figure 4c**), indicating that ectopic Centrobin is unlikely to contribute significantly in our model.
- d. Importantly, our new transmission electron microscopy (TEM) analyses provide direct ultrastructural evidence of centriole abnormalities upon GPER1 activation, including a significant reduction in centriole length (new **Figures 4e and 4f**; see also responses to comment #17 and reviewer 1's comment #2). This reduction in centriole size was somewhat unexpected, as it does not correspond with the presence of enlarged Centrin foci observed upon GPER1 activation (see revised **Figure 3**). However, together with our new immunofluorescence analyses showing that GPER1 activation leads to an incomplete composition of centriole-specific proteins at these centriole-like structures containing enlarged Centrin foci (new **Figures 4b and 4c**), these data suggest that the enlarged Centrin signals do not represent intact centrioles, but rather structurally aberrant or immature centriole-like assemblies.

Taken together, these data support a model where GPER1 signaling leads to the formation of aberrant centriole-like structures characterized by incomplete protein composition and enlarged Centrin foci. These aberrant structures appear largely independent of ectopic Centrobin foci and may represent partially assembled or structurally compromised centrioles rather than fully functional centrioles.

In light of these results and the reviewer's concerns, we have revised the manuscript to apply more cautious terminology throughout, referring to these Centrin-positive structures as "enlarged Centrin foci" or "centriole-like structures" rather than unequivocally as "centrioles." Accordingly, we updated the Results (line 389 onwards) and Discussion section (line 783 onwards).

2. The authors provide evidence that estrogen signaling leads to PKA-dependent centrosome amplification. They use gamma-tubulin as marker to count centrosomes. However, the connection between centrin and extra gamma-tubulin foci remains unclear as discussed above. To conclude that centrin induces centrosome amplification, the authors must show that centrosome amplification (gamma tubulin staining) upon PKA activation does not happen in centrin-deleted cells. Centrin is not essential and can be deleted in HCT116 cells using standard CRISPR-Cas9 gene editing. This is a straightforward experiment, which is necessary to determine the role of centrin in the described mechanism.

We appreciate the reviewer's insightful suggestion, which significantly enhances the mechanistic understanding of centrosome amplification in our study. In response, we generated Centrin-2 knockout HCT116 cell clones using CRISPR-Cas9 and assessed centrosome amplification upon GPER1-PKA activation by immunofluorescence microscopy (see new **Figure 5c** and new **Figure S5b**). Successful knockout of Centrin-2 was confirmed

via Western blot (new **Figure S5a**) and loss of Centrin staining (new **Figure 5b**). Notably, GPER1-PKA-induced centrosome amplification, detected by γ -tubulin staining, was abolished in Centrin-2-deficient cells, returning to baseline levels (see new **Figure 5c**, and new **Figure S5b**). These results establish that Centrin-2 is functionally required for the formation of supernumerary centrosomes downstream of the GPER1-PKA signaling axis (see line 465 onwards).

To further dissect the role of Centrin-2 and specifically assess the importance of its phosphorylation by PKA, we re-expressed either wild-type human Centrin (WT) or a non-phosphorylatable Centrin mutant (S170A) in the Centrin-2 knockout cells (see new **Figure S5a**). Protein expression and phosphorylation status were confirmed by Western blot and immunofluorescence microscopy, respectively (see new **Figure 6j**, and new **Figure S6c**). While wild-type Centrin restored BPA-induced centrosome amplification, the S170A mutant failed to do so (see new **Figure 6l**). Interestingly, the frequency of enlarged Centrin foci was comparable between WT and mutant-expressing cells (new **Figure 6k**), suggesting that phosphorylation at S170 is specifically required for numerical centrosome amplification, but not for Centrin foci enlargement per se. Moreover, Centrin displacement, which is a hallmark phenotype observed in GPER1-activated cells (see revised **Figure 3c**), was markedly reduced in cells expressing the S170A mutant (**Figure S6e**). These findings are consistent with earlier work by Lutz et al. (2001), which demonstrated that PKA activation promotes centriole separation in interphase HeLa cells via centrosomal phosphorylation.

3. The authors claim that Centrin associates with PKA using immunoprecipitation experiments. However, all IP experiments were performed without negative control (IgG control, empty beads). Therefore, these experiments are inconclusive and do not provide any evidence that Centrin associates with PKA.

We thank the reviewer for this important comment. We fully agree that appropriate IgG controls are essential to demonstrate the specificity of the observed interaction between Centrin and PKA. In response, we have refined our experimental approach and now include IgG control immunoprecipitations as suggested by the reviewer (see new **Figure S7c**). These new experiments confirm the specificity of phospho-Centrin binding to PKA-C α in both HCT116 and HCT-15 interphase cells upon estrogenic stimulation (see line 661-667).

Other Points:

1. The authors quantify throughout the paper centrosome amplification as shown in figure 1a. They classify cells with 2 centrosomes as "Not amplified". This represents an inaccuracy, which should be avoided or explained. In G1-phase just one centrosome should be present. Two centrosomes in G1-phase indicate a prior defect, possibly centrosome amplification or cell division failure. Since there is no discrimination between G1 and S/G2-phase, there is no way to determine whether cells with 2 centrosomes have the normal number of centrosomes.

We thank the reviewer for this important and thoughtful comment. The reviewer is correct that cells in G1 typically possess a single centrosome, which is composed of a pair of centrioles. However, as shown by Wang et al. (2011), centrioles of G1 phase-cells were modified during the preceding mitosis in a Plk1-dependent manner, enabling them to independently recruit pericentriolar material (PCM) in the subsequent G1 phase. Since γ -tubulin is a core component of the PCM, this can result in two distinct γ -tubulin-positive foci, which may appear as two centrosomes, even during G1. Additionally, Piel et al. (2001) demonstrated that daughter centrioles are highly motile and can spatially separate within the cytoplasm, each forming a distinct γ -tubulin-positive focus that contains a single centriole. Therefore, the presence of two γ -tubulin signals in G1 does not unequivocally indicate centrosome amplification. However, since our analysis (particularly Figures 1 and 2) relied solely on γ -tubulin staining without co-labeling for centrioles (e.g., Centrin) or cell cycle markers (e.g., Cyclins), we agree with the reviewer that it is challenging to distinguish G1 from S/G2-phase cells based on centrosome number alone.

To address this uncertainty and avoid overinterpretation, we applied a conservative classification strategy: only cells with more than two centrosomes (i.e., >2 γ -tubulin foci) were scored as “amplified,” in line with established criteria (Bühler et al., 2023; Wang et al., 2019). We acknowledge that this may underestimate early amplification events (e.g., in G1), including cases resulting from cytokinesis failure or premature centriole duplication. Nevertheless, our approach minimizes false positives and ensures consistency across conditions.

We have now clarified this limitation in the revised Results section (see line 130 onwards).

2. Cells are exposed to NF449 and estrogens for 48h prior to quantification. This seems to be quite long considering that the cell division time for HCT116 cells is less than 20h. The authors should address the possibility that mitotic failure rather than centrosome amplification leads to the increase in centrosome number. One possibility is to show that centrosome amplification happens in cells during interphase.

We thank the reviewer for this insightful comment. We agree that a 48-hour treatment window allows for multiple cell divisions in HCT116 cells, whose doubling time is less than 20 hours. We chose this time point to ensure that centrosome amplification (CA), which depends on at least one completed cell cycle, can be adequately captured. Notably, the frequency of CA after 48 h of treatment remains relatively low (<10% upon GPER1/PKA activation, see revised **Figures 1 and 2**), consistent with previous reports using similar time frames and estrogen concentrations (Ho et al., 2017; Kim et al., 2009; Tarapore et al., 2014).

Importantly, we recognize the reviewer’s concern that elevated centrosome numbers could reflect mitotic failure or cytokinesis defects rather than bona fide centrosome overduplication. In our previous publication (Bühler et al., 2023) and within this manuscript (see **Figure S1a**), we addressed this possibility by showing that (xeno)estrogen treatment did not affect cell cycle distribution or cell proliferation, but leads to elevated centrosomal levels of Sas-6 during S phase. Since Sas-6 is a key component of the cartwheel structure required for initiating centriole duplication (Arquint & Nigg, 2016), this suggests that parental

centrioles may initiate the formation of multiple daughter centrioles within a single cycle, supporting genuine overduplication as a mechanism. Furthermore, and as the reviewer rightly pointed out, all centrosome counts in the current study are based on interphase cells (see revised **Figures 1a and b**), as determined by nuclear morphology. Together, this makes mitotic failure an unlikely cause of the observed amplification.

3. Does NF449 treatment impair cell cycle progression? If yes, then the observed effect on centrosome number could be independent of centrosome amplification per se.

We thank the reviewer for raising this important point. To directly address the possibility that NF449 might impair cell cycle progression, and thereby affect centrosome numbers independently of centrosome amplification, we performed additional cell proliferation assays in HCT116 and HCT-15 colorectal cancer cells. To do so, cells were treated with NF449 in the presence or absence of GPER1 activation.

Our results showed that NF449 treatment does not impair cell proliferation in either cell line (see new **Figure S1a**), indicating that the observed suppression of centrosome amplification upon G α s inhibition occurs independently of effects on cell cycle progression or proliferation.

4. In Figure S1C-D the significance of the difference between H₂O+DMSO and DMSO+G1 is shown. However, the relevant difference, for which the significance is missing, is between DMSO+G1 and SQ22,536+G1. These measurements have to be significantly different to support the related claims made in the manuscript.

In the **original Figure S1c-d**, statistical significance was shown between the H₂O + DMSO and DMSO + G-1 conditions (other conditions were not significantly different from the H₂O + DMSO control, as shown in Table S1). These comparisons demonstrate that GPER1 activation by G-1 effectively increased intracellular cAMP levels.

However, we agree with the reviewer that the comparison between DMSO + G-1 and SQ22,536 + G-1 is particularly relevant, as it would confirm that adenylyl cyclase inhibition by SQ22,536 successfully impaired cAMP production. Demonstrating a significant difference between these conditions is indeed important to substantiate the related claims made in the manuscript.

Accordingly, we have revised **Figure S1c (originally Figure S1c-d)** to now include the following p-values for both HCT116 and HCT-15 cell lines:

DMSO + G-1 vs. NF449 + G-1

DMSO + G-1 vs. SQ22,536 + G-1

5. The authors claim that PKA phosphorylates proteins that are involved in centrosome function. However, they never mention which proteins are supposed to be phosphorylated by PKA. I think this is important to clarify.

We thank the reviewer for this valuable comment. In response, we have revised the manuscript and now provide specific examples of centrosome-associated proteins that are phosphorylated by PKA. As noted in the revised Introduction (**starting at line 84**), PKA is a serine/threonine kinase that regulates various cellular processes, including centrosome function. Reported PKA substrates relevant to centrosomal regulation include Aurora A

kinase, Nde1, centrin, pericentrin, and dynein (Lutrz et al., 2001, Bradshaw et al., 2008, Ong et al., 2018, Walter et al., 2000). We have also added the corresponding references to support this point (see line 86-87).

6. The connection between AKAP9 and AKAP450 should be clarified. Readers most likely do not know that they are the same protein/gene.

We agree that the relationship between AKAP9 and AKAP450 should be clarified for the readers. In the revised manuscript, we have explicitly addressed this point. For example, at line 94-95, we now state that AKAP450 is also known as AKAP9, AKAP350, or CG-NAP. Additionally, at line 226, we clarify that AKAP9 is the gene encoding AKAP450 and describe our siRNA-mediated repression experiments targeting AKAP9 in GPER1-activated cells.

7. AKAP9 is not essential. Considering potential siRNA-related side effects, the authors should confirm their results with CRISPR-Cas9-mediated gene knockout.

Following the reviewer's valuable suggestion, we confirm and strengthen our siRNA knockdown results, by using several AKAP9 knockout clones generated by CRISPR-Cas9 methodology. We assessed centrosome amplification in these knockout clones following activation of GPER1-PKA signaling by treatment with BPA, G-1, or forskolin (see new **Figure 2i** and new **Figure S2g**). Successful AKAP9 knockout was confirmed by Western blot analysis (see new **Figure S2f**).

Importantly, centrosome numbers in AKAP9 knockout cells were restored to control levels, supporting the essential role of AKAP450 in PKA-dependent centrosome amplification. Furthermore, we showed that PKA's recruitment to the centrosome is dependent on AKAP450, as PKA is absent from centrosomes in AKAP450 knockout cells (see new **Figure 2h** and new **Figure S2g**).

Taken together, these findings corroborate a model in which AKAP450 recruits PKA to interphase centrosomes and suggest the existence of a GPER1-PKA-AKAP450 signaling axis regulating centrosome numbers in colorectal cancer cells in response to GPER1 activation.

8. The authors mention several times that the observed effect is related to PKA at centrosomes. However, they never provide any evidence for this. To make these claims they must address the effect of estrogen signaling on centrosome number and PKA localization in AKAP9-deleted cells.

We thank the reviewer for this important comment. As detailed in our response to comment #7, we have addressed this point by generating several AKAP9 knockout clones using CRISPR-Cas9 methodology. In these knockout cells, we evaluated both centrosome amplification and PKA localization at centrosomes upon activation of GPER1-PKA signaling by BPA, G-1, or forskolin treatment.

Our data clearly show that in AKAP450 knockout cells, PKA is no longer recruited to the centrosomes (see new **Figure 2h** and new **Figure S2g**), and importantly, the increase in centrosome number induced by estrogen signaling is abolished (see new **Figure 2i** and new **Figure S2g**). These findings provide direct evidence that PKA localization at centrosomes

depends on AKAP450, and that the estrogen/GPER1-mediated centrosome amplification requires AKAP450-mediated PKA recruitment (see line 222-224 and line 230 onwards). Together, these results strongly support our model of a GPER1-PKA-AKAP450 signaling axis regulating centrosome numbers in colorectal cancer cells. We hope this adequately addresses the reviewer's concern.

9. Does GPER activation or cAMP signaling influence PKA activity specifically at the centrosome or global? If this question cannot be addressed, the authors must discuss this point in the discussion and tone down related claims.

As described in the Materials and Methods section, we used whole cell lysates of asynchronously growing cells to show increased global intracellular cAMP levels and PKA activity upon GPER1 or adenylyl cyclase/PKA activation (see revised **Figures 1e and f**, **Figure 2e and f**, and **Figure S1c**).

By using immunofluorescence microscopy, we confirmed that PKA is localized specifically at centrosomes in DMSO-treated HCT116 cells (see new **Figures 2g and h**). Upon GPER1 activation by BPA or G-1, we observed a marked increase in centrosomal PKA signal intensity, indicating recruitment of the kinase to centrosomes (see new **Figure 2g**). Of note, in AKAP450 knockout cells, PKA failed to localize to centrosomes (see new **Figure 2h** and new **Figure S2g**), suggesting that AKAP450 mediates GPER1-dependent recruitment of PKA. Finally, we show that GPER1 activation causes phosphorylation of Centrin at serine 170 at interphase centrosomes (see revised **Figure 6a-h** and **Figure S6a and b**) and that this phosphorylation is dependent on PKA activity. Thus, inhibition of PKA strongly reduces Centrin-phosphorylation levels (see new **Figure S6d**) and restores centrosome numbers to basal levels (see new **Figure 2c and d**).

Together, these findings demonstrate that increased GPER1-activity causes AKAP450-dependent recruitment of PKA to interphase centrosomes, where the kinase phosphorylate Centrin at serine 170.

10. It is unclear to me how G1 and G2 cells were distinguished in Figure 3a/b?

We thank the reviewer for raising this important point. We agree that a definitive distinction between G1 and G2 phase cells requires the use of cell cycle-specific markers such as Cyclins (see also our response to question #1).

Notably, by using Centrin as a distal centriole marker in combination with PCM-associated γ -tubulin staining, we are able to approximate the cell cycle stage based on centriole duplication status, which correlates with cell cycle progression:

Late S/G2-phase cells typically contain two centrosomes, each with a pair of centrioles, resulting in four Centrin signals, each associated with its own γ -tubulin-positive PCM focus. In contrast, G1-phase cells contain a single centriole pair. However, as shown by Wang et al. (2011), these centrioles are modified during the preceding mitosis in a Plk1-dependent manner, enabling each to recruit PCM individually in early G1. This can lead to two spatially distinct γ -tubulin foci, despite the presence of only two centrioles in total.

While this approach does not provide the same resolution as cyclin-based staging, it serves as a reasonable proxy to differentiate G1 from late S/G2-phase cells based on centriole number and PCM distribution.

To improve clarity and avoid potential misinterpretation, we have revised the text (see line 277 onwards and **Figures 3a** and **S3b** and now explicitly label the corresponding cells as “G1” and “late S/G2”.

11. The conclusions based on PLK4 overexpression are not clear. Instead of showing rosette-like structure, which is not clear at all (Fig. 3J), the authors must quantify centrosome number as they did before (e.g., gamma-tubulin foci after 48h of PLK4 overexpression).

We fully acknowledge that the typical rosette-like arrangement of Centrin foci following PLK4 overexpression shown in the original **Figure 3j** (new **Figure S3e**) may be ambiguous and open to interpretation. However, this immunofluorescence image was intended only as a qualitative supplement to the Western blot data shown in the revised **Figure S3d**, which demonstrates successful PLK4 overexpression. The immunofluorescence image serves to confirm the expected functional consequence of PLK4 overexpression - namely, the formation of rosette-like Centrin structures, as has been described in the literature (habedanck et al., 2005). To address the reviewer's concern, we have now replaced the immunofluorescence image in the revised **Figure S3e**, and shifted the emphasis of our conclusions toward the quantitative analysis in the main **Figure 3g**.

As suggested, we have quantified centrosome numbers after 48 hours of PLK4 overexpression using γ -tubulin staining, consistent with our previous analyses. The original **Figure 3k** and **l** (now revised **Figure 3g**) presents this data. Specifically, the bar graph indicates the percentage of cells exhibiting centrosome amplification, shown as grey and purple bars. In addition, we highlighted the proportion of these centrosome-amplified cells that exhibit enlarged Centrin foci, shown as white segments within each bar.

We hope that these clarifications and the revised figure address the reviewer's concerns.

12. From the data shown in Figure S3f-g, I'm not convinced that all extra centrin foci are centrioles. There is no quantification for S3f-g. CP110 and CEP135 are co-stained with gamma-tubulin, but not with Centrin. In the main figures, it is shown that Centrin foci do not always co-localize with gamma tubulin. Thus, they are most likely not centrioles. The authors have to show co-localization of centrin with other centriole markers to claim that all centrin foci are centrioles.

We thank the reviewer for this important and insightful comment. We fully agree that it is crucial to rigorously examine whether the Centrin foci observed in our experiments correspond to bona fide centrioles, and to consider the possibility that they may instead represent Centrin-positive structures that are not true centrioles. Please see our response to the similar reviewer comment #1.

13. The PKA inhibitor PKI only had a marginal effect on Centrin phosphorylation (Fig. S4a,b), yet the authors also show that PKI completely rescues centrosome amplification upon estrogen treatment (Fig. 2). To support the claim that PKA-mediated phosphorylation of centrin leads to extra centrin foci, the authors have to show the effect of PKA inhibition (PKI) on the phosphorylation of centrin in interphase cells.

We thank the reviewer for this excellent suggestion, which we have addressed in the revised manuscript through additional experiments.

Establishing a direct causal link between PKA-mediated Centrin phosphorylation and the formation of enlarged Centrin foci in interphase cells is technically challenging, as phospho-Centrin (Ser170) is predominantly detectable at amplified centrosomes, which are largely absent following PKA inhibition (see revised **Figure 2c, d**).

To overcome this limitation, we applied delayed PKA inhibition using PKI during the final 5 hours of G-1 treatment, allowing sufficient time for initial centrosome amplification to occur. As shown in new **Figure 6i** and new **Figure S6d**, this treatment reduced phospho-Centrin levels from ~50% to ~10%. However, the size of Centrin foci remained unchanged, suggesting that the maintenance of enlarged foci does not depend on continued PKA activity.

To further dissect the role of Centrin phosphorylation, we re-expressed either wild-type human Centrin-2 or a non-phosphorylatable S170A mutant in Centrin-2 knockout cells. Protein expression and phospho-status were confirmed by Western blot and IF (see new **Figure 6j**, and new **Figure S6c**). Upon BPA treatment, only wild-type Centrin restored centrosome amplification (see new **Figure 6l**), indicating that phosphorylation at Ser170 is required for this process. However, both wild-type and mutant Centrin induced a similar frequency of enlarged Centrin foci (see new **Figure 6k**), indicating that S170 phosphorylation is not required for the structural remodeling of Centrin foci per se. Interestingly, the typical Centrin displacement observed upon GPER1 activation (see revised **Figure 3c**) was largely absent in cells expressing the S170A mutant (see new **Figure S6e**). This finding aligns with prior work by Lutz et al. (2001), which showed that PKA activity promotes centriole separation in interphase cells.

In summary, our data support a model in which PKA-mediated phosphorylation of Centrin-2 is essential for initiating centrosome amplification and structural remodeling, but not for the maintenance of these changes. This points to a two-step mechanism, with PKA acting as an initial priming factor, followed by stabilization via additional, PKA-independent pathways.

14. The authors “hypothesize that Centrin-2 and PKA physically interact at centrosomes” (line 378). However, they never test this idea with appropriate methods that can determine the location of interaction (e.g.; Proximity ligation assay).

To directly assess the proposed interaction between Centrin-2 and PKA at the centrosome, we performed a proximity ligation assay (PLA) in colorectal cancer cells as suggested by the reviewer. This sensitive in situ approach confirmed close spatial association between Centrin and the catalytic subunit of PKA (PKA-C α) in both DMSO- and BPA-treated HCT116 cells (see new **Figure 7g**). To determine the subcellular localization of these interactions, we combined PLA with γ -tubulin immunostaining, which allows precise visualization of

centrosomes. The PLA signals co-localized with γ -tubulin-positive centrosomes in interphase cells, supporting a direct interaction of Centrin and PKA-C α at interphase centrosomes. These results not only support our findings received from our immunoprecipitation studies (see revised **Figures 7e and f**), which show that phospho-Centrin binds to PKA-C α . The observed interaction between PKA and Centrin shown by the PLA-assay further support our immunofluorescence studies, which demonstrate increased PKA-specific phosphorylation of enlarged Centrin foci at amplified interphase centrosomes upon GPER1 activation (see new **Figure S6d**). Interestingly, while PKA localization to centrosomes increases upon GPER1 activation (see new **Figure 2g**), displaced centrosomes seem to lack detectable PLA signals (see new **Figure 7g**), consistent with our previous immunofluorescence findings that such centrosomes are devoid of PKA (see new **Figure S2d**). These results provide direct spatial evidence supporting our hypothesis of a centrosomal Centrin-PKA interaction and further suggest that this interaction results in phosphorylation of Centrin at serine 170, but is lost upon GPER1/PKA-induced centrosome displacement.

15. I don't understand this sentence "Of note, while Hsp90, a well-known binding partner of PKA β , co-immunoprecipitated with PKA regulatory subunits, Centrin-pS170 did not interact with PKA-R11 α , as expected" (lines 392,393).

We acknowledge that the original phrasing was unclear and have revised the sentence (starting at line 649) as follows:

"Of note, Hsp90, a well-known binding partner of PKA β , co-immunoprecipitated with PKA-R11 α , confirming the functionality and specificity of the immunoprecipitation protocol. In contrast, Centrin-pS170, as expected²⁴, did not associate with PKA-R11 α under these conditions."

This revised version clarifies that Hsp90 served as a positive control for the immunoprecipitation, confirming that the assay was functional. Centrin-pS170, by contrast, showed no interaction with the regulatory subunit PKA-R11 α , consistent with previously published findings.

16. What does the asterisk show in Figure 6b-h?

The asterisks indicate the light and heavy chains of the immunoprecipitation antibody, as noted in the figure legend of the original **Figure b-h** (revised **Figure 7b-f, and h-i**). We apologize if this was not sufficiently clear and have adjusted the legend slightly to improve clarity.

17. Line 449/450: "In this study, we have now filled this gap by showing that activated GPER1 triggers the recruitment of PKA to the centrosome, where the kinase affects the centriole size and centrosome number." The authors do not show that activated GPER1 triggers recruitment of PKA to centrosomes. They also do not show that GPER1 influences centriole size.

To clarify the mechanism by which GPER1 activation affects centrosomal structure and function, we conducted additional experiments addressing both aspects raised by the reviewer.

First, using immunofluorescence microscopy, we confirmed that PKA is localized at centrosomes in DMSO-treated HCT116 cells (see new **Figures 2g and h**). Upon GPER1 activation by BPA or G-1, we observed a marked increase in centrosomal PKA signal intensity, indicating recruitment of the kinase to centrosomes (see new **Figure 2g**). Importantly, in AKAP450 knockout cells, PKA failed to localize to centrosomes (see new **Figure 2h** and new **Figure S2g**), suggesting that AKAP450 mediates GPER1-dependent recruitment of PKA. These findings support the existence of a functional GPER1-PKA-AKAP450 axis in centrosome regulation.

Second, to determine whether GPER1 influences centriole structure, we performed transmission microscopy (TEM) to assess centriole morphology and size. In DMSO-treated cells, centrioles exhibited normal barrel-shaped ultrastructure with dimensions consistent with previous reports (diameter ~220 nm, length ~409 nm; **Fig. 4f**). In contrast, BPA-treated cells showed frequent abnormalities, including disrupted microtubule walls and asymmetric protrusions (**Fig. 4e**). Quantitative analysis revealed a significant reduction in centriole length with an average of ~322 nm and minimum values as low as 114 nm, while diameter and width remained unchanged (**Fig. 4f**).

Together, these results demonstrate that GPER1 activation leads to PKA recruitment to the centrosome and induces structural abnormalities in centrioles, including a measurable reduction in their length. These findings directly address the reviewer's comment and support the conclusion presented in the revised manuscript.

18. Line 86: References 24 and 27 do not show that PKA localizes at centrosomes. Reference 26 is a 22-year-old review that cites one paper from 1999 (PMID: 10328961) that claims that PKA localizes at centrosomes with the resources that were available at that time. Considering the importance of PKA for the conclusion of this paper the authors should provide the most current and most solid references that show the localization of PKA at centrosomes.

We appreciate the reviewer's insightful comment and agree that the original references did not adequately demonstrate the centrosomal localization of PKA.

Accordingly, we have revised the Introduction (**starting at line 84**) to better reflect the current understanding of PKA's role in centrosome regulation: "PKA is a serine-threonine protein kinase complex that governs numerous cellular processes through substrate phosphorylation, including those involved in centrosome function, such as Aurora A kinase, Nde1 (neurodevelopment protein 1), centrin, pericentrin and dynein^{24, 26, 27, 28}."

Furthermore, **starting at line 94**, we clarify: "AKAP450 (also known as AKAP9/350 or CG-NAP) and PKA localize at the centrosome, suggesting a potential role for AKAP450 in recruiting PKA^{27, 34, 35, 36, 37}."

While direct evidence for PKA localization at centrosomes is somewhat limited, several recent studies have demonstrated phosphorylation of centrosomal proteins by PKA in

different cellular contexts (Centrin, Lutz et al., 2001 [24]; NDE1, Bradshaw et al., 2008 [26]; pericentrin and dynein, Ong et al., 2018 [27]; Aurora A, Walter et al., 2000 [28]).

More direct evidence includes:

Terrin et al., 2012 [35], who developed PKA-GFP biosensors targeted to centrosomes showing clear centrosomal localization;

Vandame et al., 2014 [37], reporting high concentration of the PKA catalytic subunit at centrosomes during mitosis in HeLa cells;

Ong et al., 2018 [27], demonstrating AKAP450-mediated coordination of PKA localization and activity at the centrosome and Golgi in T-cells.

Importantly, our own immunofluorescence data in HCT116 cells confirm PKA localization at centrosomes under basal conditions, with increased enrichment upon GPER1 activation (see new **Figures 2g and h**), further supporting a centrosomal role for PKA in this system. In the Results section, we updated the text to state (see line 216 onwards):

“PKA localizes to diverse cellular compartments by binding to different A kinase-anchoring proteins (AKAPs)³³. AKAP450 localizes with PKA at the centrosome, suggesting a potential role as a docking platform for PKA^{27, 34, 35, 36, 37}. By using immunofluorescence microscopy to analyze intracellular PKA localization, we confirmed not only the centrosomal localization of PKA in DMSO-treated HCT116 cells (**Fig. 2g, h**), but also observed a marked increase in centrosomal PKA levels upon GPER1 activation with BPA or G-1 (**Fig. 2g**).”

We hope these additions adequately address the reviewer’s concerns.

19. Line 96: References 24, 27 and 32 do not show that AKAP450 recruits PKA to centrosomes. Please provide references that support the provided information that is used for the argumentation.

We agree that the original references did not adequately demonstrate AKAP450-dependent recruitment of PKA to centrosomes.

Accordingly, we have revised the Introduction (starting at line 94) and included to better reflect the current understanding of PKA’s role in centrosome regulation (please also see answer to reviewer comment #18):

“AKAP450 (also known as AKAP9/350 or CG-NAP) and PKA localize at the centrosome, suggesting a potential role for AKAP450 in recruiting PKA^{27, 34, 35, 36, 37}.”

20. Line 111-112: The authors write “Upon activation, PKA is recruited to centrosomes via AKAP-450, where it orchestrates the phosphorylation of Centrin-2 at serine 170”. However, the authors don’t provide any evidence for their claim that PKA is recruited to centrosomes following GPER1 activation.

We appreciate the reviewer’s insightful comment and have addressed it by investigating PKA localization at interphase centrosomes both after GPER1 activation and in AKAP450-deficient cells, using AKAP9 knockout cells generated with CRISPR-Cas9 technology. Using immunofluorescence microscopy, we confirmed PKA localization at centrosomes in DMSO-treated HCT116 cells (see new **Figures 2g and h**). Upon GPER1 activation by BPA

or G-1, centrosomal PKA signal intensity increased significantly, consistent with recruitment of PKA to the centrosome (see new **Figure 2g**). In AKAP450 knockout cells, PKA was absent from centrosomes, supporting a model in which AKAP450 mediates GPER1-dependent PKA recruitment. The data are shown in the new **Figures 2h and S2f and g**. Accordingly, we updated the Results section (**line 216 onwards**) to state: “PKA localizes to diverse cellular compartments by binding to different A kinase-anchoring proteins (AKAPs)³³. AKAP450 localizes with PKA at the centrosome, suggesting a potential role as a docking platform for PKA^{27, 34, 35, 36, 37}. By using immunofluorescence microscopy to analyze intracellular PKA localization, we confirmed not only the centrosomal localization of PKA in DMSO-treated HCT116 cells (see new **Figure 2g and h**), but also observed a marked increase in centrosomal PKA levels upon GPER1 activation with BPA or G-1 (see new **Figure 2g**). “

We hope these additions comprehensively address the reviewer's concerns.

21. Line 113: The authors write that centrin phosphorylation by PKA “occurs inappropriately at non-mitotic amplified centrosomes”. As described above, there is no evidence that shows that the centrin foci belong to amplified centrosomes.

We fully agree that it is essential to rigorously demonstrate whether the observed Centrin foci correspond to bona fide amplified centrosomes, as Centrin-positive structures may not necessarily represent true centrioles.

To address this important point, we performed additional experiments (detailed in our response #1) that provide further evidence supporting the assignment of these Centrin foci as amplified centrosomes within structurally abnormal centriole-like assemblies.

We appreciate the reviewer's constructive suggestion, which helped us to strengthen both our analysis and interpretation.

22. Line 125-127: The authors write “GPER1 binding to G protein $G_{\alpha s}$ triggers the activation of PKA, which is recruited to centrosomes”. However, none of the provided references (26, 43, 44, 45) provides any evidence that support this claim.

Following the reviewer's concern, we revised the Results section (**line 126 onwards**) to state: “GPER1 binding to G protein $G_{\alpha s}$ triggers the activation of PKA, which regulates various cellular processes, including mitosis^{37, 48, 49}.” Appropriate reviews summarizing the connection between GPER signaling and PKA is provided by Prossnitz et al., 2023 [⁴⁸] and Bushi et al., 2025 [⁴⁹]. Evidence for PKA playing a role during mitosis comes from Vandame et al., 2014 [³⁷].

23. Line 176: The authors write that “upon activation, PKA catalyzes the phosphorylation of a variety of proteins including those involved in centrosome function”. Reference 19 does not mention PKA. Reference 24 shows centrin phosphorylation. Reference 26 mention only centrin as relevant centrosomal target of PKA. Thus, I could only find centrin as centrosomal target of PKA. Are there other publications that support the claim that PKA phosphorylates a variety of centrosomal proteins?

We thank the reviewer for this important observation. To more accurately reflect the current literature, we have revised the text starting at **line 199** as follows:

“Upon activation, PKA catalyzes the phosphorylation of a variety of proteins including those associated with the centrosome^{24, 26, 27, 28.}”; and at **line 84 onwards**, as follows:

“PKA is a serine-threonine protein kinase complex that governs numerous cellular processes through substrate phosphorylation, including those involved in centrosome function, such as Aurora A kinase, Nde1 (neurodevelopment protein 1), centrin, pericentrin and dynein^{24, 26, 27, 28.}”

Corresponding references have been updated and include:

Centrin: Lutz et al., 2001 [²⁴]

NDE1: Bradshaw et al., 2008 [²⁶]

Pericentrin and dynein: Ong et al., 2018 [²⁷]

Aurora A: Walter et al., 2000 [²⁸]

These studies demonstrate that PKA can phosphorylate a variety of centrosome-associated proteins in different cellular contexts, supporting our revised statement.

We believe this clarification strengthens the manuscript and provides more accurate and up-to-date references.

24. Line 192-193: None of the references 26, 32 and 44 shows that AKAP450 recruits PKA to centrosomes.

We agree with the reviewer that the originally cited references (26, 32, and 44) do not directly support the statement that AKAP450 recruits PKA to centrosomes. We have therefore revised the sentence and included more appropriate literature to support this claim. The revised text (**starting at line 216**) now reads:

“PKA localizes to diverse cellular compartments by binding to different A kinase-anchoring proteins (AKAPs)³³. AKAP450 localizes with PKA at the centrosome, suggesting a potential role as a docking platform for PKA^{27, 34, 35, 36, 37.}”

In addition to these references, our own data support this model. Using immunofluorescence microscopy, we observed centrosomal localization of PKA in HCT116 cells, which was markedly increased upon GPER1 activation with BPA or G-1 (see new **Figure 2g**).

Furthermore, in AKAP450 knockout cells generated via CRISPR-Cas9, centrosomal recruitment of PKA was abolished (see new **Figures 2h** and **Figure S2g**), demonstrating that AKAP450 is required for centrosomal targeting of PKA in this context.

Please also see answer to question #19.

25. Line 342-343: The authors create the impression that PKA is the major kinase phosphorylating Centrin. However, Figures S4a-b indicate that the majority of phosphorylation is independent of PKA. Please rephrase to make this point clear.

We appreciate the reviewer’s important comment. We fully agree that Centrin-2 phosphorylation is likely mediated by multiple kinases depending on the cellular context (e.g., interphase vs. mitosis). Besides PKA, Aurora A kinase also phosphorylates Centrin-2

at Ser170, regulating its stability, centrosome amplification, and centriole separation (Lukasiewicz et al., 2011; Lutz et al., 2001; line 79ff and line 822ff). Importantly, these points are already addressed in the original manuscript both in the Introduction (original line starting at 79ff) and in the Discussion (original line starting at 492ff), where we highlight that both PKA and Aurora A target Ser170 of Centrin-2. Thus, no further changes to these sections were necessary.

We acknowledge that PKA inhibition only modestly reduces overall Centrin-2 phosphorylation (see revised **Figure S5c and d**). Therefore, we rephrased the relevant sentence to: “Notably, inhibition of PKA led to a reduced proportion of phospho-Centrin-2 at mitotic centrosomes (**Fig. S5c, d**).”, (see line 516 onwards). Importantly, our new data show that during interphase - where Centrin enlargement and centrosome amplification are observed - PKA inhibition dramatically reduces phospho-Centrin-2 levels following GPER1 activation (from ~50% to 5-10%; see new **Figure S6d**). This suggests a dominant role of the GPER1-PKA-Centrin axis specifically in interphase.

Furthermore, re-expression of wild-type Centrin-2, but not a non-phosphorylatable S170A mutant, in Centrin-2 knockout cells restored centrosome amplification upon BPA treatment (see new **Figure 6l**), indicating that Ser170 phosphorylation is essential for this process. Both variants induced enlarged Centrin foci, however, suggesting that phosphorylation is not required for initial structural remodeling (see new **Figure 6k**). Interestingly, the Centrin displacement typically induced by GPER1 activation was largely absent in S170A-expressing cells (**Figure S6e**), consistent with previous findings linking PKA activity to centriole separation (Lutz et al. (2001).

In summary, our data support a two-step model: PKA-mediated Centrin-2 phosphorylation at Ser170 initiates centrosome amplification and remodeling, but maintenance likely involves additional, PKA-independent pathways.

26. Line 361 and 372: There is no evidence that centriole size is altered by PKA.

We thank the reviewer for this important observation. We agree that our previous wording may have unintentionally conflated “enlarged Centrin foci” with “enlarged centrioles,” which could lead to misinterpretation. To correct this imprecision, we have revised the manuscript by replacing terms such as “enlarged centrioles” or “enlarged Centrin-labeled centrioles” with more accurate descriptions like “enlarged Centrin-2,” “enlarged Centrin-2 foci,” or “enlarged Centrin-labeled foci.”

As shown by our new transmission electron microscopy (TEM) data (see new **Figure 4e, f**), GPER1 activation leads to centrioles with morphological abnormalities, including irregular protrusions, loosened microtubule walls, and a reduction in overall centriole size (also see answer #17). Accordingly, we revised the text throughout the manuscript to clearly distinguish between Centrin foci, which are enlarged upon GPER1/PKA activation, and centrioles, which are structurally altered and reduced in size upon GPER1 activation. This clarification better reflects the experimental data and addresses the reviewer’s concern.

27. Line 428-431: The authors don’t provide any evidence that GPER1 triggers the recruitment of PKA by AKAP450 to centrosomes.

We are now pleased to provide new data that directly address these important points: (i) We demonstrate that PKA recruitment to centrosomes is mediated by AKAP450, as shown in new **Figure 2h** and **Supplementary Figures S2f and S2g**. (ii) We show binding and a physical interaction between Centrin and PKA (revised **Figures 7e, f, h, and i**, and new **Figure 7g**). (iii) Finally, we provide evidence that PKA phosphorylates centriolar Centrin-2 inappropriately during interphase (new **Figure 6j** and **Supplementary Figure S6d**). Together, these findings support the existence of a functional GPER1-AKAP450-PKA axis that regulates centrosomal signaling and structure in interphase cells.

28. Line 218-220, 203-268, 432-433, 443, 450, 451, 460, 466-467, 476, 486-488, 498-499: The size of the centrin foci cannot be used to conclude the length or size of centrioles. The authors do not provide any conclusive evidence that centrioles are enlarged.

We thank the reviewer for this important comment. We fully agree that Centrin foci size alone cannot be used to infer centriole length or size. To address this, we performed transmission electron microscopy (TEM), as described in our responses to comments #17 and #26. These new data provide direct ultrastructural evidence of centriole morphology changes upon GPER1 activation, including aberrant structural features and a significant reduction in centriole length, which clarify and support our conclusions regarding centriole alterations.

We agree that our previous wording may have unintentionally conflated "enlarged Centrin foci" with "enlarged centrioles," which could lead to misinterpretation. To correct this imprecision, we have revised the manuscript as mentioned in response #26 by replacing terms such as "enlarged centrioles" or "enlarged Centrin-labeled centrioles" with more accurate descriptions like "enlarged Centrin-2," "enlarged Centrin-2 foci," or "enlarged Centrin-labeled foci."

The reviewer is absolutely correct and we thank for this comment.

Reviewer #3 (Remarks to the Author):

A GPER1-PKA-Centrin axis regulates centrosome numbers and centriole integrity in colon cancer cells

Centrosome number control is a critical aspect of cell division and an intriguing area of research. During normal cell division, centrioles duplicate once per cell cycle during the S phase and subsequently mature into centrosomes during G2 and mitosis. Aberrations in centrosome number, frequently observed in cancer cells, contribute to chromosome mis-segregation and tumor progression. Mechanisms underlying centrosome amplification include the upregulation of PLK4 kinase, defects in cytokinesis, centrosome fragmentation, or de novo centriole formation. This manuscript by Fahrlander et al. explores the role of GPER1 and its interaction with protein kinase A (PKA), facilitated by the adaptor protein AKAP450 localized at centrosomes, in regulating centrosome number and integrity. The au-

thors report that PKA phosphorylates Centrin-2, which may contribute to centrosome abnormalities observed in colon cancer cell lines. The study employs a range of approaches, including the use of antagonists, agonists, and GPER1-activating hormones, with centrosomes analyzed 48 hours after treatment initiation. Statistical analyses robustly support the findings. Centrosome amplification during the 48-hour period increases moderately, from 2% to 8%. The data convincingly demonstrate that GPER1 activation via PKA (mediated by cAMP) and its binding to AKAP450 induces the appearance of extra gamma-tubulin (at the pericentriolar material and within centrioles) and centrin-2 (components of SF11 and POC5 complexes) signals. Intriguingly, these signals do not always co-localize upon centrosome amplification, unlike the control cells. This observation suggests centriole fragmentation or protein accumulation, akin to phenomena seen with overexpression of CEP250 or centrobilin. Furthermore, amplified centrin-2 signals frequently appear as "super spot structures," interpreted as signs of centriole enlargement. This conclusion, supported by conventional immunofluorescence imaging (Fig. 3c), would benefit from confirmation via electron microscopy (EM) analysis using conventional serial sections. Such analysis could also reveal additional aberrations in centriole structure following GPER1 activation. The authors also demonstrate that centrin-2 phosphorylation at Ser170 is mediated by the GPER1-PKA pathway. In cells exhibiting additional centrin-2 signals, a significant fraction shows phosphorylation at Ser170, further linking this modification to centrosome abnormalities.

Conclusion and Recommendations

This manuscript offers important insights into centrosome regulation via the GPER1-PKA-AKAP450 pathway. While the findings are compelling, further validation using EM and improved ultrastructure expansion microscopy (u-ExM) to analyze additional centrin-2 and gamma-tubulin signals is essential. These additional analyses will strengthen the evidence and provide deeper structural understanding of the observed centrosome anomalies. Overall, this study provides a significant contribution to the field and merits publication following major revision.

We thank the reviewer for the valuable comments and appreciate the concerns raised regarding the relation of extra Centrin positive foci and bona fide centrioles. As suggested, we have carefully addressed this point by performing additional immunofluorescence and transmission electron microscopy (TEM) experiments and revising the relevant sections of the manuscript accordingly. A detailed response to this important issue is provided in our point-by-point reply below.

Major points

1. The authors use HCT116 and HCT-15 cells as models. How general are the findings of this manuscript?

We acknowledge the limitation of using only two CRC cell lines and recognize the importance of assessing the generalizability of our findings. HCT116 and HCT-15 were selected based on their genetic backgrounds, relevance to CRC biology, and defined

baseline levels of centrosome amplification (CA), allowing for robust and reproducible analysis of GPER1-dependent centrosome dynamics.

Our focus on CRC is intentional: as a non-classically hormone-driven tissue, the colon provides a unique context in which to explore the role of estrogen signaling in tumorigenesis. In our previous study, we already demonstrated a GPER1-induced increase in CA and whole-chromosome instability (w-CIN)/aneuploidy in a CRC-derived cell system using HCT116 and a non-transformed epithelial model (Bühler et al., 2023). The current study extends these findings by delineating a downstream signaling pathway, namely the GPER1-PKA-Centrin-2 axis, that contributes to centrosome dysfunction.

Supernumerary centrosomes and aberrant centriole structures are known to promote w-CIN, aneuploidy, and invasiveness through increased microtubule nucleation and elevated RAC1 activity (Ganem et al., 2009; Godinho et al., 2014). Enlarged Centrin-positive structures may similarly enhance centrosomal activity and disrupt chromosomal segregation. Together with our previous findings on xenoestrogen-induced GPER1 activation (Bühler et al., 2023), the involvement of Centrin-2 and the upstream signaling components identified here suggest broader biological relevance. Supporting this, prior studies have shown elevated Centrin phosphorylation in breast tumors with centrosome amplification (Lingle et al., 1998; Lukasiewicz et al., 2011), indicating that similar mechanisms may operate in other hormone-influenced malignancies.

While our data are limited to in vitro CRC models, the identified pathway likely reflects a broader mechanism. Nonetheless, further validation in additional CRC subtypes, patient-derived models, and in vivo systems will be important to fully assess broader applicability.

2. The authors have to confirm centrosome amplification using electron microscopy and improved u-ExM. U-ExM in Fig. 2e is of insufficient quality.

We thank the reviewer for the valuable feedback regarding the need to confirm centrosome amplification using electron microscopy and improved u-ExM.

Given the limited time available during the revision process, we prioritized transmission electron microscopy (TEM) over improved u-ExM imaging to provide high-resolution insights into centriole ultrastructure following GPER1 activation. We believe that the TEM data offer a robust and conclusive structural assessment.

Our new data, reveal significant structural abnormalities in BPA-treated cells, such as loosening of microtubule walls and the presence of irregular appendage-like extensions, compared to DMSO-treated controls (new **Figure 4e, f and Figure S4b**). Quantification of centriole size measurements showed an unexpected notable *reduction* in centriole length in BPA-treated cells, which does not correspond with the presence of *enlarged* Centrin foci observed upon GPER1 activation (see revised **Figure 3**). However, together with our new immunofluorescence analyses showing that GPER1 activation leads to an incomplete composition of centriole-specific proteins at these centriole-like structures containing enlarged Centrin foci (new **Figures 4b and 4c**), these data suggest that the enlarged Centrin signals do not represent intact centrioles, but rather structurally aberrant or immature centriole-like assemblies.

Further analysis of centriole number in BPA-treated cells revealed only a single cell containing three centrioles, which is likely attributable to the relatively low number of cells analyzed. However, among the 72 BPA-treated cells examined, 24.1% exhibited two centrioles per cell, whereas over 90% of DMSO-treated control cells harbored only one centriole per cell (**Figure S4c**). Given that GPER1 activation does not affect cell cycle progression or proliferation (Bühler et al., 2023; and this study), the increased number of centrioles compared to control cells is unlikely to reflect cell cycle-dependent fluctuations. Rather, these findings support a GPER1-dependent induction of centriole amplification.

Together, these findings indicate that GPER1 signaling induces structurally aberrant centriole-like assemblies, which are distinct from both intact centrioles and previously described ectopic Centrobin structures.

In light of these results, we have updated the manuscript accordingly, using more precise terminology, such as "enlarged Centrin-2 foci" rather than unequivocally referring to them as "centrioles", to clarify the nature of these structures.

Accordingly, we updated the Results (line 389 onwards) and Discussion section (line 783 onwards). We kindly refer the reviewer to our responses to reviewer reviewer 2's comments #1, #12, and #17 for further details regarding the TEM analyses.

3. AKAP450 is indicated as a 40 kDa protein in Fig. 2j. AKAP450 has a molecular weight of 450 kDa. Depletion efficiency is moderate. On p. 5 the authors state that AKAP450 knock-down is more efficient in HCT-15 cells. I cannot see this in Fig. 2j.

a. We thank the reviewer for this careful observation. The incorrect labeling of AKAP450 as a 40 kDa protein in the original Western blot shown in **Figure 2j** has been corrected in the revised version (revised **Figure S2c**).

b. We agree with the reviewer's comment regarding the depletion efficiency and have removed the corresponding sentences from the manuscript, as they were not supported by the data presented in **Figure 2j**.

4. The P-centrin2 IP with PKACalpha in Fig. 6 is not so clear mainly because of the IgG light chain. The authors should repeat the IB using TrueBlot (specific to whole IgG).

We thank the reviewer for this valuable comment. We agree that the immunoglobulin light chain in the Centrin-pS170 antibody Western blot (original **Figure 6**) compromised the clarity of the specific phospho-Centrin signal. Although we attempted to implement TrueBlot methodology, we unfortunately lost immunoprecipitated signals in the process. To address the reviewer's issue, we refined our experimental approach by incorporating IgG control immunoprecipitations (see new **Figure S7c**). These new experiments provide more accurate controls and confirm the specificity of the phospho-Centrin binding to PKA-C α in both HCT116 and HCT-15 interphase cells following estrogenic stimulation. We believe these updated controls strengthen the interpretation of our results.

5. What happens to centrosomes when a phospho-inhibitory centrin2-Ser170Ala is expressed in combination with GPER1 activation?

As suggested by the reviewer, we re-expressed either wild-type human Centrin (WT) or a non-PKA-phosphorylatable Centrin mutant (S170A) in Centrin-2 knockout cells, generated by CRISPR-Cas9 technology (see new **Figure 6j**, new **Figure S6c** and new **Figure S5a**). Upon BPA treatment, wild-type Centrin induced centrosome amplification, while the S170A mutant did not (see new **Figure 6l**). However, the frequency of enlarged Centrin foci was similar in both wild-type and mutant-expressing cells (see new **Figure 6k**). This suggests that phosphorylation at Ser170 is necessary for centrosome amplification, but not for Centrin enlargement itself.

Interestingly, the typical Centrin displacement observed in GPER1-activated cells (see revised **Figure 3c**) was significantly reduced in cells expressing the non-phosphorylatable Centrin mutant (see new **Figure S6e**). This finding is consistent with previous work by Lutz et al. (2001), which demonstrated that PKA activation promotes centriole separation via centrosomal phosphorylation in interphase cells (Lutz et al., 2001).

To further investigate whether PKA activity is required to maintain enlarged Centrin foci, we applied delayed PKA inhibition during the final 5 hours of G-1 treatment (see new **Figure 6i** and **Figure S6d**). In GPER1-activated cells, this led to a decrease in Ser170 phosphorylation from approximately 50% to ~10% (see new **Figure S6d**), but the size of Centrin foci remained unchanged. However, because the enlargement of Centrin foci and phosphorylation are coupled to centrosome amplification, which is rescued by PKA inhibition (**Figure 2c, d**), we are unable to examine the induction of the phenotype in the absence of PKA activity. The persistence of enlarged Centrin foci in the presence of PKA inhibition suggests that the maintenance of these foci is independent of sustained PKA activity.

In summary, our data support a model in which PKA-mediated phosphorylation of Centrin-2 is crucial for centrosome amplification and initiating structural changes at the centrosome, but not for maintaining those changes. This indicates a two-step mechanism where PKA activity primes centrosomal remodeling, followed by stabilization through additional, PKA-independent pathways.

Minor points

1. The authors use AKAP450 and AKAP9. They should use AKAP450 throughout the manuscript.

We agree with the reviewer that the relationship between AKAP9 and AKAP450 should be clearer for the readers. In the revised manuscript, we have explicitly addressed this point to avoid any confusion. For example, at **line 94-95**, we now clarify that AKAP450 is also known as AKAP9, AKAP350, or CG-NAP. Additionally, at **line 226**, we specify that AKAP9 is the gene encoding AKAP450 and provide a detailed explanation of our siRNA-mediated knockdown experiments targeting AKAP9 in GPER1-activated cells. This revision ensures that AKAP450 and AKAP9 are consistently and accurately referred to throughout the manuscript.

2. The p-centrin2 signal in Fig. 5b is weak.

We appreciate the reviewer's comment regarding the weak p-Centrin2 signal in the original **Figure 5b**. To address this, we have replaced the original **Figure 5b** with a revised version (**Figure 6b**), which now shows a clearer and more robust p-Centrin signal. We believe this revision improves the clarity and interpretation of the data.

3. AKAP450 is not associated with centrosomes in mitosis. How is centrin2 phosphorylated in mitosis? AuroraA?

We thank the reviewer for this important question. Centrin-2 phosphorylation is indeed mediated by multiple kinases, depending on the cellular context (e.g., interphase vs. mitosis). In addition to PKA, Aurora A kinase also phosphorylates Centrin-2 at Ser170, which plays a crucial role in regulating its stability, centrosome amplification, and centriole separation during mitosis (Lukasiewicz et al., 2011, Lutz et al., 2011; see line 79ff and line 822-825). We would like to highlight that these points are already addressed in the original manuscript. Specifically, in the Introduction (line 79) and the Discussion (starting at line 492), we mention that both PKA and Aurora A target Ser170 of Centrin-2, which we believe clarifies the role of these kinases in Centrin-2 phosphorylation.